# HIGH-DIMENSIONAL HETTMANSPERGER-RANDLES ESTIMATOR AND ITS APPLICATIONS

## ABSTRACT

The classic Hettmansperger-Randles estimator has found extensive use in robust statistical inference. However, it cannot be directly applied to high-dimensional data. In this paper, we propose a high-dimensional Hettmansperger-Randles estimator for the location parameter and scatter matrix of elliptical distributions in high-dimensional scenarios. Subsequently, we apply these estimators to two prominent problems: the one-sample location test problem and quadratic discriminant analysis. We discover that the corresponding new methods exhibit high effectiveness across a broad range of distributions. Both simulation studies and real-data applications further illustrate the superiority of the newly proposed methods.

## 1 INTRODUCTION

Estimating the mean vector and covariance matrix is a fundamental task in statistics. In low-dimensional settings, when the data are multivariate normal, the sample mean and covariance matrix are efficient estimators (Härdle et al., 2007). Their performance, however, deteriorates under deviations from normality, motivating the development of robust alternatives. For elliptical distributions, robust estimators such as the spatial median for location and Tyler's scatter matrix for dispersion have been extensively studied (Oja, 2010). Furthermore, Hettmansperger & Randles (2002) proposed a unified procedure for jointly and robustly estimating both location and scatter.

The increasing prevalence of high-dimensional data in areas such as genomics and finance has introduced new challenges. When the number of features approaches or exceeds the sample size, traditional estimators like the sample covariance matrix become singular and non-invertible. This has spurred extensive research on high-dimensional covariance estimation, including thresholding, regularization, and shrinkage techniques (Bickel & Levina, 2008a;b); for a comprehensive overview, see Fan et al. (2016). Nevertheless, these approaches largely based on the sample covariance matrix, and thus are not robust to heavy-tailed distributions.

To address these challenges, robust estimation techniques under elliptical distributions, which naturally accommodate a broad class of heavy-tailed models such as the multivariate $t$-distribution and certain multivariate normal mixtures (those with a common mean and proportional covariances), have garnered increasing attention in high-dimensional statistics. Several studies have explored the properties of the sample spatial median and its use in high-dimensional sphericity testing and location parameter testing problems, including Zou et al. (2014), Li & Xu (2022), and Cheng et al. (2023). However, these estimators are not scalar invariant. To address this issue, scale-invariant spatial median estimators (Feng et al., 2016; Feng & Sun, 2016; Liu et al., 2024) were developed as extensions of the simultaneous estimation framework of Hettmansperger & Randles (2002). However, these approaches are not affine invariant with respect to scatter transformations, limiting their flexibility and applicability in practice. In parallel, robust scatter estimation has advanced through the study of spatial-sign covariance matrices, known for their affine equivariance. Recent works have developed linear shrinkage methods tailored for high-dimensional settings (Raninen et al., 2021; Raninen & Ollila, 2021; Ollila & Breloy, 2022; Ollila, 2024), and sparse precision matrix estimation based on spatial-sign covariance (Lu & Feng, 2025), extending previous advances such as Cai et al. (2011) and Yuan & Lin (2007). However, most existing methods address location and scatter matrix separately, lacking a unified framework that integrates both aspects in high dimensions.

Motivated by these limitations, we propose a novel framework for robust high-dimensional inference. Specifically, we introduce the high-dimensional Hettmansperger-Randles (HR) estimator, from which both the spatial median and the scatter matrix estimators inherit affine equivariance. The resulting spatial median estimator is therefore affine invariant with respect to scatter transformations, overcoming certain limitations of previous approaches and enhancing

robustness in high-dimensional inference under elliptical distributions. We demonstrate the practical utility of the HR estimator through its applications to two core problems in modern high-dimensional statistics: one-sample location testing and quadratic discriminant analysis.

For the high-dimensional one-sample location testing problem, substantial research has been conducted over the past two decades, leading to three main categories of testing procedures. The first category comprises sum-type tests, which aggregate statistics across all variables and are powerful against dense alternatives (Bai & Saranadasa, 1996; Chen et al., 2010; Wang et al., 2015; Ayyala et al., 2017; Feng et al., 2015; Feng & Sun, 2016; Feng et al., 2016; 2021). The second category includes max-type tests, which focus on the maximum of individual statistics and excel under sparse alternatives, explored in works such as (Zhong et al., 2013; Cai et al., 2013; Cheng et al., 2023; Chang et al., 2017). The third category consists of adaptive type tests, which combine sum-type and max-type strategies to achieve robustness across diverse sparsity regimes, with important contributions from Xu et al. (2016); He et al. (2021); Feng et al. (2022a; 2024); Chang et al. (2023); Chen et al. (2024); Ma et al. (2024). Comprehensive overviews are available in Huang et al. (2022) and Liu et al. (2024).

Since the seminal contribution of Chernozhukov et al. (2013; 2017), Gaussian approximation has become a cornerstone of high-dimensional statistical inference. Inspired by their theoretical framework, we first derive a Bahadur representation for the standardized spatial median estimator and establish its Gaussian approximation over a class of simple convex sets. This theoretical development provides a solid foundation for analyzing the limiting distributions of our proposed test statistics and facilitates the verification of the asymptotic independence between the max-type and sum-type statistics. Specifically, we introduce two types of test statistics based on the $L_2$ and $L_\infty$ norms of the corresponding standardized spatial-median estimator, which correspond to sum-type and max-type test procedures, respectively. We rigorously establish that these statistics are asymptotically independent. Leveraging this property, we develop a Cauchy combination test that integrates both sources of information. While Liu et al. (2024) focuses on the sparsity of the original mean vector $\boldsymbol{\mu}$, our approach targets the sparsity of the transformed mean $\boldsymbol{\Sigma}^{-1/2}\boldsymbol{\mu}$, which removes correlations among coordinates. This notion of sparsity is natural in settings with strong correlations among observed variables, such as financial or gene expression data, where the underlying signal is often concentrated in a small number of latent directions. Such sparsity assumptions on the decorrelated mean vector have been widely used and formally justified in the high-dimensional classification and testing literature (Chen & Tang, 2021; Cai & Liu, 2011). Furthermore, given that the true sparsity structure (whether in $\boldsymbol{\mu}$ or in $\boldsymbol{\Sigma}^{-1/2}\boldsymbol{\mu}$) is generally unknown in practice, we further extend our procedure by combining four test statistics to achieve greater adaptability across various sparsity regimes. Simulation studies confirm that the resulting Cauchy combination tests perform well under a wide range of distributional settings and sparsity levels, highlighting their robustness and wide applicability for high-dimensional hypothesis testing.

We further apply the proposed HR estimator to improve quadratic discriminant analysis (QDA), which is a natural extension of linear discriminant analysis (LDA) (Friedman, 1989; Muirhead, 2009). When population parameters are known, QDA achieves optimal classification by comparing likelihood ratios. In low-dimensional settings, replacing population parameters with sample estimates generally preserves strong classification performance. However, in high-dimensional regimes, the singularity of the sample covariance matrix renders classical QDA infeasible. To address this, previous works have proposed sparse estimators for the covariance matrix (Wu et al., 2019; Xiong et al., 2016) or its inverse (Cai et al., 2011; Yuan & Lin, 2007). Nonetheless, these approaches fundamentally rely on the sample covariance matrix, which is highly sensitive to heavy-tailed distributions, and thus undermines robustness. To overcome this limitation, we propose a robust QDA procedure by replacing the sample mean and precision matrix with the HR-based spatial median and scatter matrix estimators. The resulting classifier retains high efficiency even under heavy-tailed distributions. We rigorously establish the asymptotic properties of the proposed method under mild moment conditions and demonstrate its superior performance through extensive simulations and real data application. These results highlight the significant gains in robustness and classification accuracy offered by our framework in high-dimensional, non-normal settings.

The remainder of this paper is structured as follows. Section 2 introduces the high-dimensional HR estimator. Section 3 develops the corresponding theoretical results and proposes a new adaptive test for the high-dimensional one-sample location problem. Section 4 presents simulation studies related to this test. Section 5 concludes the paper. Due to space constraints, additional results, including the second adaptive test for the one-sample location problem, its asymptotic theory, and simulation results, as well as the full study on high-dimensional quadratic discriminant analysis, are provided in the Appendix.

**Notations:** For $d$-dimensional $\boldsymbol{x} \in \mathbb{R}^d$, $\|\boldsymbol{x}\|$ and $\|\boldsymbol{x}\|_\infty$ denote its Euclidean norm and maximum-norm, respectively. Denote $a_n \lesssim b_n$ if there exists constant $C$, $a_n \leq C b_n$ and $a_n \asymp b_n$ if both $a_n \lesssim b_n$ and $b_n \lesssim a_n$ hold. Let $\psi_\alpha(x) = \exp(x^\alpha) - 1$ be a function defined on $[0, \infty)$ for $\alpha > 0$. Then the Orlicz norm $\|\cdot\|_{\psi_\alpha}$ of a random variable $X$ is defined as $\|X\|_{\psi_\alpha} = \inf\{t > 0, \mathbb{E}\{\psi_\alpha(|X|/t)\} \leqslant 1\}$. Let $\mathrm{tr}(\cdot)$ be a trace of matrix, $\lambda_{\min}(\cdot)$ and $\lambda_{\max}(\cdot)$ be the minimum and maximum eigenvalue for symmetric matrix. For a matrix $\mathbf{A} = (a_{ij}) \in \mathbb{R}^{p \times q}$, we define the elementwise $\ell_\infty$ norm $\|\mathbf{A}\|_\infty = \max_{1 \leq i \leq p, 1 \leq j \leq q} |a_{ij}|$, the operation norm $\|\mathbf{A}\|_{\mathrm{op}} = \sup_{\|\boldsymbol{x}\| \leq 1} \|\mathbf{A}\boldsymbol{x}\|$, the matrix $\ell_1$ norm $\|\mathbf{A}\|_{L_1} = \max_{1 \leq j \leq q} \sum_{i=1}^p |a_{ij}|$, the Frobenius norm $\|\mathbf{A}\|_F = (\sum_{i,j} a_{ij}^2)^{1/2}$, and the elementwise $\ell_1$ norm $\|\mathbf{A}\|_1 = \sum_{i=1}^p \sum_{j=1}^q |a_{ij}|$. $\mathbf{I}_p$ represents a $p$-dimensional identity matrix, $\mathrm{diag}(v_1, \ldots, v_p)$ represents the diagonal matrix with entries $\boldsymbol{v} = (v_1, \ldots, v_p)$. The notation $\mathbf{1}_d$ denotes a $d$-dimensional vector whose elements are all one. And $\mathbb{S}^{d-1}$ represents the unit sphere in $\mathbb{R}^d$. $\xrightarrow{d}$ stands for convergence in distribution. Unless stated otherwise, the notation in the supplementary material are consistent with those in the main text.

## 2 High-dimensional HR estimator

Let $\boldsymbol{X}_1, \ldots, \boldsymbol{X}_n$ be independently and identically distributed (i.i.d) observations from $p$-variate elliptical distribution with density function $|\boldsymbol{\Sigma}|^{-1/2} g\{\|\boldsymbol{\Sigma}^{-1/2}(\boldsymbol{x} - \boldsymbol{\mu})\|\}$, where $\boldsymbol{\mu}$ is the location parameter, $\boldsymbol{\Sigma}$ is a positive definite symmetric $p \times p$ scatter matrix, and $g(\cdot)$ is a scale function. The spatial sign function is defined as $U(\mathbf{x}) = \|\mathbf{x}\|^{-1} \mathbf{x} \mathbb{I}(\mathbf{x} \neq \mathbf{0})$. Denote $\boldsymbol{\varepsilon}_i = \boldsymbol{\Sigma}^{-1/2}(\boldsymbol{X}_i - \boldsymbol{\mu})$. The modulus $\|\boldsymbol{\varepsilon}_i\|$ and the direction $\boldsymbol{U}_i = U(\boldsymbol{\varepsilon}_i)$ are independent, and the direction vector $\boldsymbol{U}_i$ is uniformly distributed on $\mathbb{S}^{p-1}$. It is then well known that $\mathbb{E}(\boldsymbol{U}_i) = \mathbf{0}$ and $\mathrm{Cov}(\boldsymbol{U}_i) = p^{-1}\mathbf{I}_p$. Without loss of generality, we assume that the scatter matrix satisfies $\mathrm{tr}(\boldsymbol{\Sigma}) = p$.

The Hettmansperger-Randles (HR) (Hettmansperger & Randles, 2002) estimates for the location and scatter matrix are the values that simultaneously satisfy the following two equations:

$$\frac{1}{n} \sum_{i=1}^n U(\hat{\boldsymbol{\varepsilon}}_i) = \mathbf{0} \quad \text{and} \quad \frac{p}{n} \sum_{i=1}^n \{U(\hat{\boldsymbol{\varepsilon}}_i) U(\hat{\boldsymbol{\varepsilon}}_i)^\top\} = \mathbf{I}_p,$$

where $\hat{\boldsymbol{\varepsilon}}_i = \hat{\boldsymbol{\Sigma}}^{-1/2}(\boldsymbol{X}_i - \hat{\boldsymbol{\mu}})$. These estimators are affine equivariant and provide robust estimates of both the location parameter and the scatter matrix. Hettmansperger & Randles (2002) further established their asymptotic distributions and showed that the HR estimators possess bounded influence functions and a positive breakdown point.

The HR estimator is computed via the iterative procedure summarized in Algorithm 1, which alternately updates the residuals, location, and scatter matrix. In high-dimensional settings, however, the sample spatial-sign covariance matrix (SSCM) $\hat{\mathbf{S}} \doteq n^{-1} \sum_{i=1}^n U(\hat{\boldsymbol{\varepsilon}}_i) U(\hat{\boldsymbol{\varepsilon}}_i)^\top$ becomes singular, making Step 3 infeasible. A naive workaround is to restrict $\boldsymbol{\Sigma}$ to be diagonal (Feng et al., 2016), but this loses the full scatter structure.

---

**Algorithm 1** HR estimator

---

1: **procedure** Update($\boldsymbol{X}_1, \ldots, \boldsymbol{X}_n, \hat{\boldsymbol{\mu}}, \hat{\boldsymbol{\Sigma}}, p$)

2:     **Step 1**: $\hat{\boldsymbol{\varepsilon}}_i \leftarrow \hat{\boldsymbol{\Sigma}}^{-1/2}(\boldsymbol{X}_i - \hat{\boldsymbol{\mu}})$

3:     **Step 2**: $\hat{\boldsymbol{\mu}} \leftarrow \hat{\boldsymbol{\mu}} + \frac{\hat{\boldsymbol{\Sigma}}^{1/2} \sum_{i=1}^n U(\hat{\boldsymbol{\varepsilon}}_i)}{\sum_{i=1}^n \|\hat{\boldsymbol{\varepsilon}}_i\|^{-1}}$

4:     **Step 3**: $\hat{\boldsymbol{\Sigma}} \leftarrow p\hat{\boldsymbol{\Sigma}}^{1/2}\{n^{-1} \sum_{i=1}^n U(\hat{\boldsymbol{\varepsilon}}_i) U(\hat{\boldsymbol{\varepsilon}}_i)^\top\}\hat{\boldsymbol{\Sigma}}^{1/2}$

5:     **Step 4**: Repeat Steps 1 - 3 until convergence.

6:     **return** $\hat{\boldsymbol{\mu}}, \hat{\boldsymbol{\Sigma}}$

7: **end procedure**

---

Our key insight comes from the elliptical symmetry of the population: if the initial location and precision estimates are reasonable, the scaled SSCM $p^{-1}\hat{\mathbf{S}}$ is approximately equal to the identity matrix $\mathbf{I}_p$. This implies that most off-diagonal entries are negligible, allowing us to safely ignore them in Step 3 without imposing any structural assumptions. Therefore, we adopt the banding approach proposed by Bickel & Levina (2008b) for $\hat{\mathbf{S}}$, defining $\mathcal{B}_h(\mathbf{M}) = \{m_{ij}\mathbb{I}(|i-j| \leq h)\}$ with $0 \leq h < p$ to simplify computation while retaining the essential scatter information. The bandwidth parameter $h$ exhibits low sensitivity to the final results, the relevant explanations are located in

Appendix C. To initialize the procedure, we use the spatial median for the location parameter,

$$\hat{\boldsymbol{\mu}}_0 = \arg \min_{\boldsymbol{\mu} \in \mathbb{R}^p} \sum_{i=1}^{n} \|\boldsymbol{X}_i - \boldsymbol{\mu}\|, \tag{1}$$

which is consistent in high dimensions (Zou et al., 2014; Feng et al., 2016; Feng, 2024), and the sparse graphical Lasso (SGLASSO) for the precision matrix (Lu & Feng, 2025):

$$\hat{\boldsymbol{\Omega}}_0 = \arg \min_{\boldsymbol{\Theta} \succ \mathbf{0}} \operatorname{tr}(p\boldsymbol{\Theta}\hat{\mathbf{S}}_0) - \log\{\det(\boldsymbol{\Theta})\} + \lambda_{\mathbf{n}}\|\boldsymbol{\Theta}\|_{\mathbf{1}}, \tag{2}$$

where $\boldsymbol{\Theta} \succ \mathbf{0}$ indicates $\boldsymbol{\Theta}$ is positive define, $\hat{\mathbf{S}}_0 = n^{-1}\sum_{i=1}^{n} U(\boldsymbol{X}_i - \hat{\boldsymbol{\mu}}_0)U(\boldsymbol{X}_i - \hat{\boldsymbol{\mu}}_0)^{\top}$ is the sample spatial-sign covariance matrix based on the initial location estimate.

Combining these components, we present Algorithm 2, a high-dimensional extension of the HR estimator that robustly estimates both the location and scatter matrix.

---

**Algorithm 2** High-dimensional HR estimator

---

1: **procedure** UPDATE($\boldsymbol{X}_1, \dots, \boldsymbol{X}_n, \hat{\boldsymbol{\mu}}, \hat{\boldsymbol{\Sigma}}, p$)

2:  Initial estimator $\hat{\boldsymbol{\mu}} = \hat{\boldsymbol{\mu}}_0, \hat{\boldsymbol{\Sigma}} = \hat{\boldsymbol{\Omega}}_0^{-1}$

3:  **Step 1**: $\hat{\boldsymbol{\varepsilon}}_i \leftarrow \hat{\boldsymbol{\Sigma}}^{-1/2}(\boldsymbol{X}_i - \hat{\boldsymbol{\mu}})$

4:  **Step 2**: $\hat{\boldsymbol{\mu}} \leftarrow \hat{\boldsymbol{\mu}} + \frac{\hat{\boldsymbol{\Sigma}}^{1/2}n^{-1}\sum_{i=1}^{n}\mathbf{U}(\hat{\boldsymbol{\varepsilon}}_i)}{n^{-1}\sum_{i=1}^{n}\|\hat{\boldsymbol{\varepsilon}}_i\|^{-1}}$

5:  **Step 3**: $\hat{\boldsymbol{\Sigma}} \leftarrow p\hat{\boldsymbol{\Sigma}}^{1/2}\mathcal{B}_h\left\{n^{-1}\sum_{i=1}^{n}U(\hat{\boldsymbol{\varepsilon}}_i)U(\hat{\boldsymbol{\varepsilon}}_i)^{\top}\right\}\hat{\boldsymbol{\Sigma}}^{1/2}, \hat{\boldsymbol{\Sigma}} \leftarrow \frac{p\hat{\boldsymbol{\Sigma}}}{\operatorname{tr}(\hat{\boldsymbol{\Sigma}})}$

6:  **Step 4**: Repeat Steps 1 - 3 until convergence.

7:  **return** $\hat{\boldsymbol{\mu}}, \hat{\boldsymbol{\Sigma}}$

8: **end procedure**

---

In the next section, we will prove the consistency of the high-dimensional HR estimator $\hat{\boldsymbol{\mu}}$, and then apply it together with $\hat{\boldsymbol{\Sigma}}$ to the one-sample location testing problem.

## 3 HIGH-DIMENSIONAL ONE-SAMPLE LOCATION PROBLEM

In this section, we consider the following one-sample hypothesis testing problem:

$$H_0 : \boldsymbol{\mu} = \mathbf{0} \quad \text{versus} \quad H_1 : \boldsymbol{\mu} \neq \mathbf{0}.$$

When the dimension $p$ is fixed and the observations $\boldsymbol{X}_1, \dots, \boldsymbol{X}_n \overset{\text{i.i.d.}}{\sim} \mathcal{N}(\mathbf{0}, \boldsymbol{\Sigma}_X)$, the classical Hotelling's $T^2$ test statistic commonly used: $T^2 = n\bar{\boldsymbol{X}}^{\top}\hat{\boldsymbol{\Sigma}}_X^{-1}\bar{\boldsymbol{X}}$, where $\bar{\boldsymbol{X}}$ and $\hat{\boldsymbol{\Sigma}}_X$ represent the sample mean vector and the sample covariance matrix, respectively. However, when the dimension $p$ exceeds the sample size $n$, the sample covariance matrix $\hat{\boldsymbol{\Sigma}}_X$ becomes singular, rendering Hotellings $T^2$ test inapplicable.

To overcome the limitation, Fan et al. (2015) proposed replacing the sample covariance matrix with a sparse estimator $\hat{\boldsymbol{\Sigma}}_{\tau}$ and introduced the following test statistic:

$$T_{FLY} = \frac{n\bar{\boldsymbol{X}}^{\top}\hat{\boldsymbol{\Sigma}}_{\tau}^{-1}\bar{\boldsymbol{X}} - p}{\sqrt{2p}}.$$

Under the null, they showed that as $(n, p) \to \infty$, $T_{FLY} \overset{d}{\to} \mathcal{N}(0, 1)$. As a sum-type test, $T_{FLY}$ is effective under dense alternatives but deteriorates in performance under sparse ones. To better handle sparse alternatives, Chen et al. (2024) introduced a max-type test statistic:

$$T_{CFL} = \max_{1 \le i \le p} W_i^2 - 2\log p + \log\log p,$$

where $\boldsymbol{W} = (W_1, \cdots, W_p)^\top = n^{1/2}\hat{\boldsymbol{\Sigma}}_\tau^{-1/2}\bar{\boldsymbol{X}}$. They show that under the null, $T_{CFL}$ follows a Gumbel distribution.

Both $T_{FLY}$ and $T_{CFL}$ rely on multivariate normality or an independent component model, which limits their robustness under heavy-tailed distributions such as multivariate $t$ or multivariate mixture normal. This motivates the need for test procedures that remain effective when the data deviate from normality.

For elliptical distributions, spatial-sign methods provide a natural robust alternative and have been extensively studied (Oja, 2010). When the dimension $p$ is fixed, the spatial-sign test with inner standardization (Randles, 2000) is defined as $Q^2 = np\bar{\boldsymbol{U}}_T^\top\bar{\boldsymbol{U}}_T$, $\bar{\boldsymbol{U}}_T = n^{-1}\sum_{i=1}^n \hat{\boldsymbol{U}}_{i,T}$, $\hat{\boldsymbol{U}}_{i,T} = U(\hat{\boldsymbol{\Sigma}}_T^{-1/2}\boldsymbol{X}_i)$. where $\hat{\boldsymbol{\Sigma}}_T$ denotes Tyler's scatter matrix (Tyler, 1987). This construction standardizes the data in the spatial-sign framework, providing a test that is affine-invariant and resistant to heavy tails.

However, in high-dimensional settings where $p > n$, Tylers scatter matrix is no longer well-defined, making $Q^2$ inapplicable. To overcome this limitation, we propose novel test procedures based on high-dimensional HR estimators, aiming to maintain robustness and efficiency under heavy-tailed distributions while adapting to the challenges of high dimensionality.

First, we investigate some theoretical properties of the high dimensional HR estimator $\hat{\boldsymbol{\mu}}$. Let $\boldsymbol{U}_i = U(\boldsymbol{\varepsilon}_i)$, $r_i = \|\boldsymbol{\varepsilon}_i\|$, $\boldsymbol{\Omega} = \boldsymbol{\Sigma}^{-1}$, $\mathbf{S} = \mathbb{E}\{U(\boldsymbol{X}_i - \boldsymbol{\mu})U(\boldsymbol{X}_i - \boldsymbol{\mu})^\top\}$ and $\zeta_k = \mathbb{E}(r_i^{-k})$ for $i = 1, \ldots, n$.

**Assumption 1.** *There exist constants $\underline{b}$, $\bar{B} > 0$ such that $\underline{b} \leq \limsup_p \mathbb{E}\{(r_1/\sqrt{p})^{-k}\} \leq \bar{B}$ for $k \in \{-1, 1, 2, 3, 4\}$. And $\zeta_1^{-1}r_1^{-1}$ is sub-Gaussian distributed, i.e. $\|\zeta_1^{-1}r_1^{-1}\|_{\psi_2} \leq K_1 < \infty$.*

**Assumption 2.** *$\exists \eta, h > 0$, s.t. $\eta < \lambda_{\min}(\boldsymbol{\Sigma}) \leq \lambda_{\max}(\boldsymbol{\Sigma}) < \eta^{-1}$, $\mathrm{tr}(\boldsymbol{\Sigma}) = p$ and $\|\boldsymbol{\Sigma}\|_{L_1} \leq h$. The diagonal matrix of $\boldsymbol{\Sigma}$ is denoted as $\mathbf{D} = \mathrm{diag}\{d_1^2, d_2^2, \ldots, d_p^2\}$, $\liminf_{p\to\infty}\min_{j=1,\ldots,p} d_j > \underline{d}$ for some constant $\underline{d} > 0$ and $\limsup_{p\to\infty}\max_{j=1,\ldots,p} d_j < \bar{D}$ for some constant $\bar{D} > 0$.*

**Assumption 3.** *$\exists T > 0$, $0 \leq q < 1$, $s_0(p) > 0$, s.t. (1) $\|\boldsymbol{\Omega}\|_{L_1} \leq T$, (2) $\max_{1\leq i\leq p}\sum_{j=1}^p |\omega_{ij}|^q \leq s_0(p)$.*

**Assumption 4.** *$\limsup_p \|\mathbf{S}\|_{op} < 1 - \psi < 1$ for some positive constant $\psi$.*

Assumption 1 aligns with Assumption 1-2 in Liu et al. (2024), which requires that $\zeta_k \asymp p^{-k/2}$. Assumptions 2 and 3 are standard conditions in high-dimensional data analysis, as seen in Bickel & Levina (2008b) and Cai et al. (2011), ensuring the sparsity of the covariance and precision matrices. Assumption 4 corresponds to Assumption (A2) in Feng (2024), guaranteeing the consistency of the initial sample spatial median.

The following lemma provides a Bahadur representation of the standardized estimator $\hat{\boldsymbol{\mu}}$, which lays the foundation for the Gaussian approximation in Lemma 2.

**Lemma 1.** *(Bahadur representation) Under the Assumptions 1–4 and $\log p = o(n^{1/3})$, there exist constants $C_{\eta,T}$ and $C$, such that if we pick $\lambda_n = T\{\sqrt{2}C(8 + \eta^2 C_{\eta,T})\eta^{-2}n^{-1/2}\log^{1/2} p + p^{-1/2}C_{\eta,T}\}$, and $\lambda_n^{1-q}s_0(p)\log^{1/2} p = o(1)$, then*

$$n^{1/2}\hat{\boldsymbol{\Omega}}^{1/2}(\hat{\boldsymbol{\mu}} - \boldsymbol{\mu}) = n^{-1/2}\zeta_1^{-1}\sum_{i=1}^n \boldsymbol{U}_i + C_n,$$

*where*

$$\begin{aligned}\|C_n\|_\infty = &O_p\{n^{-1/4}\log^{1/2}(np) + n^{-(1-q)/2}(\log p)^{(1-q)/2}\log^{1/2}(np)s_0(p) \\ &+ p^{-(1-q)/2}\log^{1/2}(np)s_0(p)\}.\end{aligned}$$

Let $\mathcal{A}^{\mathrm{si}}$ be the class of simple convex sets (Chernozhukov et al., 2017) in $\mathbb{R}^p$. Based on the Bahadur representation of $\hat{\boldsymbol{\mu}}$, we establish the following Gaussian approximation for $\hat{\boldsymbol{\Omega}}^{1/2}(\hat{\boldsymbol{\mu}} - \boldsymbol{\mu})$ over the class $\mathcal{A}^{\mathrm{si}}$, where $\hat{\boldsymbol{\Omega}} = \hat{\boldsymbol{\Sigma}}^{-1}$.

**Lemma 2.** *(Gaussian approximation) Assume the Assumptions 1–4 holds. If $\log p = o(n^{1/5})$, then*

$$\rho_n(\mathcal{A}^{\mathrm{si}}) = \sup_{A\in\mathcal{A}^{\mathrm{si}}}\left|\mathbb{P}\{n^{1/2}\hat{\boldsymbol{\Omega}}^{1/2}(\hat{\boldsymbol{\mu}} - \boldsymbol{\mu}) \in A\} - \mathbb{P}(\boldsymbol{Z} \in A)\right| \to 0,$$

*as $n \to \infty$, where $\boldsymbol{Z} \sim \mathcal{N}(0, p^{-1}\zeta_1^{-2}\mathbf{I}_p)$.*

Consequently, we derive the following corollary, which establishes the limiting distributions of the $L_2$- and $L_\infty$-norms of $n^{1/2}\hat{\boldsymbol{\Omega}}^{1/2}(\hat{\boldsymbol{\mu}} - \boldsymbol{\mu})$.

**Corollary 1.** *Assume the conditions of Lemma 2 hold. Set $A$ to $\{\boldsymbol{x}|\|\boldsymbol{x}\|_\infty \leq t\}$, $\{\boldsymbol{x}|\|\boldsymbol{x}\| \leq t\}$ and $\{\boldsymbol{x}|\|\boldsymbol{x}\|_\infty \leq t_1, \|\boldsymbol{x}\| \leq t_2\}$ we have*

$$\tilde{\rho}_{n,\infty} = \sup_{t\in\mathbb{R}}\left|\mathbb{P}\{n^{1/2}\|\hat{\boldsymbol{\Omega}}^{1/2}(\hat{\boldsymbol{\mu}} - \boldsymbol{\mu})\|_\infty \leqslant t\} - \mathbb{P}(\|\boldsymbol{Z}\|_\infty \leqslant t)\right| \to 0,$$

$$\tilde{\rho}_{n,2} = \sup_{t\in\mathbb{R}}\left|\mathbb{P}\{n^{1/2}\|\hat{\boldsymbol{\Omega}}^{1/2}(\hat{\boldsymbol{\mu}} - \boldsymbol{\mu})\| \leqslant t\} - \mathbb{P}(\|\boldsymbol{Z}\| \leqslant t)\right| \to 0,$$

$$\tilde{\rho}_{n,comb} = \sup_{t_1,t_2\in\mathbb{R}}\left|\mathbb{P}\{n^{1/2}\|\hat{\boldsymbol{\Omega}}^{1/2}(\hat{\boldsymbol{\mu}} - \boldsymbol{\mu})\|_\infty \leqslant t_1, n^{1/2}\|\hat{\boldsymbol{\Omega}}^{1/2}(\hat{\boldsymbol{\mu}} - \boldsymbol{\mu})\| \leqslant t_2\}\right.$$
$$\left. - \mathbb{P}(\|\boldsymbol{Z}\|_\infty \leqslant t_1, \|\boldsymbol{Z}\| \leqslant t_2)\right| \to 0,$$

*as $n \to \infty$, where $\boldsymbol{Z} \sim \mathcal{N}\left(0, \zeta_1^{-2}p^{-1}\mathbf{I}_p\right)$.*

We know that $\{\boldsymbol{x}|\|\boldsymbol{x}\|_\infty \leq t\}$ and $\{\boldsymbol{x}|\|\boldsymbol{x}\| \leq t\}$ are simple convex sets. The third equation holds because the intersection of a finite number of simple convex sets is still simply convex.

From Cai et al. (2013), we can see that $p\zeta_1^2 \max_{1\leq i\leq p} Z_i^2 - 2\log p + \log\log p$ converges to a Gumbel distribution with the cumulative distribution function (cdf) $F(x) = \exp(-\frac{1}{\sqrt{\pi}}e^{-x/2})$ as $p \to \infty$. Combining this with Corollary 1, we obtain

$$\mathbb{P}\left\{n\|\hat{\boldsymbol{\Omega}}^{1/2}(\hat{\boldsymbol{\mu}} - \boldsymbol{\mu})\|_\infty^2 p\zeta_1^2 - 2\log p + \log\log p \leq x\right\} \to \exp\left(-e^{-x/2}/\sqrt{\pi}\right). \tag{3}$$

We estimate $\zeta_1$ by $\hat{\zeta}_1 := n^{-1}\sum_{i=1}^n \tilde{r}_i^{-1}$, where $\tilde{r}_i = \|\hat{\boldsymbol{\Omega}}^{1/2}(\mathbf{X}_i - \hat{\boldsymbol{\mu}})\|$ and establish its consistency in Lemma 6. We then propose the following max-type test statistic:

$$T_{MAX} = n\left\|\hat{\boldsymbol{\Omega}}^{1/2}\hat{\boldsymbol{\mu}}\right\|_\infty^2 \hat{\zeta}_1^2 p - 2\log p + \log\log p.$$

It is evident that $T_{MAX}$ is affine invariant.

**Theorem 1.** *Suppose the Assumptions 1-4 hold. Under the null hypothesis, as $(n, p) \to \infty$, we have*

$$\mathbb{P}(T_{MAX} \leq x) \to \exp\left(-e^{-x/2}/\sqrt{\pi}\right).$$

According to Theorem 1, $H_0$ will be rejected when our proposed statistic $T_{MAX}$ is larger than the $(1 - \alpha)$ quantile $q_{1-\alpha} = -\log\pi - 2\log\log(1-\alpha)^{-1}$ of the Gumbel distribution $F(x)$. We next establish the consistency of the test in the following theorem.

**Theorem 2.** *Suppose the conditions assumed in Theorem 1 hold, for any given $\alpha \in (0,1)$, if $\|\boldsymbol{\Omega}^{1/2}\boldsymbol{\mu}\|_\infty \geq \widetilde{C}n^{-1/2}(\log p + q_{1-\alpha})^{1/2}$, for some large enough constant $\widetilde{C}$, then*

$$\mathbb{P}(T_{MAX} > q_{1-\alpha}|H_1) \to 1, \ \ as \ (n, p) \to \infty.$$

Next, we consider a special case of alternative hypothesis:

$$H_1 : \boldsymbol{\Omega}^{1/2}\boldsymbol{\mu} = (\mu_1, 0, \cdots, 0)^\top, \mu_1 > 0, \tag{4}$$

which means there are only one variable with nonzero mean. Similar to the calculation in Liu et al. (2024), we can easily show the power function of new proposed $T_{MAX}$ test is

$$\beta_{MAX}(\boldsymbol{\mu}) \in \left(\Phi\{-x_\alpha^{1/2} + (np)^{1/2}d_1^{-1}\mu_1\zeta_1\}, \Phi\{-x_\alpha^{1/2} + (np)^{1/2}d_1^{-1}\mu_1\zeta_1\} + \alpha\right),$$

where $x_\alpha = 2\log p - \log\log p + q_{1-\alpha}$. Similarly, the power function of Chen et al. (2024)'s test is

$$\beta_{CFL}(\boldsymbol{\theta}) \in \left(\Phi(-x_\alpha^{1/2} + n^{1/2}\varsigma_1^{-1}\mu_1), \Phi(-x_\alpha^{1/2} + n^{1/2}\varsigma_1^{-1}\mu_1) + \alpha\right),$$

where $\varsigma_i^2$ is the variance of $X_{ki}, i = 1, \cdots, p$. Thus, the asymptotic relative efficiency of $T_{MAX}$ with respective to Cai et al. (2013)'s test could be approximated as

$$\mathrm{ARE}\left(T_{MAX}, T_{CFL}\right) = \left\{\mathbb{E}\left(r_i^{-1}\right)\right\}^2 \mathbb{E}\left(r_i^2\right) \geq 1,$$

which indicates the superior performance of spatial sign-based methods over least-square-based methods. This observation is well-documented in the literature, including Feng & Sun (2016), Feng et al. (2016), and Liu et al. (2024). If $\boldsymbol{X}_i$ are generated from standard multivariate $t$-distribution with $\nu$ degrees of freedom ($\nu > 2$),

$$\text{ARE}\left(T_{MAX}, T_{CFL}\right) = \frac{2}{\nu - 2} \left[\frac{\Gamma\{(\nu+1)/2\}}{\Gamma(\nu/2)}\right]^2.$$

For different $\nu = 3, 4, 5, 6$, the above ARE are $2.54, 1.76, 1.51, 1.38$, respectively. Under the multivariate normal distribution ($\nu = \infty$), our $T_{MAX}$ test is the same powerful as Chen et al. (2024)'s test. However, our $T_{MAX}$ test is much more powerful under the heavy-tailed distributions.

Similarly, we can see that $(2p)^{-1/2}(\sum_{i=1}^{p} p\zeta_1^2 Z_i^2 - p)$ converges to a standard Gaussian distribution with cdf $\Phi(x)$. In combining with the Corollary 1 we can conclude that,

$$\mathbb{P}\left[(2p)^{-1/2}\left\{n\|\hat{\boldsymbol{\Omega}}^{1/2}(\hat{\boldsymbol{\mu}} - \boldsymbol{\mu})\|^2 p\zeta_1^2 - p\right\} \le x\right] \to \Phi(x). \tag{5}$$

Then we propose the sum-type test statistic

$$T_{SUM} = \frac{\sqrt{2p}}{2}\left(n\hat{\zeta}_1^2 \hat{\boldsymbol{\mu}}^\top \hat{\boldsymbol{\Omega}} \hat{\boldsymbol{\mu}} - 1\right). \tag{6}$$

**Theorem 3.** *Suppose the Assumptions 1-4 hold. Under $H_0 : \boldsymbol{\mu} = \boldsymbol{0}$, as $(n, p) \to \infty$, we have $T_{SUM} \xrightarrow{d} \mathcal{N}(0, 1)$. Furthermore, under $H_1 : \boldsymbol{\mu}^\top \boldsymbol{\Omega}\boldsymbol{\mu} = o(pn^{-1})$, as $(n, p) \to \infty$, we have $T_{SUM} - 2^{-1/2}np^{1/2}\zeta_1^2 \boldsymbol{\mu}^\top \boldsymbol{\Omega}\boldsymbol{\mu} \xrightarrow{d} \mathcal{N}(0, 1)$.*

By Theorem 3, the asymptotic power function of $T_{SUM}$ is

$$\beta_{SUM}(\boldsymbol{\mu}) = \Phi\left(-z_{1-\alpha} + 2^{-1/2}np^{1/2}\zeta_1^2 \boldsymbol{\mu}^\top \boldsymbol{\Omega}\boldsymbol{\mu}\right).$$

After some simply calculations, we can obtain the power function of $T_{FLY}$ is

$$\beta_{FLY}(\boldsymbol{\mu}) = \Phi\left(-z_{1-\alpha} + 2^{-1/2}p^{-1/2}n\boldsymbol{\mu}^\top \boldsymbol{\Sigma}_s^{-1}\boldsymbol{\mu}\right).$$

where $\boldsymbol{\Sigma}_s = \mathbb{E}(\boldsymbol{X}_i \boldsymbol{X}_i^\top)$ is the covariance matrix and $\boldsymbol{\Sigma}_s = p^{-1}\mathbb{E}(r_i^2)\boldsymbol{\Sigma}$. So the asymptotic relative efficiency (ARE) of $T_{SUM}$ with repective to $T_{FLY}$ is $\text{ARE}(T_{SUM}, T_{FLY}) = \{\mathbb{E}(r_i^{-1})\}^2 \mathbb{E}(r_i^2) \ge 1$, which is the same as $\text{ARE}(T_{MAX}, T_{CFL})$.

However, when the dimension gets larger, there would be a non-negligible bias term in $T_{SUM}$ and $T_{MAX}$. To use the above sum-type and max-type test procedure, we adopt the bootstrap method to calculate the bias term. We simply generate $n$ samples $\boldsymbol{z}_1, \cdots, \boldsymbol{z}_n$ from the multivariate normal distribution $\mathcal{N}(\boldsymbol{0}, \hat{\boldsymbol{\Omega}}^{-1})$. Then, based on the random sample $\boldsymbol{z}_1, \cdots, \boldsymbol{z}_n$, we calculate the sum-type test statistic $T_{SUM}^*$ and max-type test statistic $T_{MAX}^*$. Repeat this procedure $M$ times, we could get a bootstrap sample of $T_{SUM}$ and $T_{MAX}$. Then, we calculate the sample mean and the sample variance of these bootstrap samples, denoted as $\mu_S^*$ and $\sigma_S^{2*}$ for $T_{SUM}^*$ and $\mu_M^*$ and $\sigma_M^{2*}$ for $T_{MAX}^*$. The corresponding $p$-values of $T_{SUM}$ and $T_{MAX}$ are

$$p_{SUM} = 1 - \Phi\{(T_{SUM} - \mu_S^*)/\sigma_S^*\}, p_{MAX} = 1 - F\left\{\sigma_0(T_{MAX} - \mu_M^*)/\sigma_M^* + \mu_0\right\},$$

where $\mu_0 = -\log(\pi) + 2\gamma$ and $\sigma_0^2 = 3^{-1}2\pi^2$ are the expectation and variance of the Gumbel distribution $F(x)$. Here $\gamma$ is the Euler constant. Because we only need the mean and variance of the bootstrap samples, so the bootstrap size $M = 50$ is always enough for controlling the empirical sizes.

It is well known that sum-type and max-type tests are powerful against dense and sparse alternatives, respectively. To accommodate unknown sparsity in the real world, we adopt the Cauchy combination test Liu & Xie (2020) to integrate their advantages, leveraging their asymptotic independence.

**Theorem 4.** *Under Assumptions 1-4, if $\|\boldsymbol{\mu}\|_\infty = o(n^{-1/2})$ and $\|\boldsymbol{\mu}\| = o(p^{1/4}n^{-1/2})$, as $n, p \to \infty$, $T_{MAX}$ and $T_{SUM}$ are asymptotic independent.*

Based on Theorem 4, we define the Cauchy combination test as follows:

$$T_{CC1} = 1 - G\left[0.5\tan\left\{(0.5 - p_{MAX})\pi\right\} + 0.5\tan\left\{(0.5 - p_{SUM})\pi\right\}\right],$$

where $G(\cdot)$ is the cdf of the standard Cauchy distribution. We reject $H_0$ if $T_{CC1} < \alpha$ for a given significance level $\alpha \in (0, 1)$.

## 4 SIMULATION

We consider the following three elliptical distributions:

(i) Multivariate normal distribution: $\boldsymbol{X}_i \sim \mathcal{N}(\boldsymbol{\mu}, \boldsymbol{\Sigma})$;

(ii) Multivariate $t$-distribution: $\boldsymbol{X}_i \sim t(\boldsymbol{\mu}, \boldsymbol{\Sigma}, 3)/\sqrt{3}$;

(iii) Multivariate mixture normal distribution: $\boldsymbol{X}_i \sim \mathcal{MN}(\boldsymbol{\mu}, \boldsymbol{\Sigma}, 10, 0.8)/\sqrt{22.8}$.

Four covariance matrices are considered. Model I: $\boldsymbol{\Sigma} = (0.6^{|i-j|})_{1 \leq i,j \leq p}$; Model II: $\boldsymbol{\Sigma} = 0.5\mathbf{I}_p + 0.5\mathbf{1}\mathbf{1}^\top$; Model III: $\boldsymbol{\Omega} = (0.6^{|i-j|})_{1 \leq i,j \leq p}$, $\boldsymbol{\Sigma} = \boldsymbol{\Omega}^{-1}$. Model IV: $\boldsymbol{\Omega} = (\omega_{i,j})_{p \times p}$ where $\omega_{i,i} = 2$ for $i = 1, \ldots, p, \omega_{i,i+1} = 0.8$ for $i = 1, \ldots, p-1, \omega_{i,i+2} = 0.4$ for $i = 1, \ldots, p-2, \omega_{i,i+3} = 0.4$ for $i = 1, \ldots, p-3, \omega_{i,i+4} = 0.2$ for $i = 1, \ldots, p-4, \omega_{i,j} = \omega_{j,i}$ for $i,j = 1, \ldots, p$ and $\omega_{i,j} = 0$ otherwise. As the performance of our method is not sensitive to bandwidth choice, we set $h = 3$ throughout the paper for simplicity.

Table 1 reports the empirical sizes of the new proposed test procedures $T_{SUM}$, $T_{MAX}$ and $T_{CC1}$ with $n = 100$, $p = 120, 240$. We found that all the tests could control the empirical sizes very well. Next, we conduct a comparison between our proposed methods and several test procedures based on the sample covariance matrix. Specifically, Chen et al. (2024) proposed a max-type test, denoted by $T_{CFL}$, based on the sample mean and a sparse precision matrix estimator. Fan et al. (2015) introduced a sum-type test, $T_{FLY}$, which uses a sparse covariance matrix estimator. For a fair comparison, both $T_{CFL}$ and $T_{FLY}$ adopt the graphical lasso to estimate the corresponding matrices. Furthermore, we consider a Cauchy combination of the two, denoted by $T_{CCF}$. In particular, $T_{FLY}$ suffers from size distortion under heavy-tailed distributions when using its asymptotic critical value. To address this and ensure fair comparison, we employ a size-corrected power comparison framework, where empirical critical values are computed under the null for all tests, guaranteeing matching empirical sizes.

We focus on Model II with $n = 100$ and $p = 120$. The power of each test procedure is evaluated under various distributions. For the alternative hypothesis, we specify $\boldsymbol{\mu} = \kappa\sqrt{\log p/(ns)}\boldsymbol{\Sigma}^{1/2}(\mathbf{1}_s^\top, \mathbf{0}_{p-s}^\top)^\top$ to guarantee $\boldsymbol{\Omega}^{1/2}\boldsymbol{\mu} = \kappa\sqrt{\log p/(ns)}(\mathbf{1}_s^\top, \mathbf{0}_{p-s}^\top)^\top$, where $s$ represents the sparsity parameter of the alternative hypothesis. Specifically, for the normal distribution, we set $\kappa = 2$, for the multivariate $t$-distribution with 3 degrees of freedom, $\kappa = 1.5$, and for the multivariate mixture normal distribution, $\kappa = 0.6$.

Figure 1 shows the power curves for each test across various scenarios. Under normal distribution, $T_{SUM}$ and $T_{MAX}$ perform similarly to $T_{FLY}$ and $T_{CFL}$, respectively. However, for non-normal distributions, our robust methods $T_{SUM}$, $T_{MAX}$, and $T_{CC1}$ significantly outperform $T_{FLY}$, $T_{CFL}$, and $T_{CCF}$, demonstrating their robustness in heavy-tailed settings. When the sparsity parameter $s$ is small, max-type tests ($T_{MAX}$, $T_{CFL}$) exhibit higher power than sum-type tests ($T_{SUM}$, $T_{FLY}$). In contrast, for dense alternatives ($s$ large), sum-type tests outperform max-type ones. The Cauchy combination tests, $T_{CC1}$ and $T_{CCF}$, consistently perform well across different sparsity levels. In conclusion, $T_{CC1}$ demonstrates superior performance under both heavy-tailed distributions and varying sparsity levels, exhibiting double robustness.

Table 1: Empirical sizes (%) of the three proposed test procedures under different models with $n = 100$.

| Dist. | Test | Model I | | Model II | | Model III | | Model IV | |
|---|---|---|---|---|---|---|---|---|---|
| | | $p = 120$ | $p = 240$ | $p = 120$ | $p = 240$ | $p = 120$ | $p = 240$ | $p = 120$ | $p = 240$ |
| (i) | $T_{SUM}$ | 4.3 | 4.9 | 4.2 | 5.7 | 4.4 | 5.5 | 4.7 | 5.2 |
| | $T_{MAX}$ | 5.2 | 4.8 | 5.1 | 5.9 | 4.1 | 4.6 | 4.3 | 5.9 |
| | $T_{CC1}$ | 4.7 | 5.3 | 5.6 | 4.5 | 4.9 | 5.4 | 4.5 | 5.5 |
| (ii) | $T_{SUM}$ | 4.5 | 4.1 | 5.8 | 4.6 | 5.1 | 5.3 | 5.3 | 4.8 |
| | $T_{MAX}$ | 4.8 | 4.2 | 5.7 | 5.0 | 4.9 | 4.4 | 4.3 | 5.6 |
| | $T_{CC1}$ | 5.5 | 4.7 | 4.3 | 5.6 | 4.0 | 5.2 | 4.8 | 5.2 |
| (iii) | $T_{SUM}$ | 4.2 | 5.6 | 4.9 | 4.4 | 5.8 | 4.3 | 4.1 | 5.7 |
| | $T_{MAX}$ | 5.2 | 4.7 | 5.5 | 4.1 | 5.0 | 4.6 | 5.1 | 4.7 |
| | $T_{CC1}$ | 4.8 | 5.3 | 4.5 | 5.7 | 4.0 | 5.4 | 4.4 | 5.8 |

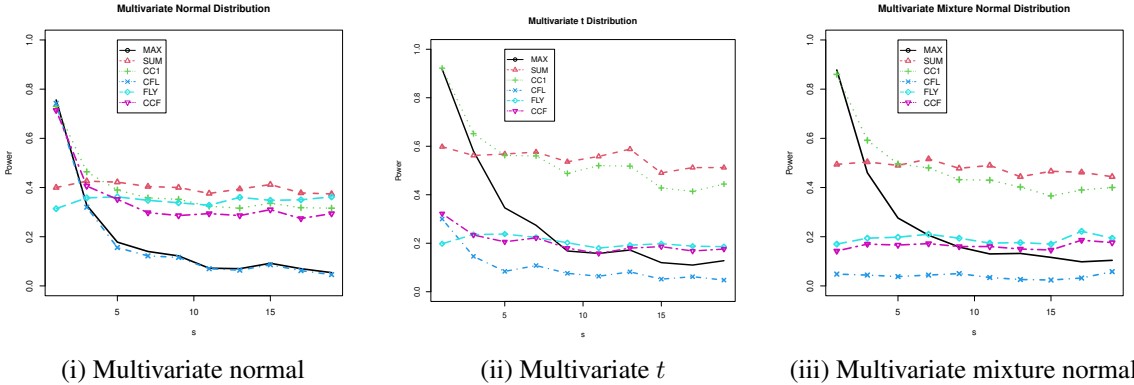

(i) Multivariate normal      (ii) Multivariate $t$      (iii) Multivariate mixture normal

Figure 1: Power curves of each method with different sparsity under Model II and $n = 100$, $p = 120$.

## 5 CONCLUSION

In this paper, we proposed a high-dimensional extension of the Hettmansperger-Randles estimator and applied it to two problems in high-dimensional statistics: the one-sample location testing problem and quadratic discriminant analysis. Simulation studies and theoretical analysis confirm the superior efficiency and robustness of our estimator in high-dimensional settings.

In particular, it may be fruitfully applied to other important problems such as the two-sample location test (Feng et al., 2016) and the high-dimensional linear asset pricing model (Feng et al., 2022b). These potential extensions warrant further investigation in future research. In addition, our methods rely on the assumption of an elliptically symmetric distribution, which may limit their applicability in more general settings. Existing work has shown that under near-spherical directional distributions and finite-moment conditions (Cheng et al., 2023; Liu et al., 2024), Gaussian approximation theory can be established, with Liu et al. (2024) further demonstrating the asymptotic independence between sum-type and max-type test statistics. An important direction for future research is to investigate how to maintain algorithmic implementability while establishing Gaussian approximation on simple convex sets and the asymptotic independence of test statistics under such general models.

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

# Supplemental Material of "High-Dimensional Hettmansperger-Randles Estimator and Its Applications"

## A  QUADRATIC DISCRIMINANT ANALYSIS

### A.1  METHOD

Consider the problem of classifying a $p$-dimensional normally distributed vector $\boldsymbol{x}$ into one of two classes represented by two $p$-dimensional normal distributions, $N_p(\boldsymbol{\mu}_1, \boldsymbol{\Xi}_1)$ and $N_p(\boldsymbol{\mu}_2, \boldsymbol{\Xi}_2)$, where $\boldsymbol{\mu}_i$'s are mean vectors and $\boldsymbol{\Xi}_i$'s are positive definite covariance matrices. If $\boldsymbol{\mu}_i$ and $\boldsymbol{\Xi}_i$, $i = 1, 2$, are known, then an optimal classification rule having the smallest possible misclassification rate can be constructed. However, $\boldsymbol{\mu}_i$ and $\boldsymbol{\Xi}_i$, $i = 1, 2$, are usually unknown and the optimal classification rule, the Bayes rule, classifies $\boldsymbol{x}$ to class 2 if and only if

$$(\boldsymbol{x} - \boldsymbol{\mu}_1)^\top (\boldsymbol{\Xi}_2^{-1} - \boldsymbol{\Xi}_1^{-1})(\boldsymbol{x} - \boldsymbol{\mu}_1) - 2\boldsymbol{\delta}^\top \boldsymbol{\Xi}_2^{-1}(\boldsymbol{x} - \boldsymbol{\mu}_1) + \boldsymbol{\delta}^\top \boldsymbol{\Xi}_2^{-1}\boldsymbol{\delta} - \log(|\boldsymbol{\Xi}_1|/|\boldsymbol{\Xi}_2|) < 0, \tag{7}$$

where $\boldsymbol{\delta} = \boldsymbol{\mu}_2 - \boldsymbol{\mu}_1$. In practical applications, when the dimension is lower than the sample size, we substitute the mean and covariance matrix in (7) with their respective sample mean and covariance matrix. Nevertheless, when the dimension exceeds the sample size, the sample covariance matrix becomes non - invertible. As a result, a common approach, as described in Li & Shao (2015) and Wu et al. (2019), involves replacing the sample covariance matrix with various sparse covariance matrix estimators (Bickel & Levina, 2008a;b). However, it should be noted that these methods relying on the sample covariance matrix may not be highly efficient when the underlying distribution diverges from the normal distribution.

In fact it has been shown by Bose et al. (2015) that, for the class of elliptically symmetric distributions with the probability density function having the form

$$f(\boldsymbol{x}; \boldsymbol{\mu}, \boldsymbol{\Xi}) = |\boldsymbol{\Xi}|^{-1/2} g\left\{(\boldsymbol{x} - \boldsymbol{\mu})^\top \boldsymbol{\Xi}^{-1}(\boldsymbol{x} - \boldsymbol{\mu})\right\},$$

the Bayes rule leads to the partition

$$R_1 = \left\{\boldsymbol{x} : \frac{1}{2}\log\left(\frac{|\boldsymbol{\Xi}_2|}{|\boldsymbol{\Xi}_1|}\right) + k\Delta_d^2 \geq 0\right\},$$

where $\Delta_d^2(\boldsymbol{x}) = \left\{(\boldsymbol{x} - \boldsymbol{\mu}_2)^\top \boldsymbol{\Xi}_2^{-1}(\boldsymbol{x} - \boldsymbol{\mu}_2) - (\boldsymbol{x} - \boldsymbol{\mu}_1)^\top \boldsymbol{\Xi}_1^{-1}(\boldsymbol{x} - \boldsymbol{\mu}_1)\right\}$ and $k$ may depend on $\boldsymbol{x}$. Therefore, letting $\varsigma_p \doteq \log(|\boldsymbol{\Xi}_1|/|\boldsymbol{\Xi}_2|)$, a general classification rule (or classifier) proposed by Bose et al. (2015), is given by

$$\begin{aligned} \boldsymbol{x} \in R_1 \quad &\text{if } \Delta_d^2(\boldsymbol{x}) \geq c\varsigma_p, \\ \boldsymbol{x} \in R_2 \quad &\text{otherwise}, \end{aligned} \tag{8}$$

for some constant $c \geq 0$. Clearly, this classifier boils down to the minimum Mahalanobis distance (MMD) and the QDA classifiers whenever $c$ is chosen to be 0 and 1, respectively. It has a misclassification rate of

$$R_{QDA} = \frac{R_{QDA}^1 + R_{QDA}^2}{2}, \qquad R_{QDA}^m = \mathbb{P}(\text{incorrectly classify } x \text{ to class } m).$$

In practice, the parameters in the classifier (8) are unknown and need to be estimated from the training set. Suppose we observe two independent samples $\{\boldsymbol{X}_{il}\}_{l=1}^{n_i}, i = 1, 2$ from $f(\boldsymbol{x}; \boldsymbol{\mu}_i, \boldsymbol{\Xi}_i)$, respectively. Under the elliptical symmetric distribution assumption, we have $\boldsymbol{\Xi} = p^{-1}\operatorname{tr}(\boldsymbol{\Xi})\boldsymbol{\Sigma}$. So the inverse of the covariance matrix $\hat{\boldsymbol{\Xi}}^{-1} = p\operatorname{tr}^{-1}(\boldsymbol{\Xi})\boldsymbol{\Omega}$ could be estimated by $\tilde{\boldsymbol{\Omega}}_i = p\{\widehat{\operatorname{tr}(\boldsymbol{\Xi}_i)}\}^{-1}\hat{\boldsymbol{\Omega}}_i$ with

$$\widehat{\operatorname{tr}(\boldsymbol{\Xi}_i)} = \frac{1}{n_i - 1}\sum_{l=1}^{n_i} \boldsymbol{X}_{il}^\top \boldsymbol{X}_{il} - \frac{n_i}{n_i - 1}\bar{\boldsymbol{X}}_i^\top \bar{\boldsymbol{X}}_i,$$

and $\bar{\boldsymbol{X}}_i = n_i^{-1}\sum_{l=1}^{n_i} \boldsymbol{X}_{il}$. Then, we replace the parameters with its high dimensional HR estimators, i.e.

$$\hat{\Delta}_d^2(\boldsymbol{x}) = (\boldsymbol{x} - \hat{\boldsymbol{\mu}}_2)^\top \tilde{\boldsymbol{\Omega}}_2(\boldsymbol{x} - \hat{\boldsymbol{\mu}}_2) - (\boldsymbol{x} - \hat{\boldsymbol{\mu}}_1)^\top \tilde{\boldsymbol{\Omega}}_1(\boldsymbol{x} - \hat{\boldsymbol{\mu}}_1), \hat{\varsigma}_p = \log\left(|\tilde{\boldsymbol{\Omega}}_2|/|\tilde{\boldsymbol{\Omega}}_1|\right),$$

where the parameter $c$ is estimated the same as Subsection 2.1 in Bose et al. (2015), denoted as $\hat{c}$. So the final classification rule is

$$\begin{aligned} \boldsymbol{x} \in R_1 & \quad \text{if } \hat{\Delta}_d^2(\boldsymbol{x}) \geq \hat{c}\hat{\varsigma}_p, \\ \boldsymbol{x} \in R_2 & \quad \text{otherwise .} \end{aligned} \quad (9)$$

It has a misclassification rate of

$$R_{HRQDA} = \frac{R_{HRQDA}^1 + R_{HRQDA}^2}{2}, \qquad R_{HRQDA}^m = \mathbb{P}(\text{incorrectly classify } x \text{ to class } m).$$

To show the consistency of the misclassification rate of our proposed HRQDA method, we need the following additional assumptions.

**Assumption 5.** $\text{tr}(\boldsymbol{\Xi}_i) \asymp t_0(p)$ *for each* $i = 1, 2$. *And*

$$\sigma_Q(p) := \sqrt{\|\boldsymbol{\Sigma}_1^{1/2}\boldsymbol{\Omega}_2\boldsymbol{\Sigma}_1^{1/2} - \mathbf{I}_p\|_F^2 + t_0(p)^{-1}p\|\boldsymbol{\mu}_2 - \boldsymbol{\mu}_1\|^2} \asymp p.$$

**Assumption 6.** $r_i = \|\boldsymbol{\Xi}_i^{-1/2}(\boldsymbol{x} - \boldsymbol{\mu}_i)\|$ *satisfies* $\text{Var}(r_i^2) \lesssim p\sqrt{p}$ *and* $\text{Var}(r_i) \lesssim \sqrt{p}$, *for* $i = 1, 2$.

Assumption 5 assume the signal of the difference between the two distribution is larger enough. Assumption 6 is needed to show the consistency of the trace estimator $\widehat{\text{tr}(\boldsymbol{\Xi}_i)}$.

**Theorem 5.** *If* $\boldsymbol{\varepsilon}_i$, $\boldsymbol{\Sigma}_i$, $\boldsymbol{\Omega}_i$, $\mathbf{S}_i$ *for* $i = 1, 2$ *satisfy Assumptions 1-4 and Assumptions 5,6 hold. Assume that* $n_1 \asymp n_2$ *and* $n := \min\{n_1, n_2\}$, *we have*

$$|R_{HRQDA} - R_{QDA}| = O_p\{\lambda_n^{1-q/2}s_0(p)^{1/2} + \lambda_n^{1-q}s_0(p)\}.$$

The result in Theorem 5 show that HRQDA is able to mimic the optimal Bayes rule consistently under some mild assumptions, which is similar to Theorem 4.2 in Cai & Zhang (2021).

## A.2 SIMULATION

We compare our proposed method, HRQDA, with the SQDA method proposed by Li & Shao (2015) and the SeQDA method proposed by Wu et al. (2019). The SQDA method estimates the covariance matrix using the banding method proposed by Bickel & Levina (2008b), while the SeQDA method estimates the covariance matrix of the transformed sample by simplifying the structure of the covariance matrices.

We consider the following three elliptical distributions:

- (i): Multivariate normal distribution: $\boldsymbol{X}_{i1} \sim \mathcal{N}(\boldsymbol{\mu}_1, \boldsymbol{\Sigma}_1)$, $\boldsymbol{X}_{i2} \sim \mathcal{N}(\boldsymbol{\mu}_2, \boldsymbol{\Sigma}_2)$;
- (ii): Multivariate $t$-distribution: $\boldsymbol{X}_{i1} \sim t(\boldsymbol{\mu}_1, \boldsymbol{\Sigma}_1, 3)/\sqrt{3}$, $\boldsymbol{X}_{i2} \sim t(\boldsymbol{\mu}_2, \boldsymbol{\Sigma}_2, 3)/\sqrt{3}$;
- (iii): Multivariate mixture normal distribution: $\boldsymbol{X}_{i1} \sim \mathcal{MN}(\boldsymbol{\mu}_1, \boldsymbol{\Sigma}_1, 10, 0.8)/\sqrt{22.8}$, $\boldsymbol{X}_{i2} \sim \mathcal{MN}(\boldsymbol{\mu}_2, \boldsymbol{\Sigma}_2, 10.0.8)/\sqrt{22.8}$.

We consider three models for the covariance matrix:

- Model I: $\boldsymbol{\Sigma}_1 = (0.6^{|i-j|})_{1 \leq i,j \leq p}$, $\boldsymbol{\Sigma}_2 = \mathbf{I}_p$;
- Model II: $\boldsymbol{\Sigma}_1 = (0.6^{|i-j|})_{1 \leq i,j \leq p}$, $\boldsymbol{\Sigma}_2 = 0.5\mathbf{I}_p + 0.5\mathbf{1}_p\mathbf{1}_p^\top$;
- Model III: $\boldsymbol{\Omega}_1 = (0.6^{|i-j|})_{1 \leq i,j \leq p}$, $\boldsymbol{\Sigma}_1 = \boldsymbol{\Omega}_1^{-1}$, $\boldsymbol{\Sigma}_2 = \boldsymbol{\Omega}_1$.

The covariance matrices in Model I are approximately banded. In Model II, $\boldsymbol{\Sigma}_2$ satisfies the structural assumption in Wu et al. (2019) but violates the sparsity condition in Li & Shao (2015), while in Model III, $\boldsymbol{\Sigma}_1$ satisfies the latter but violates both. We set $\boldsymbol{\mu}_1 = \mathbf{0}$ and $\boldsymbol{\mu}_2 = 0.1 \times \mathbf{1}_p$, and generate $n_1 = n_2 = 100$ training and test samples of the same size and two dimensions $p = 120, 240$.

Table 2 reports the average classification rates. HRQDA generally performs best. In Model I, SQDA benefits from the banded structure and outperforms SeQDA; HRQDA is comparable under normality but superior under heavy-tailed distributions. In Model II, SQDA performs worst due to structural mismatch, while HRQDA consistently outperforms SeQDA in non-normal settings. In Model III, HRQDA still achieves the best accuracy. These results confirm that HRQDA is robust and effective across various distributions and covariance structures.

Table 2: Average classification rate (%) and standard deviation (in parenthesis) of each method.

| Model | Dist. | $p = 120$ | | | $p = 240$ | | |
| | | HRQDA | SQDA | SeQDA | HRQDA | SQDA | SeQDA |
|---|---|---|---|---|---|---|---|
| Model I | (i) | 0.99(0.01) | 0.94(0.08) | 0.63(0.03) | 1(0) | 0.96(0.08) | 0.64(0.04) |
| | (ii) | 0.95(0.06) | 0.64(0.11) | 0.55(0.04) | 0.97(0.08) | 0.60(0.09) | 0.54(0.05) |
| | (iii) | 0.92(0.13) | 0.55(0.07) | 0.52(0.04) | 0.96(0.10) | 0.55(0.08) | 0.51(0.04) |
| Model II | (i) | 1(0.01) | 0.77(0.10) | 0.97(0.01) | 1(0) | 0.78(0.14) | 1(0.01) |
| | (ii) | 0.99(0.01) | 0.55(0.06) | 0.68(0.03) | 1(0) | 0.55(0.05) | 0.68(0.03) |
| | (iii) | 0.99(0.01) | 0.53(0.04) | 0.54(0.05) | 1(0) | 0.53(0.04) | 0.53(0.03) |
| Model III | (i) | 1(0) | 1(0.02) | 0.76(0.04) | 1(0) | 1(0.02) | 0.76(0.04) |
| | (ii) | 1(0) | 0.82(0.11) | 0.66(0.03) | 1(0) | 0.81(0.12) | 0.65(0.03) |
| | (iii) | 0.77(0.09) | 0.60(0.02) | 0.52(0.05) | 0.78(0.10) | 0.60(0.02) | 0.51(0.04) |

## A.3 REAL DATA APPLICATION

We used the gene expression dataset GSE12288 from Sinnaeve et al. (2009), which includes 110 coronary artery disease (CAD) patients (CADi > 23) and 112 healthy controls. After applying two-sample $t$-tests, 297 genes with $p$-values below 0.01 were retained. To evaluate performance, we compared our HRQDA method with SQDA and SeQDA by randomly splitting the data into training (73 CAD, 75 control) and testing (37 CAD, 37 control) sets, repeating this process 200 times. Classification accuracy was averaged over the repetitions.

The performance of classifiers was evaluated using four key metrics:

- **Accuracy** (Acc): Proportion of correctly classified samples:

$$\text{Acc} = \frac{TP + TN}{TP + TN + FP + FN}.$$

- **Specificity** (Spec): Proportion of true negatives correctly identified:

$$\text{Spec} = \frac{TN}{TN + FP}.$$

- **Sensitivity** (Sens): Proportion of true positives correctly identified:

$$\text{Sens} = \frac{TP}{TP + FN}.$$

- **Matthews Correlation Coefficient** (MCC): Balanced measure of classification quality:

$$\text{MCC} = \frac{TP \cdot TN - FP \cdot FN}{\sqrt{(TP + FP)(TP + FN)(TN + FP)(TN + FN)}},$$

where $TP$ (true positive), $TN$ (true negative), $FP$ (false positive), and $FN$ (false negative) represent the counts of respective classification outcomes. All metrics range between 0 and 1, except MCC which ranges between $-1$ and 1, with higher values indicating better performance.

Table 3 shows that HRQDA outperforms SQDA and SeQDA, achieving the highest mean accuracy (0.760) and MCC (0.527). It also has the best sensitivity (0.821) and maintains good specificity (0.708), showing strong ability to detect CAD cases reliably.

## B PERFORMANCE OF THE TEST UNDER $\varepsilon$-CONTAMINATION

For the one-sample testing problem, we consider $n = 100$ and $p = 120$. The uncontaminated data are generated from a multivariate $t_3$ distribution with mean vector $\boldsymbol{\mu}$ and covariance matrix $\boldsymbol{\Sigma}$ with entries

$$\Sigma_{ij} = 0.8^{|i-j|}, \qquad 1 \leq i, j \leq p.$$

Table 3: Comparison of evaluation metrics and standard deviation (in parenthesis) for each method.

| Method | Accuracy | Specificity | Sensitivity | MCC |
|--------|----------|-------------|-------------|-----|
| HRQDA | 0.760 (0.042) | 0.708 (0.088) | 0.821 (0.069) | 0.527 (0.082) |
| SQDA | 0.710 (0.051) | 0.707 (0.112) | 0.702 (0.111) | 0.429 (0.103) |
| SeQDA | 0.729 (0.051) | 0.685 (0.084) | 0.772 (0.070) | 0.461 (0.102) |

To introduce $\varepsilon$-contamination, we randomly select $\varepsilon n$ observations and replace them by independent noise drawn from $\mathcal{N}_p(0, \text{strength} \cdot I_p)$, where the contamination rate $\varepsilon$ takes values in $\{0, 0.05, 0.10, 0.15, 0.20\}$ and the contamination strength is in $\{5, 10, 20\}$.
We consider three mean configurations:

$$\boldsymbol{\mu}_1 = \mathbf{0}_p, \qquad \boldsymbol{\mu}_2 = \frac{0.15}{1-\varepsilon} \boldsymbol{\Sigma}^{1/2}(1,1,1,0,\ldots,0)^\top, \qquad \boldsymbol{\mu}_3 = \frac{0.1}{1-\varepsilon} \boldsymbol{\Sigma}^{1/2}(\underbrace{1,\ldots,1}_{30},0,\ldots,0)^\top,$$

where the factor $0.15/(1-\varepsilon)$ is used to keep the effective signal strength comparable across different contamination rates. Here $\boldsymbol{\mu}_1$ corresponds to the null (empirical size), $\boldsymbol{\mu}_2$ to a sparse alternative (nonzero in the first 3 coordinates), and $\boldsymbol{\mu}_3$ to a dense alternative (nonzero in the first 30 coordinates). For each setting, we repeat the experiment 500 times and record the empirical rejection probabilities of the max-type test, the sum-type test, and the Cauchy combination test.
The results under $\varepsilon$-contamination are summarized in Tables 4–6 below. Table 4 reports empirical size under the null ($\boldsymbol{\mu}_1$), while Tables 5 and 6 report empirical power under the sparse ($\boldsymbol{\mu}_2$) and dense ($\boldsymbol{\mu}_3$) alternatives, respectively. Overall, the sum, max, and Cauchy combination tests maintain sizes close to the nominal level and display reasonable power even when up to 20% of the observations are contaminated.

Table 4: Empirical size under $\varepsilon$-contamination for $\boldsymbol{\mu}_1$.

| $\varepsilon$ | Strength | $\hat{\alpha}_{\max}$ | $\hat{\alpha}_{\text{sum}}$ | $\hat{\alpha}_{\text{Cauchy}}$ |
|------|----------|------|------|------|
| 0.00 | 5 | 0.056 | 0.050 | 0.060 |
| 0.05 | 5 | 0.040 | 0.028 | 0.030 |
| 0.10 | 5 | 0.040 | 0.052 | 0.052 |
| 0.15 | 5 | 0.042 | 0.030 | 0.042 |
| 0.20 | 5 | 0.042 | 0.034 | 0.042 |
| 0.00 | 10 | 0.046 | 0.048 | 0.050 |
| 0.05 | 10 | 0.042 | 0.034 | 0.038 |
| 0.10 | 10 | 0.032 | 0.040 | 0.036 |
| 0.15 | 10 | 0.044 | 0.028 | 0.036 |
| 0.20 | 10 | 0.050 | 0.036 | 0.048 |
| 0.00 | 20 | 0.050 | 0.052 | 0.052 |
| 0.05 | 20 | 0.036 | 0.052 | 0.056 |
| 0.10 | 20 | 0.040 | 0.044 | 0.050 |
| 0.15 | 20 | 0.036 | 0.034 | 0.036 |
| 0.20 | 20 | 0.042 | 0.030 | 0.044 |

Notes: $n = 100$, $p = 120$. Errors are generated from a multivariate $t_3$ distribution with covariance $\Sigma_{ij} = 0.8^{|i-j|}$. Here $\boldsymbol{\mu}_1$ corresponds to the null (empirical size). Contamination follows an $\varepsilon$-contamination scheme with $\varepsilon \in \{0, 0.05, 0.10, 0.15, 0.20\}$ and strength in $\{5, 10, 20\}$. Each entry is based on 500 Monte Carlo replications.

For HRQDA, we consider a two-class classification problem with $p = 120$ and $t_3$ distribution. In each replicate, we generate a training sample of size 100 and an independent test sample of size 100 (again 50 per class). Class 1 follows a multivariate $t_3$ distribution with mean $0_p$ and covariance matrix $\Sigma_{ij} = 0.8^{|i-j|}$, while Class 2 follows a multivariate $t_3$ distribution with mean $0.1 \cdot \mathbf{1}_p$ and covariance matrix $\mathbf{I}_p$. We then contaminate a fraction $\varepsilon \in \{0, 0.05, 0.10, 0.15, 0.20\}$ of the observations by replacing them with $\mathcal{N}_p(0, \text{strength} \cdot I_p)$ noise (using the same strengths $\{5, 10, 20\}$ as above), and record the classification accuracy of HRQDA on the test set. Table 7 reports the average classification accuracy (in %) over 500 replications.

Table 5: Empirical power under $\varepsilon$-contamination for $\boldsymbol{\mu}_2$ (sparse mean shift).

| $\varepsilon$ | Strength | $\hat{\beta}_{\max}$ | $\hat{\beta}_{\mathrm{sum}}$ | $\hat{\beta}_{\mathrm{Cauchy}}$ |
|---|---|---|---|---|
| 0.00 | 5 | 0.526 | 0.330 | 0.534 |
| 0.05 | 5 | 0.542 | 0.320 | 0.544 |
| 0.10 | 5 | 0.526 | 0.304 | 0.510 |
| 0.15 | 5 | 0.496 | 0.262 | 0.462 |
| 0.20 | 5 | 0.538 | 0.300 | 0.522 |
| 0.00 | 10 | 0.522 | 0.332 | 0.532 |
| 0.05 | 10 | 0.534 | 0.334 | 0.552 |
| 0.10 | 10 | 0.504 | 0.280 | 0.492 |
| 0.15 | 10 | 0.506 | 0.264 | 0.492 |
| 0.20 | 10 | 0.466 | 0.186 | 0.432 |
| 0.00 | 20 | 0.582 | 0.350 | 0.560 |
| 0.05 | 20 | 0.524 | 0.310 | 0.544 |
| 0.10 | 20 | 0.510 | 0.294 | 0.502 |
| 0.15 | 20 | 0.498 | 0.250 | 0.474 |
| 0.20 | 20 | 0.488 | 0.216 | 0.460 |

Notes: same data-generating mechanism as in Table 4, but $\boldsymbol{\mu}_2$ corresponds to a sparse mean shift. Entries are empirical power (rejection probabilities under the alternative) based on 500 Monte Carlo replications.

Table 6: Empirical power under $\varepsilon$-contamination for $\boldsymbol{\mu}_3$ (dense mean shift).

| $\varepsilon$ | Strength | $\hat{\beta}_{\max}$ | $\hat{\beta}_{\mathrm{sum}}$ | $\hat{\beta}_{\mathrm{Cauchy}}$ |
|---|---|---|---|---|
| 0.00 | 5 | 0.336 | 0.526 | 0.524 |
| 0.05 | 5 | 0.306 | 0.500 | 0.514 |
| 0.10 | 5 | 0.310 | 0.518 | 0.514 |
| 0.15 | 5 | 0.316 | 0.512 | 0.532 |
| 0.20 | 5 | 0.270 | 0.496 | 0.498 |
| 0.00 | 10 | 0.326 | 0.512 | 0.490 |
| 0.05 | 10 | 0.342 | 0.544 | 0.546 |
| 0.10 | 10 | 0.288 | 0.484 | 0.486 |
| 0.15 | 10 | 0.280 | 0.528 | 0.496 |
| 0.20 | 10 | 0.250 | 0.444 | 0.444 |
| 0.00 | 20 | 0.330 | 0.558 | 0.558 |
| 0.05 | 20 | 0.314 | 0.560 | 0.528 |
| 0.10 | 20 | 0.320 | 0.524 | 0.514 |
| 0.15 | 20 | 0.298 | 0.480 | 0.490 |
| 0.20 | 20 | 0.254 | 0.450 | 0.464 |

Notes: same data-generating mechanism as in Table 4, but $\boldsymbol{\mu}_3$ corresponds to a dense mean shift. Entries are empirical power based on 500 Monte Carlo replications.

Table 7: Classification accuracy (%) of HRQDA under $\varepsilon$-contamination.

| Strength | 0 | 0.05 | 0.10 | 0.15 | 0.20 |
|---|---|---|---|---|---|
| 5 | 96.82 | 95.12 | 92.90 | 90.72 | 88.50 |
| 10 | 96.82 | 95.13 | 92.90 | 90.71 | 88.42 |
| 20 | 96.92 | 95.13 | 92.80 | 90.70 | 88.41 |

Notes: Each entry is the average test-set classification accuracy (in %) of HRQDA over 500 Monte Carlo replications. Training and test samples have size 100 each (50 observations per class). Class 1 has mean $\mathbf{0_p}$ and covariance $\Sigma_{ij} = 0.8^{|i-j|}$, and Class 2 has mean $0.1 \cdot \mathbf{1}_p$ and covariance $\mathbf{I}_p$. A fraction $\varepsilon$ of the observations is replaced by $\mathcal{N}_p(0, \text{strength} \cdot \mathbf{I}_p)$ noise.

We observe that HRQDA retains high classification accuracy under moderate levels of $\varepsilon$-contamination, with performance degrading only gradually as the contamination rate increases, which is consistent with the robust behavior suggested by our theoretical developments. Due to the strict page and response-length constraints and the substantial additional space that a full grid of cellwise contamination scenarios would require, we focused our numerical study on $\varepsilon$-contamination and leave a systematic investigation of cellwise contamination to future work.

## C  ADDITIONAL METHODS FOR ONE-SAMPLE LOCATION TEST PROBLEM

For comparison, we also consider the test procedures proposed by Feng & Sun (2016) and Liu et al. (2024), which are designed for the sparsity structure of the location parameter $\boldsymbol{\mu}$.

$$T_{SUM2} = \frac{2}{n(n-1)} \sum\sum_{i<j} U\left(\tilde{\mathbf{D}}_{ij}^{-1/2} \boldsymbol{X}_i\right)^\top U\left(\tilde{\mathbf{D}}_{ij}^{-1/2} \boldsymbol{X}_j\right),$$

$$T_{MAX2} = n\hat{\zeta}_1^2 p \left\| \tilde{\mathbf{D}}^{-1/2}\tilde{\boldsymbol{\mu}} \right\|_\infty^2 \left(1 - n^{-1/2}\right),$$

$$T_{CC2} = 1 - G\left[0.5\tan\left\{(0.5 - p_{MAX2})\pi\right\} + 0.5\tan\left\{(0.5 - p_{SUM2})\pi\right\}\right],$$

where $p_{MAX2}$ and $p_{SUM2}$ are the p-values of $T_{MAX2}$ and $T_{SUM2}$, respectively. Here $\tilde{\boldsymbol{\mu}}$ and $\tilde{\mathbf{D}}_{ij}$ are the estimator of spatial-median and diagonal matrix of $\boldsymbol{\Sigma}$ by the following algorithm:

(i)  $\tilde{\boldsymbol{\varepsilon}}_i \leftarrow \tilde{\mathbf{D}}^{-1/2}\left(\mathbf{X}_i - \tilde{\boldsymbol{\mu}}\right), \quad j = 1, \cdots, n;$

(ii)  $\tilde{\boldsymbol{\mu}} \leftarrow \tilde{\boldsymbol{\mu}} + \frac{\tilde{\mathbf{D}}^{1/2} \sum_{j=1}^n U(\tilde{\boldsymbol{\varepsilon}}_i)}{\sum_{j=1}^n \|\tilde{\boldsymbol{\varepsilon}}_i\|^{-1}};$

(iii)  $\tilde{\mathbf{D}} \leftarrow p\tilde{\mathbf{D}}^{1/2} \operatorname{diag}\left\{n^{-1} \sum_{j=1}^n U\left(\tilde{\boldsymbol{\varepsilon}}_i\right) U\left(\tilde{\boldsymbol{\varepsilon}}_i\right)^\top\right\} \tilde{\mathbf{D}}^{1/2}.$

Next, we demonstrate that, under mild regularity conditions, the sum-type test statistic $T_{\mathrm{SUM2}}$ is asymptotically independent of the max-type test statistic $T_{\mathrm{MAX}}$. Furthermore, the max-type test statistic $T_{\mathrm{MAX2}}$ is also asymptotically independent of the sum-type test statistic $T_{\mathrm{SUM1}}$.

**Theorem 6.** *Under Assumptions 1-4, if $\|\boldsymbol{\mu}\|_\infty = o(n^{-1/2})$ and $\|\boldsymbol{\mu}\| = o(p^{1/4}n^{-1/2})$, as $n, p \to \infty$, and Theorem 7 in Liu et al. (2024) holds, $T_{SUM2}/\sigma_n$ is asymptotically independent with $T_{MAX}$, $T_{SUM}$ is asymptotically independent with $T_{MAX2} - 2\log p + \log\log p$.*

In practice, we could not know the sparsity level of the alternative, either $\boldsymbol{\Omega}^{1/2}\boldsymbol{\mu}$ or $\boldsymbol{\mu}$, so we suggest to use Cauchy combination test to combine all the four test procedures as follow:

$$T_{CC3} = 1 - G\Big[\frac{1}{4}\tan\left\{(0.5 - p_{MAX})\pi\right\} + \frac{1}{4}\tan\left\{(0.5 - p_{SUM})\pi\right\}$$
$$+ \frac{1}{4}\tan\left\{(0.5 - p_{MAX2})\pi\right\} + \frac{1}{4}\tan\left\{(0.5 - p_{SUM2})\pi\right\}\Big]. \tag{10}$$

We have supplemented the empirical sizes of the Cauchy combination test $T_{CC3}$ under the null hypothesis as described in Section 4. Table 8 reports the empirical sizes of $T_{CC3}$, which are consistently around $5\%$, indicating that $T_{CC3}$ can control the empirical size very well.

Next, we compare $T_{CC3}$ with $T_{CC1}$ proposed in Section 3 and $T_{CC2}$ from Liu et al. (2024). Specifically, the comparison is carried out under distributions (i)–(iii) and Models I–IV, using the same parameter settings and data-generating mechanisms for the alternatives as described in Section 4. Figure 2 displays the power curves. We observe that $T_{CC2}$ tends to outperform $T_{CC1}$ under Model I, while the reverse holds for Models III and IV. In the case of Model II, $T_{CC2}$ exhibits lower power than $T_{CC1}$ when the signal sparsity $s$ is small, but surpasses $T_{CC1}$ as $s$ increases. Overall, the relative performance of $T_{CC1}$ and $T_{CC2}$ is highly sensitive to the underlying model structure.

In contrast, the proposed $T_{CC3}$ demonstrates uniformly strong performance across all scenarios, offering both robustness to distributional variation and adaptability to different signal sparsity. It often achieves the highest power, making it a reliable choice in practice.

Table 8: Empirical sizes (%) of $T_{CC3}$ under different models with $n = 100$.

| Dist. | Test | Model I | | Model II | | Model III | | Model IV | |
|---|---|---|---|---|---|---|---|---|---|
| | | $p = 120$ | $p = 240$ | $p = 120$ | $p = 240$ | $p = 120$ | $p = 240$ | $p = 120$ | $p = 240$ |
| (i) | $T_{CC3}$ | 5.8 | 4.3 | 4.7 | 5.2 | 5.0 | 4.4 | 4.1 | 5.6 |
| (ii) | $T_{CC3}$ | 5.9 | 4.6 | 5.4 | 4.2 | 5.0 | 4.7 | 5.9 | 4.5 |
| (iii) | $T_{CC3}$ | 5.1 | 4.9 | 4.2 | 5.6 | 4.7 | 5.9 | 4.9 | 5.5 |

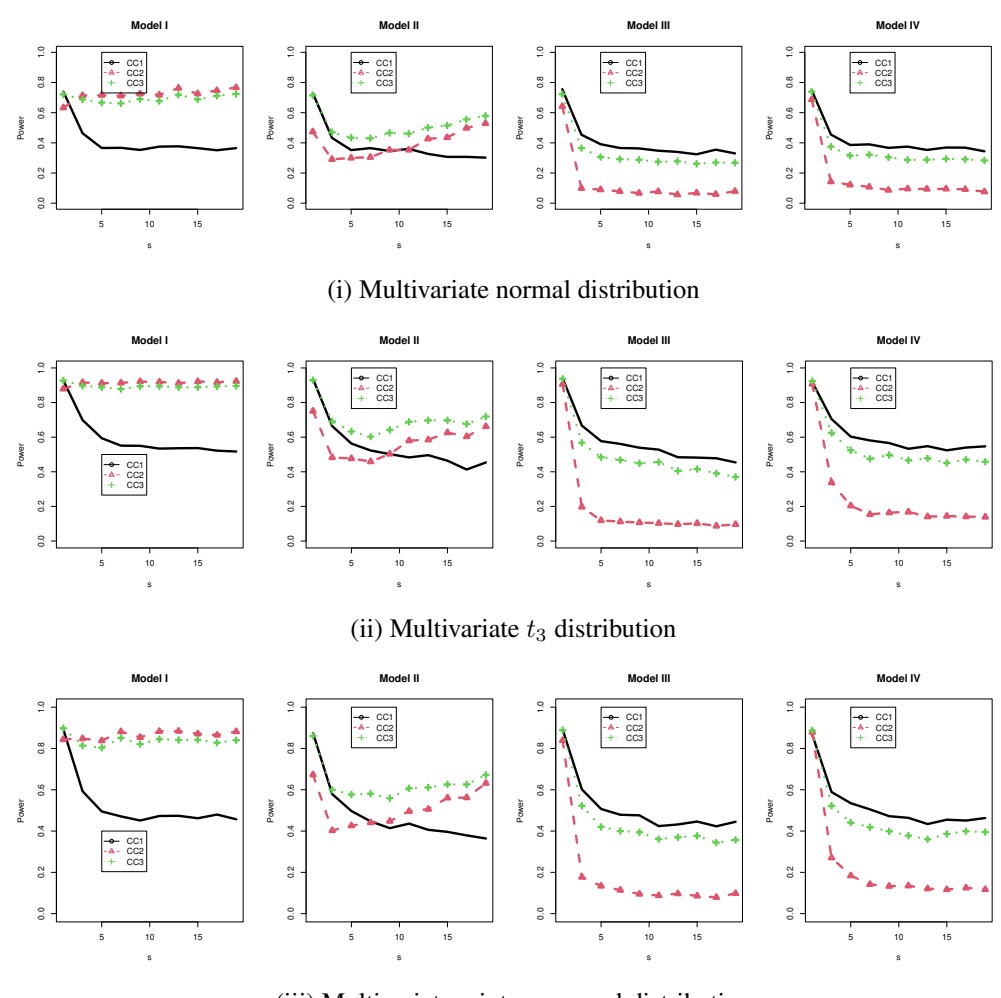

(i) Multivariate normal distribution

(ii) Multivariate $t_3$ distribution

(iii) Multivariate mixture normal distribution

Figure 2: Power curves of three Cauchy combination tests with different sparsity, models and $n = 100, p = 120$.

# D  INSENSITIVITY OF PARAMETERS TO THE ALGORITHM

The bandwidth parameter $h$ exhibits low sensitivity to the final results, in contrast to the algorithm's higher sensitivity to the choice of initial values. Specifically, when appropriate initial values are selected, the following approximation holds:

$$\frac{p}{n} \sum_{i=1}^{n} U\left(\hat{\boldsymbol{\varepsilon}}_i\right) U\left(\hat{\boldsymbol{\varepsilon}}_i\right)^{\top} \approx \mathbf{I}_p,$$

where $\mathbf{I}_p$ denotes the $p$-dimensional identity matrix. Under such circumstances, satisfactory results can be obtained regardless of the specific choice of bandwidth parameter $h$. We typically adopt $h = 3$ as the default value to balance estimation accuracy and computational cost: while moderately larger values of $h$ may yield marginal improvements in accuracy, the benefits are limited and come with increased computational time.

To visually demonstrate the influence of $h$, we first present experimental results based on simulated data. Specifically, our simulations generate observations $\boldsymbol{X}_i$ from a multivariate $t$-distribution with the following specifications:

$$\boldsymbol{X}_i \sim t(\boldsymbol{\mu}, \boldsymbol{\Sigma}, 3)/\sqrt{3},$$

where we set the sample size $n = 100$ and dimensionality $p = 120$. The covariance matrix $\boldsymbol{\Sigma}$ follows an autoregressive structure defined by $\boldsymbol{\Sigma} = (0.6^{|i-j|})_{1 \leq i,j \leq p}$.

Table 9: Influence of bandwidth parameter $h$ on robust mean and covariance estimation.

| $h$ | 1 | 2 | 3 | 4 | 5 | 10 | 20 |
|---|---|---|---|---|---|---|---|
| $\|\hat{\boldsymbol{\mu}} - \boldsymbol{\mu}\|_2$ | 1.76 | 1.76 | 1.76 | 1.76 | 1.76 | 1.76 | 1.76 |
| $\|\hat{\boldsymbol{\Sigma}} - \boldsymbol{\Sigma}\|_F$ | 3.82 | 3.79 | 3.77 | 3.76 | 3.74 | 3.66 | 3.65 |

Here, $\|\cdot\|_2$ represents the $L_2$-norm for vectors, and $\|\cdot\|_F$ denotes the Frobenius norm for matrices. These results already indicate that the bandwidth parameter $h$ exerts only a limited influence on the quality of the robust mean and covariance estimators.

To further quantify the practical impact of tuning parameters on our proposed tests, we conduct a sensitivity study for the banding width $h$, the number of bootstrap iterations $M$, the regularization parameter $\lambda$ in the SGLASSO step, and the sample size $(n, p)$. The tables below report empirical size ($\hat{\alpha}.$), empirical power ($\hat{\beta}.$), and average runtime (in seconds) in a representative one-sample setting with $n = 100$, $p = 120$, multivariate $t_3$ errors with covariance $\Sigma_{ij} = (0.8^{|i-j|})_{1 \leq i,j \leq p}$, and a sparse mean shift

$$\boldsymbol{\mu} = \boldsymbol{\Sigma}^{1/2}(1, 1, 1, 0, \ldots, 0)^{\top}$$

under the alternative. Each entry is based on 500 Monte Carlo replications.

Table 10 shows that the proposed tests are quite stable with respect to the banding width $h$. Across $h \in \{1, 2, 3, 4, 5, 10\}$, the empirical sizes of the max, sum, and Cauchy combination tests remain close to the nominal level, and the powers are broadly comparable, with slightly better performance for moderate banding (e.g., $h = 3$–$5$). The average runtime changes very little with $h$. This supports our default choice $h = 3$ and suggests that practitioners can safely vary $h$ within a moderate range without materially affecting performance.

Table 11 examines the number of bootstrap iterations $M$. For $M \in \{20, 50, 100, 200\}$, the empirical sizes are again close to $0.05$ and the powers increase only mildly with $M$, while the runtime grows approximately linearly (from about $650$ to $2300$ seconds in this experiment). This indicates that $M$ around several tens already yields stable behavior, and our default $M = 50$ represents a reasonable compromise between accuracy and computational cost.

Table 12 studies the SGLASSO regularization parameter $\lambda$. We find that overly small regularization (e.g., $\lambda = 0.05$) leads to noticeable size distortion and very high rejection probabilities, whereas moderate to larger values ($\lambda = 0.1, 0.2, 0.3$) keep empirical size closer to the nominal level but with some loss of power when $\lambda$ becomes too large. The theoretically motivated choice $\lambda = 0.1$ lies in a region where both size and power are well behaved, and runtime decreases slightly as $\lambda$ increases. These results suggest that practitioners should avoid very small $\lambda$, and that a range around the default (e.g., $\lambda$ between $0.1$ and $0.2$) is acceptable in practice.

Finally, Table 13 reports average runtime for several combinations of $(n, p)$. For the range of $n$ considered, the runtime varies only mildly with $n$ at fixed $p$, whereas it increases substantially with $p$, reflecting that the computational cost

is dominated by operations on $p \times p$ covariance and precision matrices. This provides a concrete indication of the scalability of the proposed procedures: they are feasible for moderate to high dimensions, with computational burden growing primarily in $p$ rather than $n$.

Overall, these sensitivity results indicate that (i) the proposed tests are reasonably robust to moderate perturbations of $h$, $M$, and $\lambda$ around the recommended defaults, and (ii) the computational cost behaves in a predictable way, which we believe will help practitioners choose tuning parameters and anticipate run times in their own applications.

Table 10: Sensitivity to banding width $h$.

| $h$ | $\hat{\alpha}_{\max}$ | $\hat{\alpha}_{\mathrm{sum}}$ | $\hat{\alpha}_{\mathrm{cc}}$ | $\hat{\beta}_{\max}$ | $\hat{\beta}_{\mathrm{sum}}$ | $\hat{\beta}_{\mathrm{cc}}$ | runtime |
|---|---|---|---|---|---|---|---|
| 1 | 0.042 | 0.046 | 0.050 | 0.550 | 0.392 | 0.572 | 946.28 |
| 2 | 0.048 | 0.044 | 0.052 | 0.488 | 0.378 | 0.548 | 934.95 |
| 3 | 0.056 | 0.048 | 0.052 | 0.534 | 0.440 | 0.591 | 933.89 |
| 4 | 0.054 | 0.044 | 0.048 | 0.548 | 0.446 | 0.626 | 925.86 |
| 5 | 0.054 | 0.042 | 0.044 | 0.556 | 0.480 | 0.603 | 928.32 |
| 10 | 0.046 | 0.066 | 0.054 | 0.545 | 0.546 | 0.644 | 930.70 |

Table 11: Sensitivity to the number of bootstrap iterations $M$.

| $M$ | $\hat{\alpha}_{\max}$ | $\hat{\alpha}_{\mathrm{sum}}$ | $\hat{\alpha}_{\mathrm{cc}}$ | $\hat{\beta}_{\max}$ | $\hat{\beta}_{\mathrm{sum}}$ | $\hat{\beta}_{\mathrm{cc}}$ | runtime |
|---|---|---|---|---|---|---|---|
| 20 | 0.040 | 0.050 | 0.049 | 0.478 | 0.354 | 0.584 | 655.14 |
| 50 | 0.056 | 0.048 | 0.052 | 0.534 | 0.440 | 0.594 | 933.89 |
| 100 | 0.054 | 0.052 | 0.052 | 0.550 | 0.406 | 0.584 | 1385.56 |
| 200 | 0.062 | 0.050 | 0.058 | 0.548 | 0.418 | 0.582 | 2316.18 |

Table 12: Sensitivity to the regularization parameter $\lambda$.

| $\lambda$ | $\hat{\alpha}_{\max}$ | $\hat{\alpha}_{\mathrm{sum}}$ | $\hat{\alpha}_{\mathrm{cc}}$ | $\hat{\beta}_{\max}$ | $\hat{\beta}_{\mathrm{sum}}$ | $\hat{\beta}_{\mathrm{cc}}$ | runtime |
|---|---|---|---|---|---|---|---|
| 0.05 | 0.172 | 0.314 | 0.258 | 0.724 | 0.884 | 0.878 | 1082.52 |
| 0.1 | 0.056 | 0.048 | 0.052 | 0.544 | 0.466 | 0.562 | 933.89 |
| 0.2 | 0.046 | 0.067 | 0.066 | 0.424 | 0.364 | 0.474 | 794.08 |
| 0.3 | 0.046 | 0.103 | 0.078 | 0.382 | 0.402 | 0.470 | 737.93 |

# E  PROOFS OF THEORETICAL RESULTS

Recall that for $i = 1, 2, \cdots, n$, $\boldsymbol{U}_i = U(\boldsymbol{\varepsilon}_i) = U\{\boldsymbol{\Omega}^{1/2}(\boldsymbol{X}_i - \boldsymbol{\mu})\}$ and $r_i = \|\boldsymbol{\varepsilon}_i\| = \|\boldsymbol{\Omega}^{1/2}(\boldsymbol{X}_i - \boldsymbol{\mu})\|$ as the scale-invariant spatial-sign and radius of $\boldsymbol{X}_i - \boldsymbol{\mu}$, where $U(\boldsymbol{X}) = \boldsymbol{X}/\|\boldsymbol{X}\|\mathbb{I}(\boldsymbol{X} \neq 0)$ is the multivariate sign function of $\boldsymbol{X}$, with $\mathbb{I}(\cdot)$ being the indicator function. The moments of $r_i$ is defined as $\zeta_k = \mathbb{E}\left(r_i^{-k}\right)$. We denote the estimated version $\boldsymbol{U}_i$ and $r_i$ as $\hat{r}_i = \|\hat{\boldsymbol{\Omega}}^{1/2}(\boldsymbol{X}_i - \boldsymbol{\mu})\|$ and $\hat{U}_i = \hat{\boldsymbol{\Omega}}^{1/2}(\boldsymbol{X}_i - \boldsymbol{\mu})/\|\hat{\boldsymbol{\Omega}}^{1/2}(\boldsymbol{X}_i - \boldsymbol{\mu})\|$, respectively, $i = 1, 2, \cdots, n$. Finally, we denote various positive constants by $C, C_1, C_2, \ldots$ without mentioning this explicitly.

## E.1  THE LEMMAS TO BE USED

The following result is a one-sample special case of Lemma 1 in Feng et al. (2016).

**Lemma 3.** *Under Assumption 1, for any matrix* $\mathbf{M}$, *we have*

$$\mathbb{E}[\{U(\boldsymbol{\varepsilon}_i)^{\top}\mathbf{M}U(\boldsymbol{\varepsilon}_i)\}^2] = O\{p^{-2}\mathrm{tr}(\mathbf{M}^{\top}\mathbf{M})\}.$$

As it plays a key role in our analysis, we restate Theorem 1 from Lu & Feng (2025) below.

**Lemma 4.** *Under Assumptions 1-4,* $\hat{\boldsymbol{\Omega}}$ *defined in Lemma 1 satisfies the following property. When* $n,p$ *are sufficiently large, there exist constants* $C_{\eta,T}$ *and* $C$, *such that if we pick*

$$\lambda_n = T\left\{ \frac{\sqrt{2}C(8 + \eta^2 C_{\eta,T})}{\eta^2}\sqrt{\frac{\log p}{n}} + \frac{C_{\eta,T}}{\sqrt{p}} \right\},$$

Table 13: Average runtime (seconds) for different $(n, p)$.

|  | $n = 100$ | $n = 200$ | $n = 400$ | $n = 600$ |
|---|---|---|---|---|
| $p = 120$ | 24.29 | 23.35 | 23.67 | 23.35 |
| $p = 240$ | 186.01 | 181.78 | 165.88 | 157.70 |
| $p = 480$ | 1921.63 | 1506.36 | 1405.67 | 1358.17 |

*with probability larger than $1 - 2p^{-2}$, the following inequalities hold:*

$$\|\hat{\mathbf{\Omega}} - \mathbf{\Omega}\|_\infty \le 4\|\mathbf{\Omega}\|_{L_1}\lambda_n,$$

$$\|\hat{\mathbf{\Omega}} - \mathbf{\Omega}\|_{\mathrm{op}} \le \|\hat{\mathbf{\Omega}} - \mathbf{\Omega}\|_{L_1} \le C_4\lambda_n^{1-q}s_0(p),$$

$$p^{-1}\|\hat{\mathbf{\Omega}} - \mathbf{\Omega}\|_F^2 \le C_5\lambda_n^{2-q}s_0(p),$$

*where $C_4 \le (1 + 2^{1-q} + 3^{1-q})(4\|\mathbf{\Omega}\|_{L_1})^{1-q}$ and $C_5 \le 4\|\mathbf{\Omega}\|_{L_1}C_4$.*

In fact, Theorem 1 from Lu & Feng (2025) and this Lemma are not fundamentally the same. However, our algorithm is actually not sensitive to bandwidth choice. When the initial value is well-chosen, $pn^{-1}\sum_{i=1}^n U\{\hat{\varepsilon}_i^{(k)}\}U\{\hat{\varepsilon}_i^{(k)}\}^\top \approx \mathbf{I}$, so a very small bandwidth is also acceptable. In this case, it can be regarded as projecting $n^{-1}\sum_{i=1}^n U\{\hat{\varepsilon}_i^{(k)}\}U\{\hat{\varepsilon}_i^{(k)}\}^\top$ onto the subspace where its true value resides.

**Lemma 5.** *Define a random matrix $\hat{\mathbf{Q}} = n^{-1}\sum_{i=1}^n \hat{r}_i^{-1}\hat{U}_i\hat{U}_i^\top \in \mathbb{R}^{p \times p}$, and let $\hat{\mathbf{Q}}_{jl}$ denote its $(j, l)$-th element. Assume $\lambda_n^{1-q}s_0(p)(\log p)^{1/2} = o(1)$, and satisfy Assumptions 1-4. Then we have*

$$|\hat{\mathbf{Q}}_{jl}| \lesssim p^{-3/2}\mathbb{I}(j = l) + O_p\left\{n^{-1/2}p^{-3/2} + \lambda_n^{1-q}s_0(p)p^{-3/2}\right\}.$$

*Here, the symbol $\lesssim$ has been defined in the **Notations** at the end of Section 1.*

*Proof.* Denote $\hat{\mathbf{I}} = \hat{\mathbf{\Omega}}^{1/2}\mathbf{\Sigma}^{1/2}$. Set $\hat{\mathbf{I}}_i^\top$ and $\mathbf{\Omega}_i^\top$ be the $i$th row of $\hat{\mathbf{I}}$ and $\mathbf{\Omega}$ respectively.

$$\hat{\mathbf{Q}}_{jl} = \frac{1}{n}\sum_{i=1}^n \hat{r}_i^{-1}\hat{U}_{ij}\hat{U}_{il}$$

$$= \frac{1}{n}\sum_{i=1}^n \hat{r}_i^{-3}\hat{\mathbf{I}}_j^\top \boldsymbol{\varepsilon}_i\hat{\mathbf{I}}_l^\top \boldsymbol{\varepsilon}_i$$

$$= \frac{1}{n}\sum_{i=1}^n \|\hat{\mathbf{I}}\boldsymbol{\varepsilon}_i\|^{-3}(\hat{\mathbf{I}}_j^T\boldsymbol{\varepsilon}_i)(\hat{\mathbf{I}}_l^T\boldsymbol{\varepsilon}_i)$$

$$= A_1 + A_2 + A_3,$$

where $A_1$, $A_2$ and $A_3$ are defined as follows

$$A_1 = \frac{1}{n}\sum_{i=1}^n \left(\|\hat{\mathbf{I}}\boldsymbol{\varepsilon}_i\|^{-3} - \|\mathbf{I}\boldsymbol{\varepsilon}_i\|^{-3}\right)(\hat{\mathbf{I}}_j^T\boldsymbol{\varepsilon}_i)(\hat{\mathbf{I}}_l^T\boldsymbol{\varepsilon}_i);$$

$$A_2 = \frac{1}{n}\sum_{i=1}^n \left(\|\mathbf{I}\boldsymbol{\varepsilon}_i\|^{-3} - \zeta_3\right)(\hat{\mathbf{I}}_j^T\boldsymbol{\varepsilon}_i)(\hat{\mathbf{I}}_l^T\boldsymbol{\varepsilon}_i);$$

$$A_3 = \frac{1}{n}\sum_{i=1}^n \zeta_3(\hat{\mathbf{I}}_j^T\boldsymbol{\varepsilon}_i)(\hat{\mathbf{I}}_l^T\boldsymbol{\varepsilon}_i).$$

Given Lemma 4 and under Assumption 2, we obtain that

$$\|\hat{\mathbf{I}}\boldsymbol{\varepsilon}_i\|^2 = \boldsymbol{\varepsilon}_i^T\mathbf{\Sigma}^{1/2}(\hat{\mathbf{\Omega}} - \mathbf{\Omega})\mathbf{\Sigma}^{1/2}\boldsymbol{\varepsilon}_i + r_i^2$$

$$\le r_i^2 + (\boldsymbol{\varepsilon}_i^T\mathbf{\Sigma}\boldsymbol{\varepsilon}_i)\|\hat{\mathbf{\Omega}} - \mathbf{\Omega}\|_{op}$$

$$\le r_i^2 + \eta^{-1}r_i^2\|\hat{\mathbf{\Omega}} - \mathbf{\Omega}\|_{op}$$

$$\doteq r_i^2(1 + H),$$

where $H = \eta^{-1}\|\hat{\boldsymbol{\Omega}} - \boldsymbol{\Omega}\|_{op} = O_p\{\lambda_n^{1-q}s_0(p)\}$. Therefore, for any integer $k$,

$$
\begin{aligned}
\|\hat{\mathbf{I}}\varepsilon_i\|^k &= \{\varepsilon_i^T \boldsymbol{\Sigma}^{1/2}(\hat{\boldsymbol{\Omega}} - \boldsymbol{\Omega})\boldsymbol{\Sigma}^{1/2}\varepsilon_i + r_i^2\}^{k/2} \\
&\leq r_i^k (1+H)^{k/2} \\
&:= r_i^k (1 + H_k),
\end{aligned}
\tag{11}
$$

where $H_k = (1+H)^{k/2} - 1 = O_p\{\lambda_n^{1-q}s_0(p)\}$.

Similar to the proof of Lemma A3 in Cheng et al. (2023), we have

$$
\begin{aligned}
\mathbb{E}(A_1) &= \mathbb{E}\left\{ \frac{1}{n}\sum_{i=1}^n \left( \|\hat{\mathbf{I}}\varepsilon_i\|^{-3} - \|\mathbf{I}\varepsilon_i\|^{-3} \right)(\hat{\mathbf{I}}_j^\top \varepsilon_i)(\hat{\mathbf{I}}_l^\top \varepsilon_i) \right\} \\
&= \mathbb{E}\left\{ \frac{1}{n}\sum_{i=1}^n \left( \|\varepsilon_i\|^{-3} H_{-3} \right)(\hat{\mathbf{I}}_j^\top \varepsilon_i)(\hat{\mathbf{I}}_l^\top \varepsilon_i) \right\} \\
&= \mathbb{E}\left\{ (A_2 + A_3)H_{-3} \right\}.
\end{aligned}
$$

Firstly, notice that,

$$
\begin{aligned}
\hat{\mathbf{I}}_j^\top \varepsilon_i &= (\hat{\mathbf{I}} - \mathbf{I})_j^\top \varepsilon_i + \varepsilon_{ij} \\
&= (\hat{\boldsymbol{\Omega}}^{1/2} - \boldsymbol{\Omega}^{1/2})_j^\top \boldsymbol{\Sigma}^{1/2}\varepsilon_i + \varepsilon_{ij},
\end{aligned}
$$

thus,

$$
\begin{aligned}
\hat{\mathbf{I}}_j^\top \varepsilon_i - \varepsilon_{ij} &= \frac{1}{2}\{\boldsymbol{\Omega}^{-1/2}(\hat{\boldsymbol{\Omega}} - \boldsymbol{\Omega})\}_j^\top \boldsymbol{\Sigma}^{1/2}\varepsilon_i + o_p\left[ \{\boldsymbol{\Omega}^{-1/2}(\hat{\boldsymbol{\Omega}} - \boldsymbol{\Omega})\}_j^\top \boldsymbol{\Sigma}^{1/2}\varepsilon_i \right] \\
&\lesssim \|\boldsymbol{\Omega}^{-1/2}\|_{L_1}\|\hat{\boldsymbol{\Omega}} - \boldsymbol{\Omega}\|_{L_1}\|\boldsymbol{\Sigma}^{1/2}\|_{L_1}\|r_i\boldsymbol{\Sigma}^{1/2}\boldsymbol{U}_i\|_\infty \\
&= O_p\{\lambda_n^{1-q}s_0(p)(\log p)^{1/2}\} = o_p(1).
\end{aligned}
$$

In the above equation, the second to last equation from the following facts: (1) Since $\boldsymbol{\Sigma}$ is a positive define symmetric matrix, and under the Assumption 1, we have $\|\boldsymbol{\Omega}^{-1/2}\|_{L_1} \leq \{\lambda_{\max}(\boldsymbol{\Sigma})\|\boldsymbol{\Sigma}\|_{L_1}\}^{1/2} = O(1)$. (2) Furthermore, according to the second formula of Lemma 4, $\|\hat{\boldsymbol{\Omega}} - \boldsymbol{\Omega}\|_{L_1} = O_p\{\lambda_n^{1-q}s_0(p)\}$. (3) As for $\boldsymbol{U}_i$ is uniformly distributed on a p-dimensional unit sphere, $\|\boldsymbol{U}_i\|_\infty = O_p(\sqrt{\log p/p})$ and $r_i = O_p(\sqrt{p})$, we have $\|r_i\boldsymbol{\Sigma}^{1/2}\boldsymbol{U}_i\|_\infty = O_p\{(\log p)^{1/2}\}$.

Next, we analyze $A_2$ and $A_3$. Since $\mathbb{E}(r_i^2) = p$,

$$
\begin{aligned}
A_2 &= \frac{1}{n}\sum_{i=1}^n \left( \|\mathbf{I}\varepsilon_i\|^{-3} - \zeta_3 \right)\{\varepsilon_{ij} + o_p(1)\}\{\varepsilon_{il} + o_p(1)\} \\
&= \frac{1}{n}\sum_{i=1}^n (r_i^{-1} - \zeta_3 r_i^2)U_{ij}U_{il}\mathbb{I}(j = l) + o_p(1) \\
&= \zeta_1 p^{-1}\mathbb{I}(j = l) + O_p(n^{-1/2}p^{-3/2}) \lesssim p^{-3/2}\mathbb{I}(j = l) + O_p(n^{-1/2}p^{-3/2}).
\end{aligned}
$$

and

$$
\begin{aligned}
A_3 &= \frac{1}{n}\sum_{i=1}^n \zeta_3\{\varepsilon_{ij} + o_p(1)\}\{\varepsilon_{il} + o_p(1)\} \\
&\lesssim p^{-3/2}\mathbb{I}(j = l) + O_p(n^{-1/2}p^{-3/2}).
\end{aligned}
$$

It follows that,

$$
|\hat{\mathbf{Q}}_{jl}| \lesssim \left\{ p^{-3/2}\mathbb{I}(j = l) + O_p(n^{-1/2}p^{-3/2}) \right\}[1 + O_p\{\lambda_n^{1-q}s_0(p)\}].
$$

Thus,

$$
|\hat{\mathbf{Q}}_{jl}| \lesssim p^{-3/2}\mathbb{I}(j = l) + O_p\left\{ n^{-1/2}p^{-3/2} + \lambda_n^{1-q}s_0(p)p^{-3/2} \right\}.
$$

$\square$

**Lemma 6.** *Suppose the Assumptions in Lemma 4 hold, then* $\hat{\zeta}_1 \xrightarrow{p} \zeta_1$ *as* $(n,p) \to \infty$, *where* $\hat{\zeta}_1 = n^{-1}\sum_{i=1}^n \|\hat{\Omega}^{1/2}(X_1 - \hat{\mu}_1)\|^{-1}$.

*Proof.* Denote $\hat{\theta} = \hat{\mu} - \mu$.

$$\|\hat{\Omega}^{1/2}(X_i - \hat{\mu})\| = \|\Omega^{1/2}(X_i - \mu)\|(1 + r_i^{-2}\|(\hat{\Omega}^{1/2} - \Omega^{1/2})(X_i - \mu)\|^2$$
$$+ r_i^{-2}\|\hat{\Omega}^{1/2}\hat{\theta}\|^2 + 2r_i^{-2}U_i^\top(\hat{\Omega}^{1/2} - \Omega^{1/2})\Omega^{-1/2}U_i)$$
$$- 2r_i^{-1}U_i^\top\hat{\Omega}^{1/2}\hat{\theta} - 2r_i^{-1}U_i\Omega^{-1/2}(\hat{\Omega}^{1/2} - \Omega^{1/2})\hat{\Omega}^{1/2}\hat{\theta})^{1/2}.$$

By combining the third expression in Lemma 4, the Taylor expansion and Markov's inequality, we obtain $r_i^{-2}\|(\hat{\Omega}^{1/2} - \Omega^{1/2})(X_i - \mu)\|^2 = O_p\{\lambda_n^{2-q}s_0(p)\} = o_p(1)$. Based on Lemma 1 and under the Assumption 1, we have $r_i^{-2}\|\hat{\Omega}^{1/2}\hat{\theta}\|^2 = O_p(n^{-1}) = o_p(1)$. Similarly, by the Cauchy-Schwarz inequality, the other parts are also $o_p(1)$. So,

$$n^{-1}\sum_{i=1}^n \left\|\hat{\Omega}^{1/2}(X_i - \hat{\mu})\right\|^{-1} = \left\{n^{-1}\sum_{i=1}^n \left\|\Omega^{1/2}(X_i - \mu)\right\|^{-1}\right\}\{1 + o_p(1)\}.$$

Obviously, $\mathbb{E}\left(n^{-1}\sum_{i=1}^n r_i^{-1}\right) = \zeta_1$ and $\mathrm{Var}\left(n^{-1}\zeta_1^{-1}\sum_{i=1}^n r_i^{-1}\right) = O\left(n^{-1}\right)$. Finally, the proof is completed. $\quad\square$

**Lemma 7.** *Suppose the Assumptions in Lemma 6 hold with $s_0(p) \asymp p^{1-\delta}$ for some positive constant $\delta \le 1/2$ Then, if $\log p = o(n^{1/3})$,*

$$(i) \left\|n^{-1}\sum_{i=1}^n \zeta_1^{-1}\hat{U}_i\right\|_\infty = O_p\left\{n^{-1/2}\log^{1/2}(np)\right\},$$

$$(ii) \left\|\zeta_1^{-1}n^{-1}\sum_{i=1}^n \delta_{1,i}\hat{U}_i\right\|_\infty = O_p(n^{-1}). \tag{12}$$

*where $\delta_{1,i}$ is defined in the proof in Lemma 1.*

*Proof.* From the proof of Lemma 5, we can see that $\hat{I}_j^\top\varepsilon_i - \varepsilon_{ij} = O_p\{\lambda_n^{1-q}s_0(p)(\log p)^{1/2}\}$. Moreover, for any integer $k$, we have $\hat{r}_i^k \le r_i^k(1 + H_k)$, where $H_k = O_p\{\lambda_n^{1-q}s_0(p)\}$. Recall that $\hat{U}_i = U\{\hat{\Omega}^{1/2}(X_i - \mu)\}$, since $r_i^{-1} = O_p(p^{-1/2})$, then for any $j \in \{1, 2, \cdots, p\}$,

$$\hat{U}_{ij} = \hat{r}_i^{-1}\hat{I}_j^\top\varepsilon_i \le r_i^{-1}(1 + H_{-1})\varepsilon_{ij} + r_i^{-1}(1 + H_{-1})(\hat{I}_j^\top\varepsilon_i - \varepsilon_{ij})$$
$$= (1 + H_{-1})U_{ij} + o_p\{(1 + H_{-1})U_{ij}\}.$$

Therefore, we obtain that $\hat{U}_i \le U_i(1 + H_{-1})$ for $i = 1, 2, \ldots, n$ with the assumption $\lambda_n^{1-q}s_0(p)(\log p)^{1/2} = o(1)$. According to the Lemma A4 in Cheng et al. (2023), we have $\left\|n^{-1/2}\sum_{i=1}^n \zeta_1^{-1}U_i\right\|_\infty = O_p\{\log^{1/2}(np)\}$ and $\left\|n^{-1}\sum_{i=1}^n (\zeta_1^{-1}U_i)^2\right\|_\infty = O_p(1)$ with $\log p = o(n^{1/3})$. Therefore, we have

$$\left\|n^{-1}\sum_{i=1}^n \zeta_1^{-1}\hat{U}_i\right\|_\infty = \left\|n^{-1}\sum_{i=1}^n \zeta_1^{-1}(1 + H_{-1})U_i\right\|_\infty$$

$$\le |1 + H_{-1}| \cdot \left\|n^{-1}\sum_{i=1}^n \zeta_1^{-1}U_i\right\|_\infty = O_p\left\{n^{-1/2}\log^{1/2}(np)\right\}.$$

Similarly

$$\left\|\zeta_1^{-1}n^{-1}\sum_{i=1}^n \delta_{1,i}\hat{U}_i\right\|_\infty \le |1 + H_{-1}| \cdot \left\|\zeta_1^{-1}n^{-1}\sum_{i=1}^n \delta_{1,i}U_i\right\|_\infty$$

$$= O_p\{n^{-1}(1 + n^{-1/2}\log^{1/2} p)\} = O_p(n^{-1}).$$

$\quad\square$

The proof of Lemma 8 can be found in Appendix A of Chernozhukov et al. (2017).

**Lemma 8** (Nazarov's inequality). *Let $\boldsymbol{Y}_0 = (Y_{0,1}, Y_{0,2}, \cdots, Y_{0,p})^\top$ be a centered Gaussian random vector in $\mathbb{R}^p$ and $\mathbb{E}(Y_{0,j}^2) \geq b$ for all $j = 1, 2, \cdots, p$ and some constant $b > 0$, then for every $y \in \mathbb{R}^p$ and $a > 0$,*

$$\mathbb{P}(\boldsymbol{Y}_0 \leq y + a) - \mathbb{P}(\boldsymbol{Y}_0 \leq y) \lesssim a \log^{1/2}(p).$$

We restate Lemma S9 in Feng et al. (2024).

**Lemma 9.** *For each $d \geq 1$, we have*

$$\lim_{p \to \infty} H(d, p) \leq \frac{1}{d!} \pi^{-d/2} e^{-dy/2},$$

*where $H(d, p) \doteq \sum_{1 \leq i_1 < \cdots < i_d \leq p} \mathbb{P}(B_{i_1} \cdots B_{i_d})$, $B_{i_d} = \{|y_{i_d}| \geq \sqrt{2 \log p - \log \log p + y}\}$, $\boldsymbol{Y} = (y_1, \cdots, y_p)^\top \sim \mathcal{N}(\mathbf{0}, \mathbf{R})$.*

**Lemma 10.** *Let $\boldsymbol{u} \in \mathbb{R}^p$ be a random vector uniformly distributed on the unit sphere $\mathbb{S}^{p-1}$. $\mathbf{A} \in \mathbb{R}^{p \times p}$ is a non-random matrix. Then we have $\mathbb{E}(\boldsymbol{u}^\top \mathbf{A} \boldsymbol{u}) = p^{-1} \operatorname{tr}(\mathbf{A})$ and $\operatorname{Var}(\boldsymbol{u}^\top \mathbf{A} \boldsymbol{u}) \asymp p^{-2} \|\mathbf{A}\|_F^2$ as $p \to \infty$.*

*Proof.* Since $\mathbb{E}(\boldsymbol{u}\boldsymbol{u}^\top) = p^{-1}\mathbf{I}_p$, then $\mathbb{E}(\boldsymbol{u}^\top \mathbf{A} \boldsymbol{u}) = \operatorname{tr}\{\mathbf{A}\mathbb{E}(\boldsymbol{u}\boldsymbol{u}^\top)\} = p^{-1}\operatorname{tr}(\mathbf{A})$. Let $\mathbf{A} = (a_{ij})_{i,j=1}^p$, $\boldsymbol{u} = (u_1, \ldots, u_p)^\top$,

$$
\begin{aligned}
\mathbb{E}(\boldsymbol{u}^\top \mathbf{A} \boldsymbol{u})^2 =& \mathbb{E}\left(\sum_{i=1}^p a_{ii} u_i^2 + \sum_{1 \leq i \neq j \leq p} a_{ij} u_i u_j\right)^2 \\
=& \mathbb{E}\left\{\sum_{i=1}^p a_{ii}^2 u_i^4 + \sum_{1 \leq i \neq j \leq p} (a_{ij}^2 + a_{ii} a_{jj}) u_i^2 u_j^2\right\} \\
=& \frac{3}{p(p+2)} \sum_{i=1}^p a_{ii}^2 + \frac{1}{p(p+2)} \sum_{1 \leq i \neq j \leq p} a_{ij}^2 + a_{ii} a_{jj},
\end{aligned}
$$

where the last equality because that $(u_1^2, \ldots, u_p^2)^\top$ follow a Dirichlet distribution $D_p(1/2, \ldots, 1/2)$(Oja, 2010). As a consequence, we have $\mathbb{E}(u_i^4) = 3/\{p(p+2)\}$ and $\mathbb{E}(u_i^2 u_j^2) = 1/\{p(p+2)\}$ for any $i \neq j$. Combining the two results above and after some straightforward calculations, we obtain $\operatorname{Var}(\boldsymbol{u}^\top \mathbf{A} \boldsymbol{u}) \asymp p^{-2} \|\mathbf{A}\|_F^2$. $\square$

**Lemma 11.** *Under Assumption 1, for $i = 1, 2$, we have*

$$\frac{\widehat{\operatorname{tr}(\boldsymbol{\Xi}_i)}}{\operatorname{tr}(\boldsymbol{\Xi}_i)} - 1 = O_p(n^{-1/2}).$$

*Proof.* Recall that $\widehat{\operatorname{tr}(\boldsymbol{\Xi}_i)}$ is defined as in Section A. Notice that, for $i = 1, 2$,

$$
\begin{aligned}
\widehat{\operatorname{tr}(\boldsymbol{\Xi}_i)} =& \frac{1}{n_i - 1} \sum_{j=1}^{n_i} \boldsymbol{X}_{ij}^\top \boldsymbol{X}_{ij} - \frac{n_i}{n_i - 1} \bar{\boldsymbol{X}}_i^\top \bar{\boldsymbol{X}}_i \\
=& \frac{\sum_{j=1}^{n_i} \boldsymbol{X}_{ij}^\top \boldsymbol{X}_{ij} - \sum_{j,k} \boldsymbol{X}_{ij}^\top \boldsymbol{X}_{ik}}{n_i(n_i - 1)} \\
=& \frac{\sum_{j \neq k \neq l} -\boldsymbol{X}_{ij}^\top \boldsymbol{X}_{ik} + \boldsymbol{X}_{ik}^\top \boldsymbol{X}_{ik}}{n_i(n_i - 1)(n_i - 2)} \\
=& \frac{\sum_{j \neq k \neq l} \boldsymbol{X}_{ij}^\top \boldsymbol{X}_{il} - \boldsymbol{X}_{ik}^\top \boldsymbol{X}_{il} - \boldsymbol{X}_{ij}^\top \boldsymbol{X}_{ik} + \boldsymbol{X}_{ik}^\top \boldsymbol{X}_{ik}}{n_i(n_i - 1)(n_i - 2)} \\
=& \frac{\sum_{j \neq k \neq l} (\boldsymbol{X}_{ij} - \boldsymbol{X}_{ik})^\top (\boldsymbol{X}_{il} - \boldsymbol{X}_{ik})}{n_i(n_i - 1)(n_i - 2)},
\end{aligned}
$$

which implies that our estimate of $\operatorname{tr}(\boldsymbol{\Xi}_i)$ is the same as that of Shen & Feng (2025). Thus, we complete the proof according to Lemma 8.4 of Shen & Feng (2025). $\square$

We next restate Lemma 8.9 from Shen & Feng (2025).

**Lemma 12.** *For positive matrix* $\mathbf{X}$, $\mathbf{Y}$,

$$\log |\mathbf{X}| \le \log |\mathbf{Y}| + \text{tr}\{\mathbf{Y}^{-1}(\mathbf{X} - \mathbf{Y})\}.$$

### E.2 PROOF OF MAIN LEMMAS

***Proof of Lemma 1.*** As $\boldsymbol{\mu}$ is a location parameter, we assume $\boldsymbol{\mu} = 0$ without loss of generality. Note that given $\hat{\boldsymbol{\Omega}}$, the estimator $\hat{\boldsymbol{\mu}}$ satisfies

$$\sum_{i=1}^n U\{\hat{\boldsymbol{\Omega}}^{1/2}(\mathbf{X}_i - \hat{\boldsymbol{\mu}})\} = 0.$$

Therefore, the estimator $\hat{\boldsymbol{\mu}}$ is defined as the minimizer of the following objective function:

$$L(\boldsymbol{\theta}) = \sum_{i=1}^n \left\| \hat{\boldsymbol{\Omega}}^{1/2}(\mathbf{X}_i - \boldsymbol{\theta}) \right\|. \tag{13}$$

Our goal is find $b_{n,p}$ such that $\|\hat{\boldsymbol{\mu}}\| = O_p(b_{n,p})$. The existence of a $b_{n,p}^{-1}$-consistent local minimizer is implied by the fact that for an arbitrarily small $\varepsilon > 0$, there exist a sufficiently large constant $C$, which does no depend on $n$ or $p$, such that

$$\liminf_n \mathbb{P}\left\{ \inf_{\boldsymbol{u} \in \mathbb{R}^p, \|\boldsymbol{u}\| = C} L(b_{n,p}\boldsymbol{u}) > L(\mathbf{0}) \right\} > 1 - \varepsilon. \tag{14}$$

Firstly, we prove Equation (14) holds when $b_{n,p} = p^{1/2}n^{-1/2}$. Consider the expansion of $\|\hat{\boldsymbol{\Omega}}^{1/2}(\mathbf{X}_i - b_{n,p}\boldsymbol{u})\|$:

$$\|\hat{\boldsymbol{\Omega}}^{1/2}(\mathbf{X}_i - b_{n,p}\boldsymbol{u})\| = \|\hat{\boldsymbol{\Omega}}^{1/2}\mathbf{X}_i\| \left(1 - 2b_{n,p}\hat{r}_i^{-1}\boldsymbol{u}^\top\hat{\boldsymbol{\Omega}}^{1/2}\hat{U}_i + b_{n,p}^2\hat{r}_i^{-2}\boldsymbol{u}^\top\hat{\boldsymbol{\Omega}}\boldsymbol{u}\right)^{1/2}.$$

Note that $b_{n,p}\hat{r}_i^{-1}\boldsymbol{u}^\top\hat{\boldsymbol{\Omega}}^{1/2}\hat{U}_i = O_p(n^{-1/2})$ and $b_{n,p}^2\hat{r}^{-2}\boldsymbol{u}^\top\hat{\boldsymbol{\Omega}}\boldsymbol{u} = O_p(n^{-1})$. These orders follow from the following argument. Since we already know that $\hat{r}_i^k \le r_i^k(1 + H_k)$ and $\hat{U}_i \le U_i(1 + H_{-1})$ with $H_k = O_p\{\lambda_n^{1-q}s_0(p)\}$ for any integer $k$, thus,

$$b_{n,p}\hat{r}_i^{-1}\boldsymbol{u}^\top\hat{\boldsymbol{\Omega}}^{1/2}\hat{U}_i \le b_{n,p}(1 + H_k)^2 r_i^{-1}\boldsymbol{u}^\top\boldsymbol{\Omega}^{1/2}U_i + b_{n,p}(1 + H_k)^2 r_i^{-1}\boldsymbol{u}^\top(\hat{\boldsymbol{\Omega}}^{1/2} - \boldsymbol{\Omega}^{1/2})U_i.$$

For the first term, by independence between $r_i$ and $U_i$, we have $\mathbb{E}\{(r_i^{-1}\boldsymbol{u}^\top\boldsymbol{\Omega}^{1/2}U_i)^2\} = \mathbb{E}(r_i^{-2})\mathbb{E}\{(\boldsymbol{u}^\top\boldsymbol{\Omega}^{1/2}U_i)^2\} = \zeta_2 p^{-1} \text{tr}(\boldsymbol{\Omega})$, which implies that $\hat{r}_i^{-1}\boldsymbol{u}^\top\hat{\boldsymbol{\Omega}}^{1/2}\hat{U}_i = O_p(p^{-1/2})$. Similarly, for the second term, applying Taylor expansion and Lemma 4 yields:

$$\mathbb{E}[\{r_i^{-1}\boldsymbol{u}^\top(\hat{\boldsymbol{\Omega}}^{1/2} - \boldsymbol{\Omega}^{1/2})U_i\}^2] \lesssim p^{-2}\|(\hat{\boldsymbol{\Omega}} - \boldsymbol{\Omega})^2\|_{op} \le p^{-2}\|\hat{\boldsymbol{\Omega}} - \boldsymbol{\Omega}\|_{op}^2 \lesssim p^{-2}\lambda_n^{2-2q}s_0^2(p),$$

which implies that $r_i^{-1}\boldsymbol{u}^\top(\hat{\boldsymbol{\Omega}}^{1/2} - \boldsymbol{\Omega}^{1/2})U_i = O_p\{p^{-1}\lambda_n^{1-q}s_0(p)\}$. Hence

$$b_{n,p}\hat{r}_i^{-1}\boldsymbol{u}^\top\hat{\boldsymbol{\Omega}}^{1/2}\hat{U}_i = O_p\{b_{n,p}p^{-1/2} + b_{n,p}p^{-1}\lambda_n^{1-q}s_0(p)\}$$
$$= O_p(n^{-1/2}).$$

As the same way, we have $b_{n,p}^2\hat{r}^{-2}\boldsymbol{u}^\top\hat{\boldsymbol{\Omega}}\boldsymbol{u} = O_p(n^{-1})$. Then we have

$$\|\hat{\boldsymbol{\Omega}}^{1/2}(\mathbf{X}_i - b_{n,p}\boldsymbol{u})\| = \|\hat{\boldsymbol{\Omega}}^{1/2}\mathbf{X}_i\| - b_{n,p}\boldsymbol{u}^\top\hat{\boldsymbol{\Omega}}^{1/2}\hat{U}_i$$
$$+ \frac{1}{2}b_{n,p}^2\hat{r}_i^{-1}\boldsymbol{u}\hat{\boldsymbol{\Omega}}^{1/2}\left(\mathbf{I}_p - \hat{U}_i\hat{U}_i^\top\right)\hat{\boldsymbol{\Omega}}^{1/2}\boldsymbol{u} + O_p(p^{1/2}n^{-3/2}).$$

So, it can be easily seen

$$p^{-1/2}\{L(b_{n,p}\boldsymbol{u}) - L(\mathbf{0})\}$$
$$= -n^{-1/2}\boldsymbol{u}^\top\hat{\boldsymbol{\Omega}}^{1/2}\sum_{i=1}^n \hat{U}_i$$
$$+ 2^{-1}p^{1/2}\boldsymbol{u}\hat{\boldsymbol{\Omega}}^{1/2}\left\{n^{-1}\sum_{i=1}^n\left(\hat{r}_i^{-1}\mathbf{I}_p - \hat{r}_i^{-1}\hat{U}_i\hat{U}_i^\top\right)\right\}\hat{\boldsymbol{\Omega}}^{1/2}\boldsymbol{u} + O_p(n^{-1/2}). \tag{15}$$

Notice that $\mathbb{E}\left(\|n^{-1/2}\sum_{i=1}^n\hat{U}_i\|^2\right) = O(1)$ and $\mathrm{Var}\left(\|n^{-1/2}\sum_{i=1}^n\hat{U}_i\|^2\right) = O(1)$. Accordingly

$$\left| -n^{-1/2}\boldsymbol{u}^\top\hat{\boldsymbol{\Omega}}^{1/2}\sum_{i=1}^n\hat{U}_i \right| \le \left\| \hat{\boldsymbol{\Omega}}^{1/2}\boldsymbol{u} \right\| \left\| n^{-1/2}\sum_{i=1}^n\hat{U}_i \right\| = O_p(1).$$

Recall the definition $\hat{\mathbf{Q}} = n^{-1}\sum_{i=1}^n \hat{r}_i^{-1}\hat{U}_i\hat{U}_i^\top$ in Lemma 5. After some tedious calculation, we can obtain that $\mathbb{E}\{\mathrm{tr}(\hat{\mathbf{Q}}^2)\} = O\{p^{-2} + n^{-1}p^{-1} + \lambda_n^{2-2q}s_0^2(p)p^{-1}\}$. Then $\mathbb{E}(\boldsymbol{u}^\top\hat{\boldsymbol{\Omega}}^{1/2}\hat{\mathbf{Q}}\hat{\boldsymbol{\Omega}}^{1/2}\boldsymbol{u})^2 \le \mathbb{E}\left\{(\boldsymbol{u}^\top\hat{\boldsymbol{\Omega}}\boldsymbol{u})^2\mathrm{tr}(\hat{\mathbf{Q}}^2)\right\} = O\{p^{-2}+n^{-1}p^{-1}+\lambda_n^{2-2q}s_0^2(p)p^{-1}\}$, which leads to $\boldsymbol{u}^\top\hat{\boldsymbol{\Omega}}^{1/2}\hat{\mathbf{Q}}\hat{\boldsymbol{\Omega}}^{1/2}\boldsymbol{u} = O_p\{p^{-1}+n^{-1/2}p^{-1/2}+\lambda_n^{1-q}s_0(p)p^{-1/2}\}$. Thus we have

$$p^{1/2}\boldsymbol{u}\hat{\boldsymbol{\Omega}}^{1/2}\left\{ \frac{1}{n}\sum_{i=1}^n \left( \hat{r}_i^{-1}\mathbf{I}_p - \hat{r}_i^{-1}\hat{U}_i\hat{U}_i^\top \right) \right\}\hat{\boldsymbol{\Omega}}^{1/2}\boldsymbol{u}$$

$$=p^{1/2}n^{-1}\sum_{i=1}^n \hat{r}_i^{-1}\boldsymbol{u}\hat{\boldsymbol{\Omega}}\boldsymbol{u} + o_p(1),$$

where we use the fact that $n^{-1}\sum_{i=1}^n \hat{r}_i^{-1} = \zeta_1 + O_p\{n^{-1/2}p^{-1/2} + \lambda_n^{1-q}s_0(p)p^{-1/2}\}$. By choosing a sufficient large $C$, the second term in (15) dominates the first term uniformly in $\|\boldsymbol{u}\| = C$. Hence, (15) holds and accordingly $\hat{\boldsymbol{\mu}} = O_p(b_{n,p})$. The estimator $\hat{\boldsymbol{\mu}}$ satisfies $\sum_{i=1}^n U\{\hat{\boldsymbol{\Omega}}^{1/2}(\boldsymbol{X}_i-\hat{\boldsymbol{\mu}})\} = 0$, which is is equivalent to

$$n^{-1}\sum_{i=1}^n(\hat{U}_i - \hat{r}_i^{-1}\hat{\boldsymbol{\Omega}}^{1/2}\hat{\boldsymbol{\mu}})(1 - 2\hat{r}_i^{-1}\hat{U}_i^\top\hat{\boldsymbol{\Omega}}^{1/2}\hat{\boldsymbol{\mu}} + \hat{r}_i^{-2}\hat{\boldsymbol{\mu}}^\top\hat{\boldsymbol{\Omega}}\hat{\boldsymbol{\mu}})^{-1/2} = 0.$$

By the first-Taylor expansion, the above equation can be rewritten as:

$$n^{-1}\sum_{i=1}^n \left( \hat{U}_i - \hat{r}_i^{-1}\hat{\boldsymbol{\Omega}}^{1/2}\hat{\boldsymbol{\mu}} \right)\left( 1 + \hat{r}_i^{-1}\hat{U}_i^\top\hat{\boldsymbol{\Omega}}^{1/2}\hat{\boldsymbol{\mu}} - 2^{-1}\hat{r}_i^{-2}\left\| \hat{\boldsymbol{\Omega}}^{1/2}\hat{\boldsymbol{\mu}} \right\|^2 + \delta_{1,i} \right) = 0,$$

where $\delta_{1,i} = O_p\{(\hat{r}_i^{-1}\hat{U}_i^\top\hat{\boldsymbol{\Omega}}^{1/2}\hat{\boldsymbol{\mu}} - 2^{-1}\hat{r}_i^{-2}\|\hat{\boldsymbol{\Omega}}^{1/2}\hat{\boldsymbol{\mu}}\|^2)^2\} = O_p(n^{-1})$, which implies

$$\frac{1}{n}\sum_{i=1}^n(1 - 2^{-1}\hat{r}_i^{-2}\hat{\boldsymbol{\mu}}^\top\hat{\boldsymbol{\Omega}}\hat{\boldsymbol{\mu}} + \delta_{1,i})\hat{U}_i + \frac{1}{n}\sum_{i=1}^n \hat{r}_i^{-1}(\hat{U}_i^\top\hat{\boldsymbol{\Omega}}^{1/2}\hat{\boldsymbol{\mu}})\hat{U}_i$$

$$=\frac{1}{n}\sum_{i=1}^n(1 + \delta_{1,i} + \delta_{2,i})\hat{r}_i^{-1}\hat{\boldsymbol{\Omega}}^{1/2}\hat{\boldsymbol{\mu}}, \tag{16}$$

where $\delta_{2,i} = O_p(\hat{r}_i^{-1}\hat{U}_i^\top\hat{\boldsymbol{\Omega}}^{1/2}\hat{\boldsymbol{\mu}} - 2^{-1}\hat{r}_i^{-2}\|\hat{\boldsymbol{\Omega}}^{1/2}\hat{\boldsymbol{\mu}}\|^2) = O_p(\delta_{1,i}^{1/2})$. By Assumption 1 and Markov inequality, we have that: $\max r_i^{-2} = O_p(p^{-1}n^{1/2})$, $\max\delta_{1,i} = O_p\left( \|\hat{\boldsymbol{\Omega}}^{1/2}\hat{\boldsymbol{\mu}}\|^2 \max\hat{r}_i^{-2} \right) = O_p(n^{-1/2})$ and $\max\delta_{2,i} = O_p(n^{-1/4})$. Considering the second term in Equation (16),

$$\frac{1}{n}\sum_{i=1}^n \hat{r}_i^{-1}(\hat{U}_i^\top\hat{\boldsymbol{\Omega}}^{1/2}\hat{\boldsymbol{\mu}})\hat{U}_i = \frac{1}{n}\sum_{i=1}^n \hat{r}_i^{-1}(\hat{U}_i\hat{U}_i^\top\hat{\boldsymbol{\Omega}}^{1/2})\hat{\boldsymbol{\mu}} = \hat{\mathbf{Q}}\hat{\boldsymbol{\Omega}}^{1/2}\hat{\boldsymbol{\mu}}.$$

From Lemma 5 we acquire

$$|\hat{\mathbf{Q}}_{jl}| \lesssim p^{-3/2}\mathbb{I}(j = l) + O_p\left\{ n^{-1/2}p^{-3/2} + \lambda_n^{1-q}s_0(p)p^{-3/2} \right\},$$

and this implies that,

$$\|\hat{\mathbf{Q}}\hat{\boldsymbol{\Omega}}^{1/2}\hat{\boldsymbol{\mu}}\|_\infty$$
$$\le\|\hat{\mathbf{Q}}\|_1\|\hat{\boldsymbol{\Omega}}^{1/2}\hat{\boldsymbol{\mu}}\|_\infty \tag{17}$$
$$=O_p\left\{ n^{-1/2}p^{-1/2} + \lambda_n^{1-q}s_0(p)p^{-1/2} \right\}\|\hat{\boldsymbol{\Omega}}^{1/2}\hat{\boldsymbol{\mu}}\|_\infty.$$

According to Lemma 7, we obtain

$$\left\| \zeta_1^{-1} n^{-1} \sum_{i=1}^{n} \hat{r}_i^{-2} \|\hat{\mathbf{\Omega}}^{1/2}\hat{\boldsymbol{\mu}}\|^2 \hat{\boldsymbol{U}}_i \right\|_\infty \le |1 + H_u| \cdot \left\| \zeta_1^{-1} n^{-1} \sum_{i=1}^{n} \hat{r}_i^{-2} \|\hat{\mathbf{\Omega}}^{1/2}\hat{\boldsymbol{\mu}}\|^2 \boldsymbol{U}_i \right\|_\infty$$

$$= O_p(n^{-1})\left[1 + O_p\{\lambda_n^{1-q} s_0(p)(\log p)^{1/2}\}\right] = O_p(n^{-1}).$$

Using the fact that $\zeta_1^{-1} n^{-1} \sum_{i=1}^{n} r_i^{m-1} = 1 + O_p(n^{-1/2})$ and Equation (11), we have

$$\frac{1}{n}\zeta_1^{-1} \sum_{i=1}^{n} \hat{r}_i^{-1} = \frac{1}{n}\zeta_1^{-1} \sum_{i=1}^{n} r_i^{-1}[1 + O_p\{\lambda_n^{1-q} s_0(p)\}]$$

$$= \left\{1 + O_p(n^{-1/2})\right\}\left[1 + O_p\{\lambda_n^{1-q} s_0(p)\}\right]$$

$$= 1 + O_p\{\lambda_n^{1-q} s_0(p)\}.$$

We final obtain:

$$\left\| \hat{\mathbf{\Omega}}^{1/2}\hat{\boldsymbol{\mu}} \right\|_\infty \lesssim \left\| \zeta_1^{-1} n^{-1} \sum_{i=1}^{n} \hat{\boldsymbol{U}}_i \right\|_\infty + \zeta_1^{-1} \left\| \hat{\mathbf{Q}}\hat{\mathbf{\Omega}}^{1/2}\hat{\boldsymbol{\mu}} \right\|_\infty$$

$$\lesssim p^{-1} \left\| \hat{\mathbf{\Omega}}^{1/2}\hat{\boldsymbol{\mu}} \right\|_\infty + O_p \left\{ n^{-1/2} \log^{1/2}(np) \right\}$$

$$+ O_p \left\{ n^{-1/2} p^{-1} + \lambda_n^{1-q} s_0(p) p^{-1} \right\} \|\hat{\mathbf{\Omega}}^{1/2}\hat{\boldsymbol{\mu}}\|_\infty.$$

Thus we conclude that:

$$\left\| \hat{\mathbf{\Omega}}^{1/2}\hat{\boldsymbol{\mu}} \right\|_\infty = O_p\{n^{-1/2} \log^{1/2}(np)\},$$

as $s_0(p) \asymp p^{1-\delta}$. In addition, by equation (17) we have

$$\left\| \zeta_1^{-1} \hat{\mathbf{Q}}\hat{\mathbf{\Omega}}^{1/2}\hat{\boldsymbol{\mu}} \right\|_\infty = O_p \left[ p^{1/2}\{n^{-1/2} p^{-1/2} + \lambda_n^{1-q} s_0(p) p^{-1/2}\} n^{-1/2} \log^{1/2}(np) \right]$$

$$= O_p \left\{ n^{-1} \log^{1/2}(np) + n^{-1/2} \lambda_n^{1-q} s_0(p) \log^{1/2}(np) \right\},$$

and

$$n^{-1} \sum_{i=1}^{n} \hat{r}_i^{-1} (1 + \delta_{1,i} + \delta_{2,i})$$

$$= \zeta_1 \left\{ 1 + O_p \left( n^{-1/4} \right) \right\} \left[ 1 + O_p\{\lambda_n^{1-q} s_0(p)\} \right]$$

$$= \zeta_1 \left[ 1 + O_p\{n^{-1/4} + \lambda_n^{1-q} s_0(p)\} \right].$$

Finally, we can write

$$n^{1/2}\hat{\mathbf{\Omega}}^{1/2}(\hat{\boldsymbol{\mu}} - \boldsymbol{\mu}) = n^{-1/2}\zeta_1^{-1} \sum_{i=1}^{n} \boldsymbol{U}_i + C_n,$$

where

$$C_n = \zeta_1^{-1}\left\{ \left( -2^{-1} n^{-1/2} \sum_{i=1}^{n} \hat{r}_i^{-2} \hat{\boldsymbol{U}}_i \right) \hat{\boldsymbol{\mu}}\hat{\mathbf{\Omega}}\hat{\boldsymbol{\mu}} \right\} + \zeta_1^{-1} \left( n^{-1/2} \sum_{i=1}^{n} \delta_{1,i}\hat{\boldsymbol{U}}_i \right) + \zeta_1^{-1} n^{1/2}\hat{\mathbf{Q}}\hat{\mathbf{\Omega}}^{1/2}\hat{\boldsymbol{\mu}}$$

$$+ n^{-1/2} \sum_{i=1}^{n} (\delta_{1,i} + \delta_{2,i})\hat{r}_i^{-1}\hat{\mathbf{\Omega}}^{1/2}\hat{\boldsymbol{\mu}}.$$

By previous discussion, we have

$$\|C_n\|_\infty = O_p\big[n^{-1/2} + n^{-1/2} + n^{-1} \log^{1/2}(np) + n^{-1/2}\lambda_n^{1-q} s_0(p) \log^{1/2}(np) + \big\{n^{-1/4}$$

$$+ \lambda_n^{1-q} s_0(p)\big\} \log^{1/2}(np)\big]$$

$$= O_p\{n^{-1/4} \log^{1/2}(np) + \lambda_n^{1-q} s_0(p) \log^{1/2}(np)\}.$$

Then we complete the proof. $\qquad\square$

***Proof of Lemma 2.*** Let $L_{n,p} = n^{-1/4}\log^{1/2}(np) + \lambda_n^{1-q}s_0(p)\log^{1/2}(np)$, according to Lemma 1, for any sequence $\eta_n \to \infty$ and any $t \in \mathbb{R}^p$,

$$\mathbb{P}\{n^{1/2}\hat{\boldsymbol{\Omega}}^{1/2}(\hat{\boldsymbol{\mu}} - \boldsymbol{\mu}) \leq t\} = \mathbb{P}\big(n^{-1/2}\zeta_1^{-1}\sum_{i=1}^n \boldsymbol{U}_i + C_n \leq t\big)$$

$$\leq \mathbb{P}\big(n^{-1/2}\zeta_1^{-1}\sum_{i=1}^n \boldsymbol{U}_i \leq t + \eta_n L_{n,p}\big) + \mathbb{P}(\|C_n\|_\infty > \eta_n L_{n,p}).$$

According to Lemma A4. in Cheng et al. (2023) and $\mathbb{E}\{(\zeta_1^{-1}U_{i,j})^4\} \lesssim 3$ and $\mathbb{E}\{(\zeta_1^{-1}U_{i,j})^2\} \gtrsim \bar{B}^{-2}$ uniformly for all $i = 1, 2, \cdots, n, j = 1, 2, \cdots, p$, the Gaussian approximation for independent partial sums in Koike (2021) yields:

$$\mathbb{P}\bigg(n^{1/2}\zeta_1^{-1}\sum_{i=1}^n \boldsymbol{U}_i \leq t + \eta_n L_{n,p}\bigg) \leq \mathbb{P}(\boldsymbol{Z} \leq t + \eta_n L_{n,p}) + O[\{n^{-1}\log^5(np)\}^{1/6}]$$

$$\leq \mathbb{P}(\boldsymbol{Z} \leq t) + O\{\eta_n L_{n,p}\log^{1/2}(p)\} + O[\{n^{-1}\log^5(np)\}^{1/6}],$$

where $\boldsymbol{Z} \sim \mathcal{N}\big(0, p^{-1}\zeta_1^{-2}\mathbf{I}_p\big)$, and the second inequality follows from Nazarov's inequality (Lemma 8). Thus,

$$\mathbb{P}\{n^{1/2}\hat{\boldsymbol{\Omega}}^{1/2}(\hat{\boldsymbol{\mu}} - \boldsymbol{\mu}) \leq t\} \leq \mathbb{P}(\boldsymbol{Z} \leq t) + O\{\eta_n L_{n,p}\log^{1/2}(p)\} + O(\{n^{-1}\log^5(np)\}^{1/6})$$

$$+ \mathbb{P}(|C_n|_\infty > \eta_n l_{n,p}).$$

On the other hand, we have

$$\mathbb{P}\{n^{1/2}\hat{\boldsymbol{\Omega}}^{1/2}(\hat{\boldsymbol{\mu}} - \boldsymbol{\mu}) \leq t\} \geq \mathbb{P}(\boldsymbol{Z} \leq t) - O\{\eta_n L_{n,p}\log^{1/2}(p)\} - O(\{n^{-1}\log^5(np)\}^{1/6}) - \mathbb{P}(\|C_n\|_\infty > \eta_n l_{n,p}),$$

where $\mathbb{P}(\|C_n\|_\infty > \eta_n l_{n,p}) \to 0$ as $n \to \infty$ by Lemma 1. Then we have that, if $\log p = o(n^{1/5})$,

$$\sup_{t \in \mathbb{R}^p} |\mathbb{P}\{n^{1/2}\hat{\boldsymbol{\Omega}}^{1/2}(\hat{\boldsymbol{\mu}} - \boldsymbol{\mu}) \leq t\} - \mathbb{P}(\boldsymbol{Z} \leq t)| \to 0.$$

Furthermore, by Corollary 3.1 in Chernozhukov et al. (2017), we have

$$\rho_n(\mathcal{A}^{si}) = \sup_{A \in \mathcal{A}^{si}} |\mathbb{P}\{n^{1/2}\hat{\boldsymbol{\Omega}}^{1/2}(\hat{\boldsymbol{\mu}} - \boldsymbol{\mu}) \in A\} - \mathbb{P}(\boldsymbol{Z} \in A)| \to 0.$$

The proof is thus complete. $\square$

### E.3 PROOF OF MAIN THEOREMS

***Proof of Theorem 1.*** Recall that $\boldsymbol{Z} \sim \mathcal{N}(0, p\zeta_1^2\mathbf{I}_p)$. Under the null hypothesis, Theorem 1 in Cai et al. (2013) establishes that as $p \to \infty$, we have

$$\mathbb{P}\left(p\zeta_1^2\max_{1 \leq i \leq p} Z_i^2 - 2\log p + \log\log p \leq x\right) \to F(x) = \exp\left(-\frac{1}{\sqrt{\pi}}e^{-x/2}\right),$$

for any $x \in \mathbb{R}$. Thus, by applying the triangle inequality, using Lemma 6 and Corollary 1, we obtain that under the null hypothesis,

$$\left|\mathbb{P}\left(n\left\|\hat{\boldsymbol{\Omega}}^{1/2}\hat{\boldsymbol{\mu}}\right\|_\infty^2 \hat{\zeta}_1^2 p - 2\log p + \log\log p \leq x\right) - F(x)\right|$$

$$\leq \left|\mathbb{P}\left(n\left\|\hat{\boldsymbol{\Omega}}^{1/2}\hat{\boldsymbol{\mu}}\right\|_\infty^2 \zeta_1^2 p - 2\log p + \log\log p \leq x\right) - F(x)\right| + o(1)$$

$$\leq \left|\mathbb{P}\left(n\left\|\hat{\boldsymbol{\Omega}}^{1/2}\hat{\boldsymbol{\mu}}\right\|_\infty^2 \zeta_1^2 p - 2\log p + \log\log p \leq x\right) - \mathbb{P}\left(p\zeta_1^2\max_{1 \leq i \leq p} Z_i^2 - 2\log p + \log\log p \leq x\right)\right|$$

$$+ \left|\mathbb{P}\left(p\zeta_1^2\max_{1 \leq i \leq p} Z_i^2 - 2\log p + \log\log p \leq x\right) - F(x)\right| + o(1) \to 0,$$

for any $x \in \mathbb{R}$. $\square$

**Proof of Theorem 2.** Under alternative hypothesis for small $\alpha$, we have

$$\mathbb{P}\left(T_{MAX} > q_{1-\alpha}\Big|H_1\right)$$

$$= \mathbb{P}\left(n\|\hat{\boldsymbol{\Omega}}^{1/2}\hat{\boldsymbol{\mu}}\|_\infty^2\hat{\zeta}_1^2 p - 2\log p + \log\log p > q_{1-\alpha}\Big|H_1\right)$$

$$= \mathbb{P}\left(n^{1/2}\|\hat{\boldsymbol{\Omega}}^{1/2}\hat{\boldsymbol{\mu}}\|_\infty\hat{\zeta}_1 p^{1/2} > (2\log p + \log\log p + q_{1-\alpha})^{1/2}\Big|H_1\right)$$

$$\geq \mathbb{P}\left(n^{1/2}\|\hat{\boldsymbol{\Omega}}^{1/2}\boldsymbol{\mu}\|_\infty\hat{\zeta}_1 p^{1/2} - n^{1/2}\|\hat{\boldsymbol{\Omega}}^{1/2}(\hat{\boldsymbol{\mu}} - \boldsymbol{\mu})\|_\infty\hat{\zeta}_1 p^{1/2} > (2\log p + \log\log p + q_{1-\alpha})^{1/2}\Big|H_1\right)$$

$$= \mathbb{P}\Big(n\|\hat{\boldsymbol{\Omega}}^{1/2}(\hat{\boldsymbol{\mu}} - \boldsymbol{\mu})\|_\infty^2\hat{\zeta}_1^2 p - 2\log p + \log\log p$$

$$\qquad \leq n\|\hat{\boldsymbol{\Omega}}^{1/2}\boldsymbol{\mu}\|_\infty^2\hat{\zeta}_1^2 p - 2(2\log p + \log\log p + q_{1-\alpha})^{1/2}n^{1/2}\|\hat{\boldsymbol{\Omega}}^{1/2}\boldsymbol{\mu}\|_\infty\hat{\zeta}_1 p^{1/2} + q_{1-\alpha}\Big|H_1\Big).$$

By Lemma 4, Lemma 6 and Theorem 1, we have

$$\mathbb{P}\left(T_{MAX} > q_{1-\alpha}\Big|H_1\right)$$

$$\leq \mathbb{P}\Big(n\|\hat{\boldsymbol{\Omega}}^{1/2}(\hat{\boldsymbol{\mu}} - \boldsymbol{\mu})\|_\infty^2\hat{\zeta}_1^2 p - 2\log p + \log\log p$$

$$\qquad \geq n\|\hat{\boldsymbol{\Omega}}^{1/2}\boldsymbol{\mu}\|_\infty^2\hat{\zeta}_1^2 p - 2(2\log p + \log\log p + q_{1-\alpha})^{1/2}n^{1/2}\|\hat{\boldsymbol{\Omega}}^{1/2}\boldsymbol{\mu}\|_\infty\hat{\zeta}_1 p^{1/2} + q_{1-\alpha}\Big|H_1\Big)$$

$$= \mathbb{P}\Big(n\|\hat{\boldsymbol{\Omega}}^{1/2}(\hat{\boldsymbol{\mu}} - \boldsymbol{\mu})\|_\infty^2\zeta_1^2 p - 2\log p + \log\log p$$

$$\qquad \leq n\|\boldsymbol{\Omega}^{1/2}\boldsymbol{\mu}\|_\infty^2\zeta_1^2 p - 2(2\log p + \log\log p + q_{1-\alpha})^{1/2}n^{1/2}\|\boldsymbol{\Omega}^{1/2}\boldsymbol{\mu}\|_\infty\zeta_1 p^{1/2} + q_{1-\alpha} + o(1)\Big|H_1\Big)$$

$$= F\left(n\|\boldsymbol{\Omega}^{1/2}\boldsymbol{\mu}\|_\infty^2\zeta_1^2 p - 2(2\log p + \log\log p + q_{1-\alpha})^{1/2}n^{1/2}\|\boldsymbol{\Omega}^{1/2}\boldsymbol{\mu}\|_\infty\zeta_1 p^{1/2} + q_{1-\alpha} + o(1)\right) + o(1) \to 1,$$

when $\|\boldsymbol{\Omega}^{1/2}\boldsymbol{\mu}\|_\infty \geq \widetilde{C}n^{-1/2}\{\log p - 2\log\log(1-\alpha)^{-1}\}^{1/2}$. $\square$

**Proof of Theorem 3.** By Lemma 2, we have the Gaussian approximation

$$\sup_{A\in\mathcal{A}^{\mathrm{re}}}\left|\mathbb{P}\left(n^{1/2}p^{1/2}\zeta_1\hat{\boldsymbol{\Omega}}^{1/2}\hat{\boldsymbol{\mu}} \in A\right) - \mathbb{P}\left(\boldsymbol{G} + n^{1/2}p^{1/2}\zeta_1\hat{\boldsymbol{\Omega}}^{1/2}\boldsymbol{\mu} \in A\right)\right| \to 0,$$

where $\boldsymbol{G} := p^{1/2}\zeta_1\boldsymbol{Z} \sim \mathcal{N}(0, \mathbf{I}_p)$. Then

$$\sup_{t\in\mathbb{R}}\left|\mathbb{P}\left(\left\|n^{1/2}p^{1/2}\zeta_1\hat{\boldsymbol{\Omega}}^{1/2}\hat{\boldsymbol{\mu}}\right\|^2 \leq t\right) - \mathbb{P}\left(\left\|\boldsymbol{G} + n^{1/2}p^{1/2}\zeta_1\hat{\boldsymbol{\Omega}}^{1/2}\boldsymbol{\mu}\right\|^2 \leq t\right)\right|$$

$$= \sup_{t\in\mathbb{R}}\left|\mathbb{P}\left(\left\|n^{1/2}p^{1/2}\zeta_1\hat{\boldsymbol{\Omega}}^{1/2}\hat{\boldsymbol{\mu}}\right\|^2 \leq t\right) - \mathbb{P}\left\{\chi^2\left(p, \left\|n^{1/2}p^{1/2}\zeta_1\hat{\boldsymbol{\Omega}}^{1/2}\boldsymbol{\mu}\right\|^2\right) \leq t\right\}\right|$$

$$\to \sup_{t\in\mathbb{R}}\left|\mathbb{P}\left(\left\|n^{1/2}p^{1/2}\zeta_1\hat{\boldsymbol{\Omega}}^{1/2}\hat{\boldsymbol{\mu}}\right\|^2 \leq t\right) - \mathbb{P}\left\{\chi^2\left(p, \left\|n^{1/2}p^{1/2}\zeta_1\boldsymbol{\Omega}^{1/2}\boldsymbol{\mu}\right\|^2\right) \leq t\right\}\right|$$

$$= \sup_{t\in\mathbb{R}}\Bigg|\mathbb{P}\left(\frac{\left\|n^{1/2}p^{1/2}\zeta_1\hat{\boldsymbol{\Omega}}^{1/2}\hat{\boldsymbol{\mu}}\right\|^2 - p - \left\|n^{1/2}p^{1/2}\zeta_1\boldsymbol{\Omega}^{1/2}\boldsymbol{\mu}\right\|^2}{\sqrt{2p + 4\left\|n^{1/2}p^{1/2}\zeta_1\boldsymbol{\Omega}^{1/2}\boldsymbol{\mu}\right\|^2}} \leq t\right)$$

$$\qquad - \mathbb{P}\left\{\frac{\chi^2\left(p, \left\|n^{1/2}p^{1/2}\zeta_1\hat{\boldsymbol{\Omega}}^{1/2}\boldsymbol{\mu}\right\|^2\right) - p - \left\|n^{1/2}p^{1/2}\zeta_1\boldsymbol{\Omega}^{1/2}\boldsymbol{\mu}\right\|^2}{\sqrt{2p + 4\left\|n^{1/2}p^{1/2}\zeta_1\boldsymbol{\Omega}^{1/2}\boldsymbol{\mu}\right\|^2}} \leq t\right\}\Bigg|$$

$$= \sup_{t\in\mathbb{R}}\left|\mathbb{P}\left(\frac{\left\|n^{1/2}p^{1/2}\zeta_1\hat{\boldsymbol{\Omega}}^{1/2}\hat{\boldsymbol{\mu}}\right\|^2 - p - \left\|n^{1/2}p^{1/2}\zeta_1\boldsymbol{\Omega}^{1/2}\boldsymbol{\mu}\right\|^2}{\sqrt{2p + 4\left\|n^{1/2}p^{1/2}\zeta_1\boldsymbol{\Omega}^{1/2}\boldsymbol{\mu}\right\|^2}} \leq t\right) - \Phi(t)\right| \to 0,$$

as $(n, p) \to \infty$. Therefore, under the null hypothesis, we have $T_{SUM} \xrightarrow{d} N(0, 1)$; Under the alternative hypothesis, assuming $\left\| n^{1/2} p^{1/2} \zeta_1 \boldsymbol{\Omega}^{1/2} \boldsymbol{\mu} \right\|^2 = o(p)$, we have

$$T_{SUM} - 2^{-1/2} n p^{1/2} \zeta_1^2 \boldsymbol{\mu}^\top \boldsymbol{\Omega} \boldsymbol{\mu} \xrightarrow{d} \mathcal{N}(0, 1).$$

Then we complete the proof. $\qquad\square$

***Proof of Theorem 4.*** Recall that Corollary 1, as $n \to \infty$, we have

$$\tilde{\rho}_{n,comb} = \sup_{t_1, t_2 \in \mathbb{R}} \left| \mathbb{P}\left( n^{1/2} \| \hat{\boldsymbol{\Omega}}^{1/2}(\hat{\boldsymbol{\mu}} - \boldsymbol{\mu}) \|_\infty \leqslant t_1, n^{1/2} \| \hat{\boldsymbol{\Omega}}^{1/2}(\hat{\boldsymbol{\mu}} - \boldsymbol{\mu}) \| \leqslant t_2 \right) \right.$$

$$\left. - \mathbb{P}\left( \| \boldsymbol{Z} \|_\infty \leqslant t_1, \| \boldsymbol{Z} \| \leqslant t_2 \right) \right| \to 0.$$

By $\| \boldsymbol{\mu} \|_\infty = o(n^{-1/2})$, $\| \boldsymbol{\mu} \| = o(p^{1/4} n^{-1/2})$, Assumption 3 and Lemma 4 we have

$$\sup_{t_1, t_2 \in \mathbb{R}} \left| \mathbb{P}\left( n^{1/2} p^{1/2} \zeta_1 \| \hat{\boldsymbol{\Omega}}^{1/2} \hat{\boldsymbol{\mu}} \|_\infty + o(1) \leqslant t_1, n^{1/2} p^{1/2} \zeta_1 \| \hat{\boldsymbol{\Omega}}^{1/2} \hat{\boldsymbol{\mu}} \| + o(p^{1/2}) \leqslant t_2 \right) \right.$$

$$\left. - \mathbb{P}\left( p^{1/2} \zeta_1 \| \boldsymbol{Z} \|_\infty \leqslant t_1, p^{1/2} \zeta_1 \| \boldsymbol{Z} \| \leqslant t_2 \right) \right| \to 0.$$

Hence, applying the continuous mapping theorem, we obtain that

$$\sup_{t_1, t_2 \in \mathbb{R}} \left| \mathbb{P}\left( T_{MAX} + o(1) \leqslant t_1, T_{SUM} + o(1) \leqslant t_2 \right) \right.$$

$$\left. - \mathbb{P}\left( p \zeta_1^2 \| \boldsymbol{Z} \|_\infty^2 - 2 \log p + \log \log p \leqslant t_1, (2p)^{-1/2} (p \zeta_1^2 \| \boldsymbol{Z} \|^2 - p) \leqslant t_2 \right) \right| \to 0.$$

By Theorem 3 in Feng et al. (2024), we have $p^{1/2} \zeta_1 \| \boldsymbol{Z} \|_\infty^2 - 2 \log p + \log \log p$ and $(2p)^{-1/2}(p \zeta_1^2 \| \boldsymbol{Z} \|^2 - p)$ are asymptotic independent as $p \to \infty$, so we have $T_{MAX}$ and $T_{SUM}$ are asymptotic independent as $n, p \to \infty$. $\quad\square$

***Proof of Theorem 5.*** Set $Q(\boldsymbol{x}) = \Delta_d^2(\boldsymbol{x}) - c\varsigma_p$ and $\hat{Q}(\boldsymbol{x}) = \hat{\Delta}_d^2(\boldsymbol{x}) - c\hat{\varsigma}_p$. Thus we have

$$R_{HRQDA} - R_{QDA}$$

$$= \int_{\hat{Q} < 0} \frac{1}{2} f_1(\boldsymbol{x}) \mathrm{d}\boldsymbol{x} + \int_{\hat{Q} \geq 0} \frac{1}{2} f_2(\boldsymbol{x}) \mathrm{d}\boldsymbol{x} - \left( \int_{Q < 0} \frac{1}{2} f_1(\boldsymbol{x}) \mathrm{d}\boldsymbol{x} + \int_{Q \geq 0} \frac{1}{2} f_2(\boldsymbol{x}) \mathrm{d}\boldsymbol{x} \right)$$

$$= \int_{Q(\boldsymbol{x}) \geq 0} \frac{1}{2} \{ f_1(\boldsymbol{x}) - f_2(\boldsymbol{x}) \} \mathrm{d}\boldsymbol{x} + \int_{\hat{Q}(\boldsymbol{x}) < 0} \frac{1}{2} \{ f_1(\boldsymbol{x}) - f_2(\boldsymbol{x}) \} \mathrm{d}\boldsymbol{x}.$$

Notice that $\int \frac{1}{2} \{ f_1(\boldsymbol{x}) - f_2(\boldsymbol{x}) \} \mathrm{d}\boldsymbol{x} = 0$, we have

$$|R_{HRQDA} - R_{QDA}| \tag{18}$$

$$= \left| \int_{Q(\boldsymbol{x}) \geq 0, \hat{Q}(\boldsymbol{x}) < 0} \frac{1}{2} \{ f_1(\boldsymbol{x}) - f_2(\boldsymbol{x}) \} \mathrm{d}\boldsymbol{x} \right|$$

$$\leq \frac{1}{2} \mathbb{E}_{\boldsymbol{x} \sim f_1} \mathbf{1} \{ 0 \leq Q(\boldsymbol{x}) < Q(\boldsymbol{x}) - \hat{Q}(\boldsymbol{x}) \} + \frac{1}{2} \mathbb{E}_{\boldsymbol{x} \sim f_2} \mathbf{1} \{ 0 \leq Q(\boldsymbol{x}) < Q(\boldsymbol{x}) - \hat{Q}(\boldsymbol{x}) \}$$

$$= \frac{1}{2} \mathbb{P}_{x \sim f_1} \left\{ 0 \leq \frac{1}{p} Q(\boldsymbol{x}) < \frac{1}{p} M(\boldsymbol{x}) \right\} + \frac{1}{2} \mathbb{P}_{x \sim f_2} \left\{ 0 \leq \frac{1}{p} Q(\boldsymbol{x}) < \frac{1}{p} M(\boldsymbol{x}) \right\}, \tag{19}$$

where $M(\boldsymbol{x}) := Q(\boldsymbol{x}) - \hat{Q}(\boldsymbol{x})$. By calculations, we can get

$$M(\boldsymbol{x}) = (\boldsymbol{x} - \boldsymbol{\mu}_1)^\top \{ \boldsymbol{\Xi}_2^{-1} - \boldsymbol{\Xi}_1^{-1} - (\tilde{\boldsymbol{\Omega}}_2 - \tilde{\boldsymbol{\Omega}}_1) \} (\boldsymbol{x} - \boldsymbol{\mu}_1) - 2(\boldsymbol{\mu}_1 - \hat{\boldsymbol{\mu}})^\top (\tilde{\boldsymbol{\Omega}}_2 - \tilde{\boldsymbol{\Omega}}_1)(\boldsymbol{x} - \boldsymbol{\mu}_1)$$

$$+ 2(\boldsymbol{\delta}^\top \boldsymbol{\Xi}_2^{-1} - \hat{\boldsymbol{\delta}}^\top \tilde{\boldsymbol{\Omega}}_2)(\boldsymbol{x} - \boldsymbol{\mu}_1) + (\boldsymbol{\mu}_1 - \hat{\boldsymbol{\mu}}_1)^\top (\tilde{\boldsymbol{\Omega}}_2 - \tilde{\boldsymbol{\Omega}}_1)(\boldsymbol{\mu}_1 - \hat{\boldsymbol{\mu}}_1) - 2\hat{\boldsymbol{\delta}}^\top \tilde{\boldsymbol{\Omega}}_2(\boldsymbol{\mu}_1 - \hat{\boldsymbol{\mu}}_1)$$

$$+ \boldsymbol{\delta}^\top \boldsymbol{\Xi}_2^{-1} \boldsymbol{\delta} - \hat{\boldsymbol{\delta}}^\top \tilde{\boldsymbol{\Omega}}_2 \hat{\boldsymbol{\delta}} - c(\varsigma_p - \hat{\varsigma}_p).$$

Next we calculate the variance of $p^{-1}Q(\boldsymbol{x})$

$$\mathrm{Var}_{\boldsymbol{x}\sim f_1}\left\{\frac{1}{p}Q(\boldsymbol{x})\right\}$$

$$=\mathrm{Var}_{\boldsymbol{x}\sim f_1}\left\{\frac{1}{p}(\boldsymbol{x}-\boldsymbol{\mu}_1)^\top(\boldsymbol{\Xi}_2^{-1}-\boldsymbol{\Xi}_1^{-1})(\boldsymbol{x}-\boldsymbol{\mu}_1)-\frac{1}{p}2\boldsymbol{\delta}^\top\boldsymbol{\Xi}_2^{-1}(\boldsymbol{x}-\boldsymbol{\mu}_1)\right\}$$

$$=\mathbb{E}_{\boldsymbol{x}\sim f_1}\left[\left\{\frac{1}{p}(\boldsymbol{x}-\boldsymbol{\mu}_1)^\top(\boldsymbol{\Xi}_2^{-1}-\boldsymbol{\Xi}_1^{-1})(\boldsymbol{x}-\boldsymbol{\mu}_1)\right\}^2+\left\{\frac{1}{p}2\boldsymbol{\delta}^\top\boldsymbol{\Xi}_2^{-1}(\boldsymbol{x}-\boldsymbol{\mu}_1)\right\}^2\right]$$

$$-\left[\mathbb{E}_{\boldsymbol{x}\sim f_1}\left\{\frac{1}{p}(\boldsymbol{x}-\boldsymbol{\mu}_1)^\top(\boldsymbol{\Xi}_2^{-1}-\boldsymbol{\Xi}_1^{-1})(\boldsymbol{x}-\boldsymbol{\mu}_1)\right\}\right]^2$$

$$=\mathrm{Var}_{x\sim f_1}\left\{\frac{1}{p}(\boldsymbol{x}-\boldsymbol{\mu}_1)^\top(\boldsymbol{\Xi}_2^{-1}-\boldsymbol{\Xi}_1^{-1})(\boldsymbol{x}-\boldsymbol{\mu}_1)\right\}+\mathbb{E}_{\boldsymbol{x}\sim f_1}\left\{\frac{1}{p}2\boldsymbol{\delta}^\top\boldsymbol{\Xi}_2^{-1}(\boldsymbol{x}-\boldsymbol{\mu}_1)\right\}^2$$

$$=\mathrm{Var}\left\{\frac{r^2}{p}\boldsymbol{U}^\top\boldsymbol{\Xi}_1^{1/2}(\boldsymbol{\Xi}_2^{-1}-\boldsymbol{\Xi}_1^{-1})\boldsymbol{\Xi}_1^{1/2}\boldsymbol{U}\right\}+\mathrm{Var}\left\{\frac{2r}{p}\boldsymbol{\delta}^\top\boldsymbol{\Xi}_2^{-1}\boldsymbol{\Xi}_1^{1/2}\boldsymbol{U}\right\}$$

$$=\mathbb{E}\left(\frac{r^4}{p^2}\right)\mathrm{Var}\left\{\boldsymbol{U}^\top\boldsymbol{\Xi}_1^{1/2}(\boldsymbol{\Xi}_2^{-1}-\boldsymbol{\Xi}_1^{-1})\boldsymbol{\Xi}_1^{1/2}\boldsymbol{U}\right\}+\mathbb{E}\left(\frac{4r^2}{p}\right)\mathrm{Var}\left(\frac{1}{\sqrt{p}}\boldsymbol{\delta}^\top\boldsymbol{\Xi}_2^{-1}\boldsymbol{\Xi}_1^{1/2}\boldsymbol{U}\right)$$

$$+\mathrm{Var}\left(\frac{r^2}{p}\right)\left[\mathbb{E}\left\{\boldsymbol{U}^\top\boldsymbol{\Xi}_1^{1/2}(\boldsymbol{\Xi}_2^{-1}-\boldsymbol{\Xi}_1^{-1})\boldsymbol{\Xi}_1^{1/2}\boldsymbol{U}\right\}\right]^2+\mathrm{Var}\left(\frac{2r}{\sqrt{p}}\right)\left\{\mathbb{E}\left(\frac{1}{\sqrt{p}}\boldsymbol{\delta}^\top\boldsymbol{\Xi}_2^{-1}\boldsymbol{\Xi}_1^{1/2}\boldsymbol{U}\right)\right\}^2.$$

By Assumptions 1, 6 and Lemma 10 we have

$$\mathrm{Var}_{\boldsymbol{x}\sim f_1}\left\{\frac{1}{p}Q(\boldsymbol{x})\right\}\asymp\frac{1}{p^2}\left\{\|\boldsymbol{\Xi}_1^{1/2}(\boldsymbol{\Xi}_2^{-1}-\boldsymbol{\Xi}_1^{-1})\boldsymbol{\Xi}_1^{1/2}\|_F^2+\|\boldsymbol{\Xi}_2^{-1}\boldsymbol{\Xi}_1^{1/2}\boldsymbol{\delta}\|^2\right\}.$$

By Assumptions 2 and 3 we have $\|\boldsymbol{\Xi}_1^{1/2}(\boldsymbol{\Xi}_2^{-1}-\boldsymbol{\Xi}_1^{-1})\boldsymbol{\Xi}_1^{1/2}\|_F\asymp\|\boldsymbol{\Xi}_2^{1/2}(\boldsymbol{\Xi}_1^{-1}-\boldsymbol{\Xi}_2^{-1})\boldsymbol{\Xi}_2^{1/2}\|_F\asymp\|\boldsymbol{\Sigma}_1^{1/2}(\boldsymbol{\Omega}_2-\boldsymbol{\Omega}_1)\boldsymbol{\Sigma}_1^{1/2}\|_F\asymp\|\boldsymbol{\Sigma}_2^{1/2}(\boldsymbol{\Omega}_1-\boldsymbol{\Omega}_2)\boldsymbol{\Sigma}_2^{1/2}\|_F$ and $\|\boldsymbol{\Xi}_2^{-1}\boldsymbol{\Xi}_1^{1/2}\boldsymbol{\delta}\|\asymp\sqrt{p/t_0(p)}\|\boldsymbol{\delta}\|$. Thus, we have $\mathrm{Var}_{\boldsymbol{x}\sim f_1}\{p^{-1}Q(\boldsymbol{x})\}\asymp p^{-2}\sigma_Q^2(p)\asymp 1$. Similarly, $\mathrm{Var}_{\boldsymbol{x}\sim f_2}p^{-1}Q(\boldsymbol{x})\}\asymp p^{-2}\sigma_Q^2(p)\asymp 1$.

Next, we bound the discrepancy between $\tilde{\boldsymbol{\Omega}}_i$ and $\boldsymbol{\Xi}_i^{-1}$ by Lemma 4 and 11.

$$\|\tilde{\boldsymbol{\Omega}}_i-\boldsymbol{\Xi}_i^{-1}\|_\infty=\left\|\frac{p}{\widehat{\mathrm{tr}(\boldsymbol{\Xi}_i)}}\hat{\boldsymbol{\Omega}}_i-\frac{p}{\mathrm{tr}(\boldsymbol{\Xi}_i)}\boldsymbol{\Omega}_i\right\|_\infty$$

$$\leq\left(\frac{p}{\widehat{\mathrm{tr}(\boldsymbol{\Xi}_i)}}\left|\frac{\widehat{\mathrm{tr}(\boldsymbol{\Xi}_i)}}{\mathrm{tr}(\boldsymbol{\Xi}_i)}-1\right|\right)\|\hat{\boldsymbol{\Omega}}_i\|_\infty+\|\hat{\boldsymbol{\Omega}}_i-\boldsymbol{\Omega}_i\|_\infty$$

$$=O_p(\lambda_n+n^{-1/2})=O_p(\lambda_n).$$

Similarly, we have $\|\tilde{\boldsymbol{\Omega}}_i-\boldsymbol{\Xi}_i^{-1}\|_{op}\leq\|\tilde{\boldsymbol{\Omega}}_i-\boldsymbol{\Xi}_i^{-1}\|_{L_1}=O_p\{\lambda_n^{1-q}s_0(p)\}$. And $p^{-1}\|\tilde{\boldsymbol{\Omega}}_i-\boldsymbol{\Xi}_i^{-1}\|_F^2\leq\|\tilde{\boldsymbol{\Omega}}_i-\boldsymbol{\Xi}_i^{-1}\|_{L_1}$, $\|\tilde{\boldsymbol{\Omega}}_i-\boldsymbol{\Xi}_i^{-1}\|_\infty=O_p\{\lambda_n^{2-q}s_0(p)\}$. By the proof of Lemma 1 we have $\|\boldsymbol{\mu}-\hat{\boldsymbol{\mu}}\|=O_p(p^{1/2}n^{-1/2})$ and $\|\boldsymbol{\mu}-\hat{\boldsymbol{\mu}}\|_\infty=O_p\{n^{-1/2}\log^{1/2}(np)\}$. Then we bound the $p^{-1}M(\boldsymbol{x})$ under $\boldsymbol{x}\sim f_1$.

$$\frac{1}{p}(\boldsymbol{x}-\boldsymbol{\mu}_1)^\top\{\boldsymbol{\Xi}_2^{-1}-\boldsymbol{\Xi}_1^{-1}-(\tilde{\boldsymbol{\Omega}}_2-\tilde{\boldsymbol{\Omega}}_1)\}(\boldsymbol{x}-\boldsymbol{\mu}_1)=\frac{r^2}{p}\boldsymbol{U}^\top\boldsymbol{\Sigma}_1^{1/2}\{\boldsymbol{\Xi}_2^{-1}-\boldsymbol{\Xi}_1^{-1}-(\tilde{\boldsymbol{\Omega}}_2-\tilde{\boldsymbol{\Omega}}_1)\}\boldsymbol{\Sigma}_1^{1/2}\boldsymbol{U}$$

$$\leq\frac{r^2}{p}\|\boldsymbol{\Sigma}_1^{1/2}(\boldsymbol{\Xi}_2^{-1}-\boldsymbol{\Xi}_1^{-1}-(\tilde{\boldsymbol{\Omega}}_2-\tilde{\boldsymbol{\Omega}}_1))\boldsymbol{\Sigma}_1^{1/2}\|_{op}$$

$$=O_p\{\lambda_n^{1-q}s_0(p)\},$$

$$\frac{1}{p}(\boldsymbol{\mu}_1-\hat{\boldsymbol{\mu}})^\top(\tilde{\boldsymbol{\Omega}}_2-\tilde{\boldsymbol{\Omega}}_1)(\boldsymbol{x}-\boldsymbol{\mu}_1)\leq\frac{1}{\sqrt{p}}\|\boldsymbol{\mu}_1-\hat{\boldsymbol{\mu}}_1\|\frac{r}{\sqrt{p}}\|(\tilde{\boldsymbol{\Omega}}_2-\tilde{\boldsymbol{\Omega}}_1)\boldsymbol{\Sigma}_1^{1/2}\boldsymbol{U}\|$$

$$=O_p(n^{-1/2}),$$

$$\frac{1}{p}(\boldsymbol{\delta}^\top \boldsymbol{\Xi}_2^{-1} - \hat{\boldsymbol{\delta}}^\top \tilde{\boldsymbol{\Omega}}_2)(\boldsymbol{x} - \boldsymbol{\mu}_1) \leq \frac{1}{p}(\|\boldsymbol{\delta}\boldsymbol{\Xi}_2^{-1} - \boldsymbol{\delta}\tilde{\boldsymbol{\Omega}}_2\| + \|\boldsymbol{\delta}\tilde{\boldsymbol{\Omega}}_2 - \hat{\boldsymbol{\delta}}\tilde{\boldsymbol{\Omega}}_2\|)\|\boldsymbol{x} - \boldsymbol{\mu}_1\|$$

$$\leq \frac{1}{p}(\|\boldsymbol{\delta}\|\|\boldsymbol{\Xi}_2^{-1} - \tilde{\boldsymbol{\Omega}}_2\|_{op} + \|\boldsymbol{\delta} - \hat{\boldsymbol{\delta}}\|\|\tilde{\boldsymbol{\Omega}}\|_{op})\|\boldsymbol{x} - \boldsymbol{\mu}_1\|$$

$$= O_p(\lambda_n^{1-q}s_0(p) + n^{-1/2}) = O_p\{\lambda_n^{1-q}s_0(p)\},$$

$$\frac{1}{p}(\boldsymbol{\mu}_1 - \hat{\boldsymbol{\mu}}_1)^\top (\tilde{\boldsymbol{\Omega}}_2 - \tilde{\boldsymbol{\Omega}}_1)(\boldsymbol{\mu}_1 - \hat{\boldsymbol{\mu}}_1) \leq \frac{1}{p}\|\boldsymbol{\mu}_1 - \hat{\boldsymbol{\mu}}_1\|^2\|\tilde{\boldsymbol{\Omega}}_2 - \tilde{\boldsymbol{\Omega}}_1\|_{op} = O_p(n^{-1}),$$

$$\frac{1}{p}\hat{\boldsymbol{\delta}}^\top \tilde{\boldsymbol{\Omega}}_2(\boldsymbol{\mu}_1 - \hat{\boldsymbol{\mu}}_1) \leq \frac{1}{p}\|\hat{\boldsymbol{\delta}}\|\|\tilde{\boldsymbol{\Omega}}_2\|_{op}\|\boldsymbol{\mu}_1 - \hat{\boldsymbol{\mu}}_1\| = O_p(n^{-1/2}),$$

$$\frac{1}{p}(\boldsymbol{\delta}^\top \boldsymbol{\Xi}_2^{-1}\boldsymbol{\delta} - \hat{\boldsymbol{\delta}}^\top \tilde{\boldsymbol{\Omega}}_2\hat{\boldsymbol{\delta}}) \leq \frac{1}{p}(\boldsymbol{\delta}^\top \boldsymbol{\Xi}_2^{-1}\boldsymbol{\delta} - \boldsymbol{\delta}^\top \tilde{\boldsymbol{\Omega}}_2\boldsymbol{\delta} + \boldsymbol{\delta}^\top \tilde{\boldsymbol{\Omega}}_2\boldsymbol{\delta} - \boldsymbol{\delta}^\top \tilde{\boldsymbol{\Omega}}_2\hat{\boldsymbol{\delta}} + \boldsymbol{\delta}^\top \tilde{\boldsymbol{\Omega}}_2\hat{\boldsymbol{\delta}} - \hat{\boldsymbol{\delta}}^\top \tilde{\boldsymbol{\Omega}}_2\hat{\boldsymbol{\delta}})$$

$$= O_p\{\lambda_n^{1-q}s_0(p) + n^{-1/2}\} = O_p\{\lambda_n^{1-q}s_0(p)\}.$$

Denote $\mathbf{D}_\Omega = \boldsymbol{\Xi}_2^{-1} - \boldsymbol{\Xi}_1^{-1}$, $\tilde{\mathbf{D}}_\Omega = \tilde{\boldsymbol{\Omega}}_2 - \tilde{\boldsymbol{\Omega}}_1$,

$$(\hat{\varsigma}_p - \varsigma_p) = \log|\tilde{\mathbf{D}}_\Omega \tilde{\boldsymbol{\Omega}}_1^{-1} + \mathbf{I}_p| - \log|\mathbf{D}_\Omega \boldsymbol{\Xi}_1 + \mathbf{I}_p|$$

$$\leq \text{tr}\{(\mathbf{D}_\Omega \boldsymbol{\Xi}_1 + \mathbf{I}_p)^{-1}(\tilde{\mathbf{D}}_\Omega \tilde{\boldsymbol{\Omega}}_1^{-1} - \mathbf{D}_\Omega \boldsymbol{\Xi}_1)\}$$

$$= \text{tr}\{(-\mathbf{D}_\Omega \boldsymbol{\Xi}_2 + \mathbf{I}_p)(\tilde{\mathbf{D}}_\Omega \tilde{\boldsymbol{\Omega}}_1^{-1} - \mathbf{D}_\Omega \boldsymbol{\Xi}_1)\}$$

$$= \text{tr}\{(-\mathbf{D}_\Omega \boldsymbol{\Xi}_2)(\tilde{\mathbf{D}}_\Omega \tilde{\boldsymbol{\Omega}}_1^{-1} - \mathbf{D}_\Omega \boldsymbol{\Xi}_1)\} + \text{tr}(\tilde{\mathbf{D}}_\Omega \tilde{\boldsymbol{\Omega}}_1^{-1} - \mathbf{D}_\Omega \boldsymbol{\Xi}_1)$$

$$\leq \|\mathbf{D}_\Omega \boldsymbol{\Xi}_2\|_F \cdot \|\tilde{\mathbf{D}}_\Omega \tilde{\boldsymbol{\Omega}}_1^{-1} - \mathbf{D}_\Omega \boldsymbol{\Xi}_1\|_F + \text{tr}(\tilde{\mathbf{D}}_\Omega \tilde{\boldsymbol{\Omega}}_1^{-1} - \mathbf{D}_\Omega \boldsymbol{\Xi}_1)$$

$$\lesssim \|\mathbf{D}_\Omega\|_F \|\boldsymbol{\Xi}_2\|_{op} \cdot \|\tilde{\mathbf{D}}_\Omega \tilde{\boldsymbol{\Omega}}_1^{-1} - \mathbf{D}_\Omega \boldsymbol{\Xi}_1\|_F,$$

where

$$\left\|\mathbf{D}_\Omega \boldsymbol{\Xi}_1 - \tilde{\mathbf{D}}_\Omega \tilde{\boldsymbol{\Omega}}_1^{-1}\right\|_F$$

$$\leq \left\|\mathbf{D}_\Omega \boldsymbol{\Xi}_1 - \tilde{\mathbf{D}}_\Omega \boldsymbol{\Xi}_1\right\|_F + \left\|\tilde{\mathbf{D}}_\Omega(\boldsymbol{\Xi}_1 - \tilde{\boldsymbol{\Omega}}_1^{-1})\right\|_F$$

$$\leq \|\mathbf{D}_\Omega - \tilde{\mathbf{D}}_\Omega\|_F \|\boldsymbol{\Xi}_1\|_{op} + \|\tilde{\mathbf{D}}_\Omega\|_F \|\boldsymbol{\Xi}_1 - \tilde{\boldsymbol{\Omega}}_1^{-1}\|_{op}.$$

Then, we can find $p^{-1}(\hat{\varsigma}_d - \varsigma_d) = O_p\{\lambda_n^{1-q/2}s_0(p)^{1/2} + \lambda_n^{1-q}s_0(p)\}$. Thus, under $\boldsymbol{x} \sim f_1$ we have $p^{-1}M(\boldsymbol{x}) = O_p\{\lambda_n^{1-q/2}s_0(p)^{1/2} + \lambda_n^{1-q}s_0(p)\}$. Similarly, we can get the same solution under $\boldsymbol{x} \sim f_2$. From the previous discussion, we know that $p^{-1}Q(\boldsymbol{x})$ is non-degenerate. Recall (18) we have

$$|R_{HRQDA} - R_{QDA}| \leq \frac{1}{2}\mathbb{P}_{x \sim f_1}\left\{0 \leq \frac{1}{p}Q(\boldsymbol{x}) < \frac{1}{p}M(\boldsymbol{x})\right\} + \frac{1}{2}\mathbb{P}_{x \sim f_2}\left\{0 \leq \frac{1}{p}Q(\boldsymbol{x}) < \frac{1}{p}M(\boldsymbol{x})\right\} \tag{20}$$

$$= O_p\{\lambda_n^{1-q/2}s_0^{1/2}(p) + \lambda_n^{1-q}s_0(p)\}.$$

$\square$

***Proof of Theorem 6***. From the proof of Theorem 7 in Liu et al. (2024) we can find that

$$T_{SUM2} = \frac{2}{n(n-1)}\sum\sum_{i<j}\tilde{\boldsymbol{U}}_i^\top \tilde{\boldsymbol{U}}_j + \zeta_1^2 \boldsymbol{\mu}^\top \mathbf{D}^{-1}\boldsymbol{\mu} + o_p(\sigma_n),$$

with $\sigma_n^2 = 2/\{n(n-1)p\} + o(n^{-3})$, and

$$n^{1/2}\mathbf{D}^{-1/2}\hat{\boldsymbol{\mu}} = n^{-1/2}\zeta_1^{-1}\sum_{i=1}^n \tilde{\boldsymbol{U}}_i + n^{1/2}\mathbf{D}^{-1/2}\boldsymbol{\mu} + C_n,$$

where $\tilde{\boldsymbol{U}}_i := U(\mathbf{D}^{-1/2}\boldsymbol{\Sigma}^{1/2}\boldsymbol{U}_i) = (\mathbf{R}^{1/2}\boldsymbol{U}_i)/\|\mathbf{R}^{1/2}\boldsymbol{U}_i\|$, $\mathbf{R} = \mathbf{D}^{-1/2}\boldsymbol{\Sigma}\mathbf{D}^{-1/2}$ Thus, we have

$$\frac{T_{SUM2}}{\sigma_n} = \frac{p}{n\sqrt{2\operatorname{tr}(\mathbf{R}^2)}}\left(\left\|\sum_{i=1}^n \tilde{\boldsymbol{U}}_i\right\|^2 - n\right) + O(1) = \frac{\|p^{1/2}n^{-1/2}\sum_{i=1}^n \tilde{\boldsymbol{U}}_i\|^2 - p}{\sqrt{2\operatorname{tr}(\mathbf{R}^2)}} + O(1),$$

and

$$T_{MAX2} - 2\log p + \log\log p = \left\|\sqrt{\frac{p}{n}}\sum_{i=1}^n \tilde{\boldsymbol{U}}_i\right\|_\infty^2 - 2\log p + \log\log p.$$

Notice that

$$p^{1/2}n^{-1/2}\sum_{i=1}^n \tilde{\boldsymbol{U}}_i = p^{1/2}n^{-1/2}\sum_{i=1}^n \mathbf{R}^{1/2}\boldsymbol{U}_i + p^{1/2}n^{-1/2}\sum_{i=1}^n \mathbf{R}^{1/2}\boldsymbol{U}_i(1/\|\mathbf{R}^{1/2}\boldsymbol{U}_i\| - 1).$$

Denote $v_i = 1/\|\mathbf{R}^{1/2}\boldsymbol{U}_i\| - 1$, $\operatorname{Var}(v_i) = \sigma_v^2$. We have

$$\left\|\sqrt{\frac{p}{n}}\sum_{i=1}^n v_i\mathbf{R}^{1/2}\boldsymbol{U}_i\right\|_\infty = \sqrt{\frac{p}{n}}\max_{i\leq j\leq p}\left|\sum_{i=1}^n v_iU_{ij}\right| \leq \frac{1}{\sqrt{n}}\sum_{i=1}^n |v_i|\max_{i\leq j\leq p}\sqrt{p}|U_{ij}| = O_p(\sigma_v\log p).$$

For a random variable $X$, denote $X^\star = (X - \mathbb{E}X)/\sqrt{\operatorname{Var}(X)}$. Thus the original proposition is equivalent to proving that $\|\sum_{i=1}^p \boldsymbol{U}_i\|_\infty^\star$ is asymptotically independent with $\|\mathbf{R}^{1/2}\sum_{i=1}^p \boldsymbol{U}_i\|^\star$ and $\|\sum_{i=1}^p \boldsymbol{U}_i\|^\star$ is asymptotically independent with $\|\mathbf{R}^{1/2}\sum_{i=1}^p \boldsymbol{U}_i\|_\infty^\star$. Then for any sequence $\eta_{n,p} \to \infty$ and any $t \in \mathbb{R}^p$

$$\mathbb{P}\left(\sqrt{\frac{p}{n}}\sum_{i=1}^n \tilde{\boldsymbol{U}}_i \leq t\right) = \mathbb{P}\left(\sqrt{\frac{p}{n}}\sum_{i=1}^n \mathbf{R}^{1/2}\boldsymbol{U}_i + \sqrt{\frac{p}{n}}\sum_{i=1}^n v_i\mathbf{R}^{1/2}\boldsymbol{U}_i \leq t\right)$$

$$\leq \mathbb{P}\left(\sqrt{\frac{p}{n}}\sum_{i=1}^n \mathbf{R}^{1/2}\boldsymbol{U}_i \leq t + \eta_{n,p}\sigma_v\log p\right)$$

$$+ \mathbb{P}\left(\left\|\sqrt{\frac{p}{n}}\sum_{i=1}^n v_i\mathbf{R}^{1/2}\boldsymbol{U}_i\right\|_\infty > \eta_{n,p}\sigma_v\log p\right)$$

$$\leq \mathbb{P}\left(\sqrt{\frac{p}{n}}\sum_{i=1}^n \mathbf{R}^{1/2}\boldsymbol{U}_i \leq t\right) + o(1).$$

Similarly, we have $\mathbb{P}\left(\sqrt{\frac{p}{n}}\sum_{i=1}^n \tilde{\boldsymbol{U}}_i \leq t\right) \geq \mathbb{P}\left(\sqrt{\frac{p}{n}}\sum_{i=1}^n \mathbf{R}^{1/2}\boldsymbol{U}_i \leq t\right) + o(1)$. We have

$$\sup_{t\in\mathbb{R}^p}\left|\mathbb{P}\left(\sqrt{\frac{p}{n}}\sum_{i=1}^n \tilde{\boldsymbol{U}}_i \leq t\right) - \mathbb{P}\left(\sqrt{\frac{p}{n}}\sum_{i=1}^n \mathbf{R}^{1/2}\boldsymbol{U}_i \leq t\right)\right| \to 0.$$

Further,

$$\sup_{A\in\mathcal{A}^{si}}\left|\mathbb{P}\left(\sqrt{\frac{p}{n}}\sum_{i=1}^n \tilde{\boldsymbol{U}}_i \in A\right) - \mathbb{P}\left(\sqrt{\frac{p}{n}}\sum_{i=1}^n \mathbf{R}^{1/2}\boldsymbol{U}_i \in A\right)\right| \to 0.$$

From the proof of Lemma 2 we have

$$\sup_{A\in\mathcal{A}^{si}}\left|\mathbb{P}\left(\sqrt{\frac{p}{n}}\sum_{i=1}^n \boldsymbol{U}_i \in A\right) - \mathbb{P}\left(\boldsymbol{Z} \in A\right)\right| \to 0,$$

where $\boldsymbol{Z} \sim N(0, \mathbf{I}_p)$. Thus

$$\sup_{A_1,A_2\in\mathcal{A}^{si}}\left|\mathbb{P}\left(\sqrt{\frac{p}{n}}\sum_{i=1}^n \mathbf{R}^{1/2}\boldsymbol{U}_i \in A_1, \sqrt{\frac{p}{n}}\sum_{i=1}^n \boldsymbol{U}_i \in A_2\right) - \mathbb{P}\left(\mathbf{R}^{1/2}\boldsymbol{Z} \in A_1, \boldsymbol{Z} \in A_2\right)\right| \to 0,$$

Thus the original proposition is equivalent to proving that $\|\boldsymbol{Z}\|_\infty^\star$ is asymptotically independent with $\|\mathbf{R}^{1/2}\boldsymbol{Z}\|^\star$ and $\|\mathbf{R}^{1/2}\boldsymbol{Z}\|_\infty^\star$ is asymptotically independent with $\|\boldsymbol{Z}\|^\star$. From Theorem 2.2 in Chen et al. (2024) we have $\|\boldsymbol{Z}\|_\infty^\star$ is

asymptotically independent with $\|\mathbf{R}^{1/2}\mathbf{Z}\|^\star$. Consider that $\mathbf{R}^{1/2}\mathbf{Z} \sim N(\mathbf{0}, \mathbf{R})$. Next we prove $\|\mathbf{Y}\|_\infty^\star$ is asymptotically independent with $\|\mathbf{R}^{-1/2}\mathbf{Y}\|^\star$ where $\mathbf{Y} = \mathbf{R}^{1/2}\mathbf{Z} \sim N(\mathbf{0}, \mathbf{R})$.

Define $\mathbf{Y} = (\mathbf{Y}_1^\top, \mathbf{Y}_2^\top)^\top \in \mathbb{R}^p$ where $\mathbf{Y}_1 = (Y_1, \ldots, Y_d)^\top$ and $\mathbf{Y}_2 = (Y_{d+1}, \ldots, Y_p)^\top$. And

$$\mathbf{R} = \begin{pmatrix} \mathbf{R}_1 & \mathbf{R}_{12} \\ \mathbf{R}_{21} & \mathbf{R}_2 \end{pmatrix}, \quad \mathbf{R}^{-1} := \mathbf{P} = \begin{pmatrix} \mathbf{P}_1 & \mathbf{P}_{12} \\ \mathbf{P}_{21} & \mathbf{P}_2 \end{pmatrix}.$$

$$\mathbf{K} := \begin{pmatrix} \mathbf{K}_1 & \mathbf{K}_{12} \\ \mathbf{K}_{21} & \mathbf{K}_2 \end{pmatrix} = \begin{pmatrix} \mathbf{R}_1^{1/2}\mathbf{P}_1\mathbf{R}_1^{1/2} & \mathbf{R}_1^{1/2}\mathbf{P}_{12}\mathbf{R}_2^{1/2} \\ \mathbf{R}_2^{1/2}\mathbf{P}_{21}\mathbf{R}_1^{1/2} & \mathbf{R}_2^{1/2}\mathbf{P}_2\mathbf{R}_2^{1/2} \end{pmatrix}.$$

So,

$$\mathbf{Y}^\top \mathbf{P} \mathbf{Y} = \mathbf{Y}_1^\top \mathbf{P}_1 \mathbf{Y}_1 + 2\mathbf{Y}_1^\top \mathbf{P}_{12} \mathbf{Y}_2 + \mathbf{Y}_2^\top \mathbf{P}_2 \mathbf{Y}_2.$$

For $\epsilon > 0$, set $z_i$ are i.i.d. Gaussian random variables. Define $\mathbf{z}_1 = (z_1, \cdots, z_d)^\top \in \mathbb{R}^d$, $\mathbf{z}_1 = (z_{d+1}, \cdots, z_p)^\top \in \mathbb{R}^{p-d}$. Then there exist $\eta > 0$ and $K > 0$ such that $\mathbb{E}\{\exp(\eta z_i^2)\} \leq K$. According to Assumptions of Theorem 7 in Liu et al. (2024), we can get $\lambda_{\max}(\mathbf{K}_1) \leq \lambda_{\max}(\mathbf{K}) < c_1$ for a constant $c_1 > 0$.

$$\mathbb{P}(\mathbf{Y}_1^\top \mathbf{P}_1 \mathbf{Y}_1 > \sqrt{2p}\epsilon) \leq \mathbb{P}(c_1 \mathbf{z}_1^\top \mathbf{z}_1 > \sqrt{2p}\epsilon)$$

$$= \mathbb{P}\left(\eta \sum_{i=1}^d z_i^2 > \sqrt{2p}\epsilon c_1^{-1}\eta\epsilon\right)$$

$$\leq \exp(-\sqrt{2p}\epsilon c_1^{-1}\eta\epsilon)\mathbb{E}(e^{\eta \sum_{i=1}^d z_i^2})$$

$$= \exp(-\sqrt{2p}\epsilon c_1^{-1}\eta\epsilon)\{E(e^{\eta z_i^2})\}^d$$

$$\leq K^d \exp(-\sqrt{2p}\epsilon c_1^{-1}\eta\epsilon).$$

Define $\mathbf{K} = \mathbf{O}^\top \mathbf{\Lambda} \mathbf{O}$ where $\mathbf{O} = (q_{ij})_{1 \leq i,j \leq p}$ is an orthogonal matrix and $\mathbf{\Lambda} = \mathrm{diag}\{\lambda_1, \ldots, \lambda_p\}$, $\lambda_i, i = 1, \ldots, p$ are the eigenvalues of $\mathbf{K}$. Note that $\sum_{1 \leq j \leq p} k_{ij}^2$ is the $i$-th diagonal element of $\mathbf{K}^2 = \mathbf{O}^\top \mathbf{\Lambda}^2 \mathbf{O}$. We have $\sum_{1 \leq j \leq p} k_{ij}^2 = \sum_{l=1}^p q_{li}^2 \lambda_l^2 \leq c_1^2$. Next, define $\theta = \sqrt{(2\eta)/(dc_1^2)}$. We have

$$\mathbb{P}(\mathbf{Y}_1^\top \mathbf{P}_{12} \mathbf{Y}_2 \geq \sqrt{2p}\epsilon) \leq \exp(-\sqrt{2p}\theta\epsilon)\mathbb{E}(\exp(\theta \mathbf{z}_1^\top \mathbf{K}_{12} \mathbf{z}_2)$$

$$= \exp(-\sqrt{2p}\theta\epsilon)\mathbb{E}(e^{\theta \sum_{i=1}^d \sum_{j=d+1}^p k_{ij}z_i z_j})$$

$$= \exp(-\sqrt{2p}\theta\epsilon)\mathbb{E}\{\mathbb{E}(e^{\theta \sum_{j=d+1}^p (\sum_{i=1}^d k_{ij}z_i)z_j} | \mathbf{z}_1)\}$$

$$= \exp(-\sqrt{2p}\theta\epsilon)\mathbb{E}\left[\prod_{j=d+1}^p \mathbb{E}\{e^{(\theta \sum_{i=1}^d k_{ij}z_i)z_j} | \mathbf{z}_1\}\right]$$

$$\leq \exp(-\sqrt{2p}\theta\epsilon)\mathbb{E}\left[\prod_{j=d+1}^p \exp\left\{\frac{\theta^2}{2}\left(\sum_{i=1}^d k_{ij}z_i\right)^2\right\}\right]$$

$$= \exp(-\sqrt{2p}\theta\epsilon)\mathbb{E}\left[\exp\left\{\frac{\theta^2}{2}\sum_{j=d+1}^p \left(\sum_{i=1}^d k_{ij}z_i\right)^2\right\}\right]$$

$$\leq \exp(-\sqrt{2p}\theta\epsilon)\mathbb{E}\left\{\exp\left(\frac{d\theta^2}{2}\sum_{j=d+1}^p \sum_{i=1}^d k_{ij}^2 z_i^2\right)\right\}$$

$$\leq \exp(-\sqrt{2p}\theta\epsilon)\mathbb{E}\left\{\exp\left(\frac{dc_1^2\theta^2}{2}\sum_{i=1}^d z_i^2\right)\right\}$$

$$= \exp(-\sqrt{2p}\theta\epsilon)\mathbb{E}\left\{\exp\left(\eta \sum_{i=1}^d z_i^2\right)\right\}$$

$$\leq K^d \exp(-\sqrt{2p}\theta\epsilon).$$

So

$$\mathbb{P}(\boldsymbol{Y}_1^\top \mathbf{P}_{12}\boldsymbol{Y}_2 \geq \sqrt{2p}\epsilon) \leq K^d \exp\left(-\sqrt{\frac{4\eta}{dc_1^4}}\epsilon p^{1/2}\right).$$

Similarly, we also can prove that

$$\mathbb{P}\{(-\boldsymbol{Y}_1)^\top \mathbf{P}_{12}\boldsymbol{Y}_2 \geq \sqrt{2p}\epsilon\} \leq K^d \exp\left(-\sqrt{\frac{4\eta}{dc_1^4}}\epsilon p^{1/2}\right).$$

Let $\Theta_p = \boldsymbol{Y}_1^\top \mathbf{P}_1\boldsymbol{Y}_1 + 2\boldsymbol{Y}_1^\top \mathbf{P}_{12}\boldsymbol{Y}_2$.

$$\begin{aligned}
\mathbb{P}(|\Theta_p| > \sqrt{2p}\epsilon) \leq &\mathbb{P}(\boldsymbol{Y}_1^\top \mathbf{P}_1\boldsymbol{Y}_1 > \sqrt{2p}\epsilon/2) + \mathbb{P}(|\boldsymbol{Y}_1^\top \mathbf{P}_{12}\boldsymbol{Y}_2| > \sqrt{2p}\epsilon/4) \\
\leq &\mathbb{P}(\boldsymbol{Y}_1^\top \mathbf{P}_1\boldsymbol{Y}_1 > \sqrt{2p}\epsilon/2) + \mathbb{P}(\boldsymbol{Y}_1^\top \mathbf{P}_{12}\boldsymbol{Y}_2 > \sqrt{2p}\epsilon/8) \\
&+ \mathbb{P}(-\boldsymbol{Y}_1^\top \mathbf{P}_{12}\boldsymbol{Y}_2 > \sqrt{2p}\epsilon/8).
\end{aligned}$$

Denote $A_p(x) = \left\{\frac{\boldsymbol{Y}\mathbf{R}^{-1}\boldsymbol{Y}-p}{\sqrt{2p}}\right\} \leq x$, $B_i = \{|Y_1| \geq \sqrt{2\log p - \log\log p}\}$, so there exist a constant $c_\epsilon > 0$

$$\mathbb{P}(|\Theta_p| > \sqrt{2p}\epsilon) \leq K^d \exp(-c_\epsilon p^{1/2}),$$
$$\mathbb{P}(A_p(x)B_1\cdots B_d)$$

$$= \mathbb{P}\left(\frac{\boldsymbol{Y}_2^\top \mathbf{P}_2\boldsymbol{Y}_2 - p + \Theta_p}{\sqrt{2p}} \leq x, B_1\cdots B_d\right)$$

$$\leq \mathbb{P}\left(\frac{\boldsymbol{Y}_2^\top \mathbf{P}_2\boldsymbol{Y}_2 - p + \Theta_p}{\sqrt{2p}} \leq x, |\Theta_p| \leq \sqrt{2p}\epsilon, B_1\cdots B_d\right) + \mathbb{P}(|\Theta_p| > \sqrt{2p}\epsilon)$$

$$\leq \mathbb{P}\left(\frac{\boldsymbol{Y}_2^\top \mathbf{P}_2\boldsymbol{Y}_2 - p}{\sqrt{2p}} \leq x + \epsilon, B_1\cdots B_d\right) + K^d \exp(-c_\epsilon p^{1/2})$$

$$= \mathbb{P}\left(\frac{\boldsymbol{Y}_2^\top \mathbf{P}_2\boldsymbol{Y}_2 - p}{\sqrt{2p}} \leq x + \epsilon\right) P(B_1\cdots B_d) + K^d \exp(-c_\epsilon p^{1/2})$$

$$\leq \left\{\mathbb{P}\left(\frac{\boldsymbol{Y}_2^\top \mathbf{P}_2\boldsymbol{Y}_2 - p}{\sqrt{2p}} \leq x + \epsilon, |\Theta_p| \leq \sqrt{2p}\epsilon\right) + \mathbb{P}(|\Theta_p| > \sqrt{2p}\epsilon)\right\} P(B_1\cdots B_d)$$

$$+ K^d \exp(-c_\epsilon p^{1/2})$$

$$\leq \mathbb{P}\left(\frac{\boldsymbol{Y}_2^\top \mathbf{P}_2\boldsymbol{Y}_2 - p + \Theta_p}{\sqrt{2p}} \leq x + 2\epsilon\right) \mathbb{P}(B_1\cdots B_d) + 2K^d \exp(-c_\epsilon p^{1/2})$$

$$= \mathbb{P}\{A_p(x + 2\epsilon)\}\mathbb{P}(B_1\cdots B_d) + 2K^d \exp(-c_\epsilon p^{1/2}).$$

Similarly, we can prove that

$$\mathbb{P}(A_p(x)B_1\cdots B_d) \geq \mathbb{P}\{A_p(x - 2\epsilon)\}\mathbb{P}(B_1\cdots B_d) - 2K^d \exp(-c_\epsilon p^{1/2}).$$

So, we have

$$|\mathbb{P}(A_p(x)B_1\cdots B_d) - \mathbb{P}\{A_p(x)\} \cdot \mathbb{P}(B_1\cdots B_d)| \leq \Delta_{p,\epsilon} \cdot \mathbb{P}(B_1\cdots B_d) + 2K^d \exp(-c_\epsilon p^{1/2}),$$

where

$$\begin{aligned}
\Delta_{p,\epsilon} &= |\mathbb{P}\{A_p(x)\} - \mathbb{P}(A_p(x + 2\epsilon))| + |\mathbb{P}\{A_p(x)\} - \mathbb{P}\{A_p(x - 2\epsilon)\}| \\
&= \mathbb{P}\{A_p(x + 2\epsilon)\} - \mathbb{P}\{A_p(x - 2\epsilon)\}.
\end{aligned}$$

Obviously, the equation discussed above holds for all $i_1, \ldots, i_d$. Thus,

$$\begin{aligned}
&\sum_{1 \leq i_1 < \cdots < i_d \leq p} |\mathbb{P}(A_p(x)B_{i_1}\cdots B_{i_d}) - \mathbb{P}\{A_p(x)\} \cdot \mathbb{P}(B_{i_1}\cdots B_{i_d})| \\
&\leq \sum_{1 \leq i_1 < \cdots < i_d \leq p} \{\Delta_{p,\epsilon} \cdot \mathbb{P}(B_{i_1}\cdots B_{i_d}) + 2K^d \exp(-c_\epsilon p^{1/2})\} \\
&\leq \Delta_{p,\epsilon} \cdot H(d, p) + \binom{p}{d} \cdot 2K^d \exp(-c_\epsilon p^{1/2}).
\end{aligned}$$

Because $\mathbb{P}\{A_p(x)\} \to \Phi(x)$ as $p \to \infty$. So $\Delta_{p,\epsilon} \to \Phi(x+2\epsilon) - \Phi(x-2\epsilon)$. By letting $\epsilon \to 0$, we have $\Delta_{p,\epsilon} \to 0$. By Lemma 9 as $p \to \infty$ we have

$$\sum_{1 \leq i_1 < \cdots < i_d \leq p} |\mathbb{P}(A_p(x)B_{i_1} \cdots B_{i_d}) - \mathbb{P}\{A_p(x)\} \cdot \mathbb{P}(B_{i_1} \cdots B_{i_d})| \to 0.$$

Then, repeat the procedure in proof of Theorem 2.2 in Chen et al. (2024) we have

$$\limsup_{p \to \infty} \mathbb{P}\left(\|\mathbf{R}^{-1/2}\boldsymbol{Y}\|^{2*} \leq x, \|\boldsymbol{Y}\|_{\infty}^{*2} \leq y\right) = \limsup_{p \to \infty} \mathbb{P}\left(\frac{\boldsymbol{Y}^{\top}\mathbf{R}^{-1}\boldsymbol{Y} - p}{\sqrt{2p}} \leq x, \max_{1 \leq i \leq p} |Y_i| > l_p\right)$$
$$\leq \Phi(x) \cdot \{1 - F(y)\} + \lim_{p \to \infty} H(p, 2k+1),$$

$$\liminf_{p \to \infty} \mathbb{P}\left(\|\mathbf{R}^{-1/2}\boldsymbol{Y}\|^{2*} \leq x, \|\boldsymbol{Y}\|_{\infty}^{*2} \leq y\right) = \liminf_{p \to \infty} \mathbb{P}\left(\frac{\boldsymbol{Y}^{\top}\mathbf{R}^{-1}\boldsymbol{Y} - p}{\sqrt{2p}} \leq x, \max_{1 \leq i \leq p} |Y_i| > l_p\right)$$
$$\leq \Phi(x) \cdot \{1 - F(y)\} - \lim_{p \to \infty} H(p, 2k+1).$$

Then we can get $\|\boldsymbol{Y}\|_{\infty}^{\star}$ is asymptotically independent with $\|\mathbf{R}^{-1/2}\boldsymbol{Y}\|^{\star}$ by sending $p \to \infty$ and then sending $k \to \infty$. $\qquad\square$

