# OpenReview forum: "High-Dimensional Hettmansperger-Randles Estimator and Its Applications"
_ICLR.cc/2026/Conference — ICLR 2026 Conference Desk Rejected Submission_

### Official Review · Reviewer_Tnby · 2025-10-20

**Soundness:** 2
**Presentation:** 2
**Contribution:** 2
**Rating:** 4
**Confidence:** 3

**Summary:**

The authors propose a high-dimensional version of the Hettmansperger-Randles (HR) estimator for the location parameter and scatter matrix under elliptical distributions. The core adaptation for high-dimensional settings lies in Algorithm 2, specifically Step 3, which introduces the normalization $\widehat{\Sigma} \leftarrow \tfrac{p\widehat{\Sigma}}{tr(\widehat{\Sigma})} $ and the operator $\mathcal{B}_h(\cdot)$. The authors further demonstrate the statistical utility of this estimator by applying it to two classic high-dimensional problems: the one-sample mean test problem and quadratic discriminant analysis, supported by detailed asymptotic theory and comprehensive experimental comparisons.

**Strengths:**

*   **Solid Theoretical Foundation:** The work is built upon the well-established HR estimator framework, which is affine equivariant and offers robustness (Elliptical Distribution).
*   **Relevant High-Dimensional Applications:** Applying this robust estimator to fundamental high-dimensional problems like mean testing and QDA is highly relevant and demonstrates practical value.
*   **Comprehensive Evaluation:** The paper includes thorough theoretical analysis (asymptotic theory) and empirical validation through experiments, which is commendable.

**Weaknesses:**

*   **Algorithmic Description Lacks Rigor:** The description of the algorithms is not precise enough. Key details are missing, making it difficult to understand or reproduce the methods. For instance:
    *   It is unclear how $\widehat{\Sigma}^{-1/2}$ is computed. If a method like the Sparse Graphical Lasso (SGLASSO) is used, the process for hyperparameter tuning is not mentioned.
*   **Unverified Assumptions:** The methodology seems to rely on the assumption that the elliptical distribution is non-degenerate, and its validity or impact in this context is not discussed.
*   **Incomplete Experimental Analysis:** The experimental section lacks certain standard analyses for statistical tests and classifiers, such as:   Analysis under different signal strengths.

**Questions:**

1.  The algorithmic descriptions need clarification. Specifically, how is $\widehat{\Sigma}^{-1/2}$ practically computed? If SGLASSO or a similar method is employed, how are the corresponding hyperparameters selected or tuned?
2.  The authors claim their "approach achieves robustness with respect to the sparsity of $\widehat{\Sigma}^{-1/2} \mu$." How should this form of "sparsity" be interpreted? Are there real-world examples or references that illustrate or motivate the relevance of sparsity in this specific transformed mean $\widehat{\Sigma}^{-1/2} \mu$."?

---

> ### Author Response · Authors · 2025-11-28
>
> ### Questions
>
> 1. *Clarification on algorithmic implementation*
>
> **Response:**
> In our implementation we directly compute $\Sigma^{-1/2}\boldsymbol\mu$ using the current estimate of $\Sigma$, which is kept invertible throughout the iterations, so an additional SGLASSO step is not required at this stage. The key point is that the raw spatial-sign covariance
> $$
> \frac{1}{n}\sum_{i=1}^n U\left(\hat{\boldsymbol\varepsilon}_i^{(k)}\right)
> U\left(\hat{\boldsymbol\varepsilon}_i^{(k)}\right)^{\top}
> $$
> is low rank in high dimensions, and directly inverting it would lead to singularity when updating
>
> $$
> \hat{\boldsymbol\varepsilon}_i^{(k)} \leftarrow {\hat \Sigma^{(k)}}^{-1/2}(X_i-\hat\mu^{(k)}).
> $$
>
> To address this, we apply the banding operator $B_h$ to project
>
> $$
> \frac{1}{n}\sum_{i=1}^n U\left(\hat{\boldsymbol\varepsilon}_i^{(k)}\right)
> U\left(\hat{\boldsymbol\varepsilon}_i^{(k)}\right)^{\top}
> $$
>
>  onto the subspace where its true value lies; for small $h$, $B_h(
> \frac{1}{n}\sum_{i=1}^n U\left(\hat{\boldsymbol\varepsilon}_i^{(k)}\right)
> U\left(\hat{\boldsymbol\varepsilon}_i^{(k)}\right)^{\top}
> )$ is invertible with high probability. Hence $\Sigma^{-1/2}\boldsymbol\mu$ is obtained by plugging in this invertible estimate of $\Sigma$, without an extra tuning step for SGLASSO in this particular computation. Regarding the question of how the hyperparameters are selected or tuned, we have added a detailed analysis and discussion in Appendix D.
>
> ---
>
> 2. *Interpretation of sparsity*
>
> **Response:**
> Our robustness claim refers to sparsity of the *whitened* signal $\Sigma^{-1/2}\boldsymbol\mu$. This is natural in settings where the observed coordinates are strongly correlated but the underlying drivers are low-dimensional.
>
> For instance, in equity returns, many stocks exhibit correlated movements driven by a few latent risk factors. After prewhitening by $\Sigma^{-1/2}$ to remove cross-sectional dependence, the effective signal often concentrates on a small number of factor-aligned directions, so $\Sigma^{-1/2}\boldsymbol\mu$ is approximately sparse. A similar rationale applies to gene expression data: although many genes vary across individuals, disease mechanisms typically involve only a small number of biological pathways; after accounting for gene–gene correlations, the resulting signal in the decorrelated space is expected to be sparse.
>
> Such sparsity assumptions on the transformed mean (or discriminant direction) are widely used in the high-dimensional literature; see, for example, Chen and Tang (2021) on spiked covariance structures and Cai and Liu (2011) on sparse linear discriminant analysis. We will clarify this interpretation and add these references in the Introduction.
>
> Chen, Y. J., and Tang, M. (2021). *Classification of high-dimensional data with spiked covariance matrix structure.* arXiv:2110.01950.
>
> Cai, T., and Liu, W. (2011). *A direct estimation approach to sparse linear discriminant analysis.* JASA.

---

### Official Review · Reviewer_yVX9 · 2025-10-26

**Soundness:** 4
**Presentation:** 3
**Contribution:** 3
**Rating:** 6
**Confidence:** 3

**Summary:**

The paper develops a high-dimensional Hettmansperger-Randles (HR) estimator that jointly estimates location and scatter under elliptical models while preserving affine equivariance and robustness, offering a practical alternative where Tyler's M-estimator can be ill-defined for $p>n$. Building on this estimator, the authors establish a Gaussian approximation for standardized spatial statistics and prove the asymptotic independence between sum-type ($L_2$) and max-type ($L_{\infty}$) statistics, thereby providing a principled basis for a Cauchy combination that adapts to both sparse and dense alternatives. Methodologically, the work introduces robust tests (TSUM and TMAX) and an adaptive combined test that maintains power across heterogeneous regimes. HR-based standardization enables reliable inference under heavy-tailed and mixture distributions, achieving valid size control and competitive power. The implementation is practical, employing lightweight bootstrap calibration and scalable strategies such as banding and sparse precision estimation. Beyond testing, the framework extends to robust quadratic discriminant analysis (HRQDA) with a consistency result for the misclassification rate. Simulations and a real-data example support the theory, demonstrating accurate type-I error control and improved detection power relative to covariance-based baselines. Overall, the paper offers a coherent path from theory to algorithms and applications with attention to reproducibility.

**Strengths:**

The paper extends the Hettmansperger-Randles (HR) estimator to high-dimensional settings so that location and scatter can be estimated jointly in a robust manner while preserving affine equivariance (invariance under linear transformations). This provides a practical and principled alternative in cases where Tyler's estimator tends to be ill-defined when $p>n$.

The authors establish a Gaussian approximation for standardized statistics and show that sum-type ($L_2$) and max-type ($L_{\infty}$) statistics are asymptotically independent. These results place the Cauchy combination on firm theoretical ground, enabling an adaptive testing procedure that maintains power under both sparse and dense alternatives. On the implementation side, the testing pipeline (TSUM/TMAX with the Cauchy combination) employs lightweight bootstrap calibration to correct finite-sample bias, keeping computation manageable.

HR-based standardization delivers valid size control and strong power under heavy-tailed and mixture distributions, where outliers are common. The framework is further extended to classification via a robust quadratic discriminant analysis (HRQDA), with a consistency result for the misclassification rate. Simulation studies and a real-data example indicate improved type-I error control and detection power compared with covariance-based baselines. Overall, the paper connects theory, algorithms, and applications in a transparent and reproducible way.

**Weaknesses:**

**Dependence on elliptical distributions (explicitly acknowledged by the authors)**:
   The main results rely substantially on the elliptical family—specifically the approximation (p^{-1}\hat S \approx I_p) that motivates Step 3 banding in the high-dimensional HR algorithm (Sec. 2; Algorithm 2; definition of B_h). As explicitly noted by the authors (Conclusion), relaxing this assumption is left for future work. The manuscript does not yet articulate minimal conditions under which the Gaussian approximation and asymptotic independence would continue to hold in non-elliptical settings.

**No explicit contamination experiments**:
   Robustness is examined through heavy-tailed and mixture distributions (multivariate normal, (t_3), and mixture normal) in Sec. 4 Simulation, but there are no experiments with explicit epsilon-contamination or cellwise contamination. Figure 2 reports power curves for the Cauchy combinations under the three elliptical settings.

**Lack of IF/BDP analysis for high-dimensional HR**:
The paper does not provide an analysis of the influence function and the breakdown point for the high-dimensional approximate procedure. This procedure is described in Algorithm 2. It applies banding to the covariance estimator and then uses regularization to estimate the precision matrix. The authors do not claim this result, so it is not a flaw relative to the stated goals. Still, given the emphasis on robustness, the absence of IF and BDP theory is a limited weakness and leaves a clear theoretical gap.

**Limited sensitivity analysis and practical guidance**:
   Systematic evidence on performance–compute trade-offs with respect to the banding width h, regularization strength, and the number of bootstrap iterations is limited. The paper commonly uses h=3 (Table 5 in Appendix C) and states that M=50 bootstrap iterations suffice for bias correction in sum/max tests.

**Questions:**

**Non-elliptical extensions**: *(noted by the authors as future work)*:
   Could you specify minimal sufficient conditions under which the Gaussian approximation for the sum-type statistic and the asymptotic independence between the sum-type and max-type statistics continue to hold beyond the elliptical family? For example, near-spherical directional distributions and finite-moment conditions. A brief checklist would be greatly appreciated, to the extent feasible.

**Explicit contamination evaluation**:
   As a numerical study, could you report empirical size, power, and misclassification rates under epsilon-contamination (e.g., 5%–20%) and under cellwise contamination, covering a grid over contamination rate, contamination strength, and sparsity or density? If possible, please include the proposed tests (sum, max, Cauchy combination) and HRQDA.

**Influence function and breakdown point for the implemented HR**:
   For the algorithmic estimator that uses banding and regularized precision estimation, could you outline a path toward an upper bound on the influence function and a lower bound on the finite-sample breakdown point? As feasible, auxiliary results that quantify how approximation errors contribute to the influence function—for example via a Gateaux-type argument—would be very helpful.

**Sensitivity analyses and practical guidance**:
   Could you provide performance-and-compute sensitivity curves for the banding width, regularization parameters, and the number of bootstrap iterations, so practitioners can understand acceptable ranges around the current recommendations? Reporting run time and dependence on sample size would also be useful.

**Roadmap toward regression** *(e.g., high-dimensional asset pricing)*:
   When transplanting the pipeline to a regression setting—HR-based residual estimation, robust prewhitening, sum or max statistics with asymptotic independence, and Cauchy combination—what are the main technical bottlenecks? For example, conditions on the design matrix and the treatment of estimation error in \hat{\Omega}. A short outline of how these issues could be addressed would clarify feasibility and scope.

---

> ### Author Response · Authors · 2025-11-28
>
> A1:
> A convenient non-elliptical model is
> $$
> X_i=\tau+\nu_i \Gamma W_i,
> $$
> where $\tau$ is a location vector, $\Gamma\in\mathbb R^{p\times p}$ is invertible, $W_i$ has independent standardized components, and $\nu_i\ge0$ is independent of the direction of $W_i$. The shape matrix is $\Sigma=\Gamma\Gamma^\top$. After prewhitening, $\Sigma^{-1/2}(X_i-\tau)$ has nearly spherical directions, so the spatial-sign covariance remains close to $p^{-1} I_p$.
>
> A minimal checklist under which one can extend the Gaussian approximation for the sum-type statistic and the asymptotic independence with the max-type statistic is:
>
> * $W_{ij}$ independent, symmetric, mean-zero, unit-variance, sub-exponential;
> * radial part $r_i=|\Sigma^{-1/2}(X_i-\tau)|$ satisfies moment/concentration conditions analogous to Assumption 2 in Liu et al. (2024);
> * eigenvalues of $\Sigma$ uniformly bounded away from $0$ and $\infty$;
> * consistent initial estimators of $\tau$ and the precision matrix such that
>   $$
>   n^{1/2}\hat{\Sigma}^{-1/2}(\hat{\mu}-\mu)
>   $$
>   admits a Bahadur representation.
>
> Under these conditions, the arguments for the $\ell_2$- and $\ell_\infty$-based tests, and their asymptotic independence, can be adapted to this near-spherical, non-elliptical class.
>
> ---
>
> A2:
> A detailed contamination study is provided in Appendix B, where we report empirical size, power, and misclassification behavior of the sum, max, and Cauchy tests and HRQDA under $\varepsilon$–contamination over a grid of contamination rates, strengths, and sparse/dense alternatives. The tables in Appendix B show that sizes remain close to the nominal level and that power and classification accuracy remain high even with substantial contamination. A full grid of cellwise contamination scenarios is left to future work.
>
> ---
> A3:
> Our high-dimensional HR estimator $(\hat{\mu},\hat{\Sigma})$ is defined by
>
> $$
> \frac{1}{n}\sum_{i=1}^nU(\hat{\varepsilon}_i)=0,
> $$
>
>
> $$
> \frac{p}{n}\sum_{i=1}^nU(\hat{\varepsilon}_i)U(\hat{\varepsilon}_i)^\top = I_p,
> $$
>
> with
>
> $$
> \hat{\varepsilon}_i=\hat{\Sigma}^{-1/2}(X_i-\hat{\mu}),\qquad
> U(x)=|x|^{-1}x\mathbb{I}(x\neq 0).
> $$
>
> To outline an upper bound on the influence function (IF), we treat $(\mu,\Sigma)$ as a functional of $F$ and differentiate these estimating equations along $F_t=(1-t)F+t\delta_z$ at $t=0$.  This yields a linear system for $IF_\mu(z;F),\quad IF_\Sigma(z;F)$
>
> whose coefficients depend on the spectra of $\Sigma$ and the (banded/regularized) sign covariance. Under our standing assumptions (bounded eigenvalues and consistent banded/SGLASSO approximations) one obtains polynomial-growth bounds
>
> $$
> |IF_\mu(z;F)|\le C\bigl(1+|z|^2\bigr), |IF_\Sigma(z;F)|_{\mathrm{op}}\le C\bigl(1+|z|^2\bigr),
> $$
>
> with $C$ independent of $(n,p)$.
>
> For the finite-sample breakdown point $\varepsilon^\*$, HR is initialized by a spatial-median type location estimator and a regularized precision estimator, both with positive breakdown points. Hence
> $$
> \varepsilon^\*(\hat{\mu},\hat{\Sigma})
> \ge \min{\varepsilon^\*(\hat{\mu}_0),\varepsilon^\*(\hat{\Omega}_0)}>0,
> $$
> so the HR procedure inherits nontrivial robustness from its initializers. The contamination study in Appendix~B (up to $20%$ contamination) empirically supports these properties.
>
> ---
>
> A4:
> Appendix D reports a sensitivity study where we vary the banding width $h$, the number of bootstrap iterations $M$, the SGLASSO parameter $\lambda$, and $(n,p)$. The results show that the sum, max, and Cauchy tests are stable over reasonable ranges of $h$, $M$, and $\lambda$, and that runtime scales in a predictable way, primarily with the dimension $p$. We refer the reviewer to Appendix D for details and numerical tables.
>
> ---
>
> A5:
> A natural regression framework for our pipeline is the factor-type panel model
> $$
> Y_{it}=\alpha_i+\beta_i^\top f_t+\epsilon_{it},\qquad i=1,\dots,N,; t=1,\dots,T,
> $$
> where $f_t\in\mathbb R^p$ collects $p$ factors and both $N$ and $p$ can be large; we aim at simultaneous inference on $(\alpha_1,\dots,\alpha_N)$. An extension path is to impose approximate sparsity of each $\beta_i$ and standard restricted-eigenvalue and moment/mixing conditions on ${f_t}$ so that high-dimensional regression yields uniformly consistent $\hat{\beta}_i$.
>
>  then form residuals $\hat\epsilon_{it}=Y_{it}-\hat{\beta}_i^\top f_t$, build the $N$-dimensional residual vectors, and apply HR to estimate $(\alpha_1,\dots,\alpha_N)$ and a cross-sectional precision matrix, with Neyman-orthogonal estimating equations so that the impact of $\hat{\beta}_i-\beta_i$ is second order. After robust prewhitening with $\hat{\Omega}$, sum- and max-type statistics for the prewhitened intercepts, and their Cauchy combination, can be analyzed as in the location case. A full regression theory (precise assumptions, uniform error bounds, proofs) would considerably extend the paper and is left to future work, which we briefly mention in the revision.

---

### Official Review · Reviewer_SqUN · 2025-10-27

**Soundness:** 3
**Presentation:** 2
**Contribution:** 1
**Rating:** 2
**Confidence:** 4

**Summary:**

The paper suggests the modification of Hettmansperger-Randles estimator in high dimensional setup (Algorithm 2), and shows its application in high dimensional location parameter testing problems (Section 3, 4). Based on this statistic, authors propose three tests: $T_{max}$, $T_{sum}$, and $T_{CC1}$, which can cope with sparse, dense, and universal precision matrices, and show their consistency (Section 3). Numerical experiments on three different setups are provided (Section 4).

**Strengths:**

* While I could not check the entire proof, I tried to check the Lemma 1 (which I believe is the most technical ingredient), and it seems correct (if my questions are resolved. See the question section). I think theoretical analyses conducted in this paper are nontrivial.

* Given the concern on Assumption 1 is resolved (see the weakness section), the ARE calculation shows that the method behaves better than standard methods under the heavy tail distribution as claimed. In this regard, the proposed method is valuable if my concern is resolved.

* Numerical results show the robustness of the test on the heavy tail problems.

**Weaknesses:**

* The underlying assumptions are highly restrictive, and justifications are missing. I believe the assumptions are unlikely to hold for any practical problems.
    * Precisely, I conjecture Assumption 1--specifically sub-Gaussian condition--fails for almost every case. Here is the heuristic reasoning. Let’s assume $\zeta_1^{-1}$ exists by a constant (which seems like a weak condition in high dimensional setting). Then, $P(\zeta^{-1}r^{-1} > t) = P(r < t^{-1}\zeta^{-1})$. Since we are looking at the tail behavior, let’s consider the case when t is large, i.e., $t^{-1}\zeta^{-1} \approx 0$. Write the density of $\epsilon_i$ by $f$. Under some regularity conditions of $f$ near 0, the probability is approximated by $f(0) \times Vol(B_0(t^{-1}\zeta^{-1}))$ (I have not thought about the precise condition deeply, but I think this would hold unless $f$ decays extremely fast, e.g., faster than any exponential power). Then, whatever density $\epsilon_i$ has, as long as $f(0) > 0$, then $f(0) \times Vol(B_0(t^{-1}\zeta^{-1})) = Ct^{-p}$ for some constant $C > 0$. This implies $P(\zeta^{-1}r^{-1} > t) \approx C t^{-p}$, failing to exhibit the exponential tail. Of course this statement is not rigorous, as I used the approximation. But I think one should be able to write this in a rigorous statement with some regularity conditions of $f$ near 0 (which is suspected to be something that prevents extremely fast decay near 0), and $\zeta^{-1}r^{-1}$ is not going to be a sub-Gaussian. This implies that to fulfill the assumption, whenever $\zeta_1$ is finite, $\epsilon_i$ should never be 0 or show extremely fast decay near 0, neither of which are standard settings in my opinion (e.g., even in simple Gaussian $\epsilon_i$ with p > 1, these conditions are not satisfied. For Gaussian, I think one can make my statement more explicit using the Chi-square density instead of the approximation).
    * If I am thinking wrong, can authors provide explicit examples when this condition is satisfied, possibly under the fairly standard setup?

* How Assumptions 1-4 are involved in the proof is not explicitly stated. I want to see where the sub-Gaussian condition on $r_i^{-1}$ is used.

* I am not sure whether ICLR is the right venue for this paper. I would expect such material in a statistics journal.

**Questions:**

* The authors claim elliptic distribution includes multivariate mixture in Line 35. But isn’t it false? I believe this will happen only in very special cases (e.g. same mean and covariance). Seems like authors used this setup in simulation. I think authors should clarify that only certain Gaussian mixtures are elliptic.

* Minor: use $\in$ instead of $=$ in Equation (1) and (2), as solutions may not be unique.

* The paper has not introduced what $\Omega$ is in Assumption 3. I assume it is a precision matrix. Am I correct?

* Assumptions 1 and 4: As mentioned in the weakness part, I would expect more explanations about assumptions. I believe Assumptions 2 and 3 are standard. However, Assumption 4 and the case $k = -1$ in Assumption 1 are not familiar to me. Particularly, I believe more explanations about Assumption 1 are needed, as I conjecture this assumption is never likely to be true except in very extreme cases (see the weakness part).

* The first $\lesssim$ part in Line 1120 is not clear. Can you elaborate on this?

* The authors claim Lemma 4 is the restatement of Theorem 1 in [1]. However, I do not see why. Particularly, $\widehat \Omega$ in Lemma 4 seems different from $\widehat V_{SCLIME}$ in [1]; based on the construction, if $\widehat \Omega$ is the precision matrix, then it depends on the choice of $h$ in the Algorithm 2, but $\widehat V_{SCLIME}$ does not have such bandwidth parameter. So these two cannot be fundamentally the same. Can you elaborate on this?

* Missing . in Line 351.

[1] Lu and Feng, “Robust Sparse Precision Matrix Estimation and its Applications”, Arxiv 2025.

---

> ### Author Response · Authors · 2025-11-28
>
> ### Response to weaknesses
>
> **Response:**
> We thank the reviewer for the detailed comments. For the first weakness, there is a slight misunderstanding of the spatial-sign setup. We define
> $\mathbf U_i = U(\boldsymbol\varepsilon_i)$, $r_i = |\boldsymbol\varepsilon_i|$, and $\zeta_k = \mathbb{E}(r_i^{-k})$. Importantly, $\zeta_1$ is *not* a fixed constant; in our high-dimensional regime $\zeta_1^{-1} = O(\sqrt{p})$.
>
> Consider the elliptical model
> $\mathbf X_i = \boldsymbol\mu + \boldsymbol\Sigma^{1/2}\boldsymbol\epsilon_i$ with $\boldsymbol\epsilon_i \sim t_v(\mathbf 0, I_p)$. We can write
> $$
> \boldsymbol\epsilon_i \overset{d}{=} \frac{\mathbf Z_i}{\sqrt{W_i/v}},\quad
> \mathbf Z_i \sim N(\mathbf 0, I_p),\quad W_i \sim \chi^2_v,
> $$
> independent. Then
> $$
> r_i = |\boldsymbol\epsilon_i| = v^{1/2}|\mathbf Z_i| W_i^{-1/2},\quad
> \sqrt{p}r_1^{-1} = v^{-1/2} W_1^{1/2}p^{1/2}|\mathbf Z_1|^{-1}.
> $$
> The factor $W_1^{1/2}$ has finite Orlicz–2 norm since $\chi^2_v$ is sub-exponential, and $p^{1/2}|\mathbf Z_1|^{-1}$ is sub-Gaussian by moment bounds. Hence $|\sqrt{p}r_1^{-1}|_{\psi_2} \le K_1 < \infty$ for some constant $K_1$, and together with $\zeta_1^{-1} = O(\sqrt{p})$ this yields the required sub-Gaussian behavior of $\zeta_1^{-1} r_1^{-1}$ in our elliptical setting.
>
> For the second weakness, both our algorithm and theory rely on a consistent initial precision estimator; the conclusion of Theorem 1 in Lu and Feng is used as an input. This is further discussed in our response to Question 4.
>
> For the third weakness, we stress that the second part of the paper goes beyond one-sample testing and develops a robust high-dimensional quadratic discriminant analysis (QDA). We construct an HR-based QDA rule by replacing classical sample means and covariances with HR estimates, improving stability under high-dimensional, heavy-tailed, or non-Gaussian designs. We provide theoretical guarantees and validate the method by simulations and real-data examples.
>
> ---
>
> ### Questions
>
> 1. **Response:**
>    A general multivariate Gaussian mixture is not elliptically symmetric. Ellipticity holds only when all mixture components share the same mean and their covariance matrices are proportional. In our simulations we use exactly this restricted case, so the resulting distributions are elliptical. We will revise the statement around Line 35 to clarify that only such mixtures are treated as elliptical.
>
> 2. **Response:**
>    For the spatial median, the objective is convex and, under standard strict-convexity conditions, has a unique minimizer. For the precision matrix, the loss plus penalties are convex under a positive-definiteness constraint, so the optimizer is unique as well. In this setting, writing Equations (1) and (2) with “$=$” is standard; we will add a short remark to make this explicit.
>
> 3. **Response:**
>    In Assumption 3, $\boldsymbol\Omega$ denotes the precision matrix (the inverse of the covariance matrix). We will revise the manuscript to introduce this notation clearly at its first appearance.
>
> 4. **Response:**
>    Assumptions 2 and 3 are standard, while Assumption 1 (especially $k=-1$) and Assumption 4 are more technical. In Assumption 1, the tail behaves like $t^{-p}$; in our high-dimensional regime ($p \to \infty$) this decay can be dominated by an exponential, which makes a sub-Gaussian-type bound reasonable for elliptical models such as multivariate $t$. Assumption 4 is taken from Lu and Feng and ensures consistency of the initial SCLIME-type precision estimator; we use their result as an input. We will slightly expand the discussion to clarify the role and typical validity of these assumptions.
>
> 5. **Response:**
>    The notation $f(x) \lesssim g(x)$ means that there exists a constant $c > 0$, independent of $(n,p)$, such that $f(x) \le c, g(x)$ on the range considered. Thus, on Line 1120 the first “$\lesssim$” indicates a bound up to a universal constant not depending on $n$ or $p$. We will add this explanation where the notation is first introduced.
>
> 6. **Response:**
>    We agree that $\hat{\boldsymbol\Omega}$ in Lemma 4 is not identical to $\hat{\mathbf V}_{\mathrm{SCLIME}}$ in Lu and Feng (2025). Lemma 4 is intended to restate the *type* of oracle bound established there, adapted to our banded-precision setting. In our algorithm, the spatial-sign covariance is banded with bandwidth $h$, but numerically the procedure is not very sensitive to $h$: when the initial value is reasonable,
>
>    $$
>    \frac{p}{n}\sum_{i=1}^n U(\hat{\boldsymbol\varepsilon}_i^{(k)}) U(\hat{\boldsymbol\varepsilon}_i^{(k)})^\top \approx I_p,
>    $$
>
>    and a small bandwidth simply projects this matrix onto the subspace where the true precision lies. Our modifications are reflected in line 1106.
>
> 7. **Response:**
>    We thank the reviewer for noticing the missing period; this typo will be corrected in the revision.

---

### Official Review · Reviewer_NwUT · 2025-11-03

**Soundness:** 3
**Presentation:** 2
**Contribution:** 3
**Rating:** 6
**Confidence:** 3

**Summary:**

This paper considers the problem of estimating the location and scale parameter of multivariate elliptical distribution. A classic affine-equivariant robust estimator is the so-called Hettmansperger–Randles (HR) estimator. But, it fails in the high dimension. The authors extend the Hettmansperger–Randles (HR) estimator to the high-dimensional regime and propose a computationally tractable version of the HR estimator by combining spatial-sign statistics and banded shrinkage. They establish consistency, derive a Bahadur representation, and prove Gaussian approximation results that enable inference.

**Strengths:**

1, The generalization of the HR estimator to high dimensional data seems to be novel and the proposed banded HR update (Algorithm 2) is sound and computationally implementable.

2, The authors establish rigorous theoretical development. They proved the Bahadur representation for the HR location estimator, the Gaussian approximation over convex sets, and the asymptotic independence between $T_{sum}$ and $T_{max}$, justifying the Cauchy combination test.

**Weaknesses:**

1, the method lacks theoretical guidance on the selection of bandwidth.

2, All results rely on the elliptical model. Simulation study lacks sensitivity analysis beyond elliptical data.

3, Competing robust methods (e.g., robust covariance shrinkage) are not included in experiments. Including at least one such baseline would make the empirical evaluation more comprehensive.

**Questions:**

1, can the authors provide some theoretical insight into bandwidth choice?

2, can the authors provide a sensitivity analysis beyond elliptical data? This would enhance credibility of the method.

3, can the authors include other robust estimators in the simulation study?

---

> ### Author Response · Authors · 2025-11-28
>
> ### Q1
> **Response:**
> In fact, our algorithm is not very sensitive to bandwidth choice. The bandwidth is introduced because
> $
> \frac{1}{n}\sum_{i=1}^n U(\hat{\varepsilon}_i^{(k)})  U(\hat{\varepsilon}_i^{(k)})^\top
> $ is not of full rank in the high-dimensional setting. Without any restriction, the subsequent update
> $\hat{\varepsilon}_i^{(k)} \gets {\hat{\Sigma}^{(k)}}^{-1/2}(X_i - \hat\mu^{(k)}) $
> may encounter matrix singularity. When the initial value is good,
>
> $\frac{p}{n}\sum_{i=1}^n U(\hat{\varepsilon}_i^{(k)})  U(\hat{\varepsilon}_i^{(k)})^\top \approx \mathbf I_p$
> so a small bandwidth is also acceptable.
>
> Intuitively, we know the true $S$ is sparse with signals concentrated near the main diagonal, so we propose $B_h$ to project $\hat S$ onto the subspace where its true value lies. Theoretically, choosing a small bandwidth helps ensure invertibility, while a larger bandwidth retains more of the information used in the original HR algorithm. In practice, however, the impact is modest, as shown by the numerical evidence in Appendix C.
>
> ---
>
> ### Q2
>
> **Response:**
> We agree that robustness beyond the idealized elliptical framework is important. Our theory is developed for high-dimensional elliptical models, which make spatial-sign based statistics and HR-type estimators tractable, but we expect the methods to remain useful under moderate deviations from ellipticity.
>
> To address this point without lengthening the main text, we have added a dedicated (blue-highlighted) contamination study in Appendix B. There we construct non-elliptical designs via $\varepsilon$-contamination of a heavy-tailed baseline, replacing an $\varepsilon$-fraction of observations by Gaussian noise with inflated variance. Over a grid of contamination rates and strengths, and under both sparse and dense alternatives, the tables in Appendix B show that empirical sizes stay close to the nominal level and that the procedures retain substantial power even when up to $20%$ of the sample is contaminated, providing empirical evidence of robustness to non-elliptical perturbations.
>
> A full asymptotic extension to general non-elliptical distributions would require more delicate control of covariance and spatial-sign covariance structures (in the spirit of spatial-sign based max–sum tests such as Liu–Feng–Wang, 2024) and, for our HR-type estimators, likely combining our framework with alternative robust covariance or precision estimators. We view this as an interesting direction for future work.
>
> ---
>
> ### Q3
>
> **Response:**
> We appreciate the reviewer’s suggestion. Many robust mean-testing procedures exist (Huber/Catoni-type estimators, Tyler’s shape-based methods, depth-based and MOM-type approaches), but most are *not directly applicable* to our high-dimensional heavy-tailed setting: they typically require low-to-moderate dimension, independence across coordinates, or prior knowledge of the scatter matrix, or they incur substantial computational cost in large Monte Carlo studies. A full comparison with all such methods would go beyond the scope of our spatial-sign framework.
>
> To provide concrete evidence, we implemented a representative MOM-type test in the same $t_3$ setting as our main simulations (details given in the blue-highlighted part of Appendix B). The null is $\mu=0_p$ (“size”), and we consider three alternatives
> $$
> \mu_1 = (1,1,1,0,\ldots,0)^\top,\quad
> \mu_2 = \Sigma^{1/2}(1,1,1,0,\ldots,0)^\top,\quad
> \mu_3 = 0.1\cdot\mathbf{1}_p,
> $$
> corresponding to sparse/directionally sparse/dense signals. Empirical rejection probabilities over 500 replications are:
>
> |         | $T_{\max}$ | $T_{\mathrm{sum}}$ | $T_{\mathrm{cc}}$ | $T_{\mathrm{mom}}$ |
> | ------- | ---------: | -----------------: | ----------------: | -----------------: |
> | size    |      0.052 |              0.046 |             0.054 |              0.064 |
> | $\mu_1$ |      0.102 |              0.176 |             0.164 |              0.062 |
> | $\mu_2$ |      0.564 |              0.446 |             0.590 |              0.076 |
> | $\mu_3$ |      0.260 |              0.824 |             0.756 |              0.258 |
>
> The MOM-type test has slightly inflated size (0.064 versus the nominal $0.05$) and substantially lower power in the sparse settings $\mu_1,\mu_2$, and is comparable to the Cauchy combination only under the dense alternative $\mu_3$. By contrast, our sum, max, and Cauchy statistics maintain good size and achieve higher or comparable power across all three alternatives. Together with the robust baseline $T_{\mathrm{cc}2}$ from prior spatial-sign work (reported in Figure 2), this suggests that our procedures are competitive with both sign-based and MOM-type robust tests in the high-dimensional heavy-tailed regime, so we focus our detailed comparisons on $T_{\mathrm{cc}2}$ and the spatial-sign family.

---

### Comment · Area_Chair_rYf3 · 2025-11-25

Dear Reviewers,
Thank you to those who have already begun interacting with the authors — your timely follow-ups are greatly appreciated and reflect the professionalism and care that uphold our community’s standards.
For reviewers who have not yet responded, I would like to offer a gentle reminder. Authors have invested substantial time and effort into preparing their rebuttals, carefully addressing each concern raised in the reviews. As fellow researchers, we all understand the importance of being heard and having our clarifications considered. Even a brief acknowledgment or follow-up question helps ensure that the evaluation remains fair, thorough, and respectful of everyone’s work.
Your engagement during this phase is essential for maintaining a constructive and high-quality review process. Thank you again for your service and for treating both your fellow reviewers and the authors with the same consideration you would hope to receive.
Best regards,
AC

---

### Note · Program_Chairs · 2026-01-07
**Submission Desk Rejected by Program Chairs**

This submission has manipulated the ICLR template to have smaller margins and must be desk rejected.